# Calibrated rare variant genetic risk scores for complex disease prediction using large exome sequence repositories

Ricky Lali[1,2], Michael Chong[1,3], Arghavan Omidi[1], Pedrum Mohammadi-Shemirani [1,4], Ann Le[1,4], Edward Cui[1] & Guillaume Paré [1,2,3,4,5,6,7✉]

Rare variants are collectively numerous and may underlie a considerable proportion of complex disease risk. However, identifying genuine rare variant associations is challenging due to small effect sizes, presence of technical artefacts, and heterogeneity in population structure. We hypothesize that rare variant burden over a large number of genes can be combined into a predictive rare variant genetic risk score (RVGRS). We propose a method (RV-EXCALIBER) that leverages summary-level data from a large public exome sequencing database (gnomAD) as controls and robustly calibrates rare variant burden to account for the aforementioned biases. A calibrated RVGRS strongly associates with coronary artery disease (CAD) in European and South Asian populations by capturing the aggregate effect of rare variants through a polygenic model of inheritance. The RVGRS identifies 1.5% of the population with substantial risk of early CAD and confers risk even when adjusting for known Mendelian CAD genes, clinical risk factors, and a common variant genetic risk score.

[1] Population Health Research Institute, David Braley Cardiac, Vascular and Stroke Research Institute, 237 Barton Street East, Hamilton, ON L8L 2X2, Canada. [2] Department of Health Research Methodology, Evidence, and Impact, McMaster University, Faculty of Health Sciences, 1280 Main Street West, Hamilton, ON L8S 4K1, Canada. [3] Department of Biochemistry and Biomedical Sciences, McMaster University, Faculty of Health Sciences, 1280 Main Street West, Hamilton, ON L8S 4K1, Canada. [4] Department of Medical Sciences, McMaster University, Faculty of Health Sciences, 1280 Main Street West, Hamilton, ON L8S 4K1, Canada. [5] Thrombosis and Atherosclerosis Research Institute, David Braley Cardiac, Vascular and Stroke Research Institute, 237 Barton Street East, Hamilton, ON L8L 2x2, Canada. [6] Department of Pathology and Molecular Medicine, McMaster University, Michael G. DeGroote School of Medicine, 1280 Main Street West, Hamilton, ON L8S 4K1, Canada. [7] Department of Clinical Epidemiology & Biostatistics, McMaster University, 1280 Main Street West, Hamilton, ON L8S 4K1, Canada. ✉email: pareg@mcmaster.ca

Rare variants have been hypothesized to contribute significantly to the missing heritability of complex diseases[1,2], raising the possibility of complementing a common variant genetic risk score (CVGRS) with a rare variant score. However, the ability to identify robust associations between rare variants and complex diseases is inherently limited by sample size. Though novel statistical techniques have been developed to conduct rare variant association, which involve collapsing all rare variants within a given genetic unit (i.e., a gene or gene-set) into a single count[3–7], the power to detect rare variant associations of modest effect remains limited. Large publicly available sequencing consortia can be employed as control databases in order to markedly increase power. However, there is currently no method to calibrate rare variant burden between test samples and public databases despite both population and sequencing differences, thus limiting the ability to construct a well-powered rare variant genetic risk score (RVGRS).

The recent public releases of the Exome Aggregation Consortium (ExAC)[8] and genome Aggregation Database (gnomAD)[9] have allowed users to obtain precise allele frequency estimates in many different ethnic populations. In fact, ExAC and gnomAD are often used to distinguish between benign and pathogenic variants present in single cases or families based on rarity[10]. The summary-level data contained within these public sequencing databases can also be leveraged beyond single case and pedigree analysis. Specifically, the allele frequency information in these databases is representative of the expected allele frequencies in the general population, which can be utilized as a control distribution for rare variant association studies. However, incorporating external databases into rare variant association testing presents many population-specific and technical challenges. Specifically, populations in both ExAC and gnomAD are geographically diverse (even within single continents) and consist of sample pools that utilize different exome capture chemistries and variant calling algorithms which may be distinct from those used in a local case population. Consequently, adjustments to gene-based expected frequencies are required in order to mitigate artefactual association signals. These adjustments are especially vital for rare variants as they have shown to exhibit a higher degree of geographical specificity compared to common variants[11–13], and accuracy of calling rarer variants largely depends upon the sample size for joint variant calling algorithms[14]. As such, sources of heterogeneity in both population substructure and sequencing technology must be addressed to leverage the full capabilities of publicly available sequencing datasets as external control datasets.

Here, we propose a methodological framework, Rare Variant Exome CALIBration using External Repositories (RV-EXCALIBER) (workflow described in Fig. 1), which leverages the large sample size of gnomAD as a control dataset to conduct rare variant association testing. In brief, RV-EXCALIBER works by instituting (1) a correction that accounts for the increased variance due to the presence of rare variants in LD, (2) an individual-level correction factor (iCF) that accounts for global variations in population substructure and sequencing technology and (3) a gene-level correction factor (gCF) that accounts for granular deviations in mutation burden bias amongst gene sets. As such, the iCF and gCF act by calibrating the total allele counts in gnomAD (control) to an input case dataset on a sample-by-sample basis on the whole-exome level and gene-set level, respectively. Through simulations, we demonstrate that RV-EXCALIBER adequately adjusts for sequencing false negatives and population-specific effects while also partially mitigating power loss induced by sequencing false positives. We also demonstrate the necessity for rare variant calibration by comparing deviations in total allele count between Genome In A Bottle (GIAB) consortium reference samples and gnomAD.

Lastly, we demonstrate an application for our framework through constructing individually calibrated rare variant gene risk scores (RVGRS) for a prevalent complex disorder, coronary artery disease (CAD).

## Results

**Single gene simulations using the correction factor (CF).** In order to evaluate the effect of incorporating the CFs on rare variant gene-based association, we conducted a series of simulations and determined the CF's effect on (1) estimated odds ratio (OR) under strong confounding conditions of sequencing false negatives (SFN), sequencing false positives (SFP), and population-specific factors (PSF) (Fig. 2a–c) and (2) power to detect association according to SFN (0–1), SFP (0–1), and PSF (0–2) under a model of association assuming a true OR of 1.3 (Fig. 2d–f). To accomplish this, we performed 100,000 simulations of aggregate rare allele count across 1000 cases for a single hypothetical gene by varying the baseline aggregate probability of mutation ($P$) (i.e., 5%, which is a reasonable estimate given empirically-derived expected allele counts in gnomAD (EAC)) based on true OR, SFN, SFP, and PSF. We found that incorporation of the CFs fully and significantly corrected the estimated OR to match the true OR when including either sequencing false negatives or population structure (Supplementary Fig. 1a and c). Conversely, we observed that incorporation of the CF over-corrects (i.e., estimated OR < true OR) when sequencing false positives are present (Supplementary Fig. 1b). We hold that the adjustment is still effective in the latter scenario given that CF-adjusted effect size of the gene-based association will be conservative. We also observed that the CF robustly calibrates the estimated OR in the presence of all biases simultaneously (i.e., SFN, SFP, and PSF) and adheres to the same patterns as observed with single bias simulations (Fig. 2a–f). That is, the CF adequately calibrates the estimated OR to the true OR in the presence of strong SFN and PSF bias, while providing a slightly conservative correction at high SFP rates (Fig. 2c and f). While these simulations demonstrate strong calibration of effect size under realistic PSF bias for rare variants, we show that the extent of calibration remains robust when such bias is more extreme (Supplementary Fig. 2a–f).

To estimate the power to detect a gene-based association, we similarly simulated 100,000 observed rare allele counts assuming a true OR of 1.3, which was the effect size necessary to achieve 80% power when no confounding factors exist (i.e., SFN & SFP rates of 0% and a PSF of 1) using our model (Supplementary Fig. 1d–f). Under the setting where sequencing false negatives are present, we expect a decrease in mutation rate and thus, power to detect association. Incorporation of the CF mitigates the reduction in power as we observe increased power to detect association across all SFN rates in the CF-adjusted compared to the unadjusted model. When only sequencing false positives are present, we show that incorporation of the CF ameliorates the spurious increase in power observed as mutation rate rises due to increasing rates of SFPs. It is important to note that the artificial increase in power observed in the unadjusted model is accredited solely to sequencing artefacts and not true variants, and thus would translate directly into increased type I error. Lastly, we observe an increase in power as the PSF becomes greater (i.e., the total allele count in a test sample is greater than a reference sample due to reasons such as population substructure). In this setting, the power increase is expected as the mutation rate is increasing due to variants that are truly present because of population genetic factors, as opposed to variant artefacts observed in the unadjusted model including SFP. These features were also observed when the power to detect gene-based

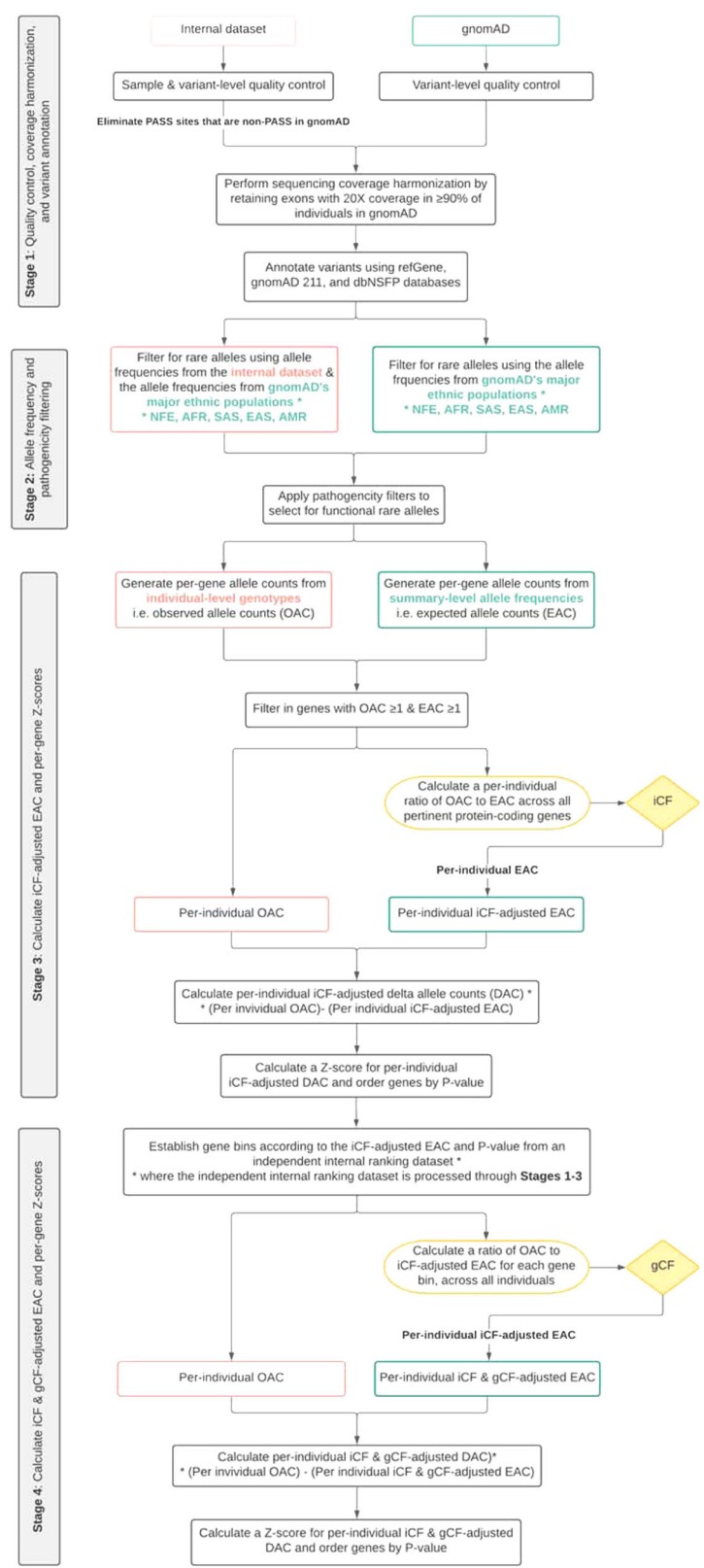

**Fig. 1 Schematic describing the RV-EXCALIBER pipeline.** RV-EXCALIBER can be divided into 4 distinct stages where stage 1 and 2 are preprocessing steps and stages 3 and 4 encapsulate the three major RV-EXCALIBER adjustments: the implementation of the correction factors (iCF and gCF) and Z-score calculations for per-gene delta counts. MAF indicates minor allele frequency, iCF indicates individual correction factor, gCF indicates gene correction factor, NFE indicates non-Finnish European, AFR indicates Africa, SAS indicates South Asian, EAS indicates East Asian, and AMR indicates Latino.

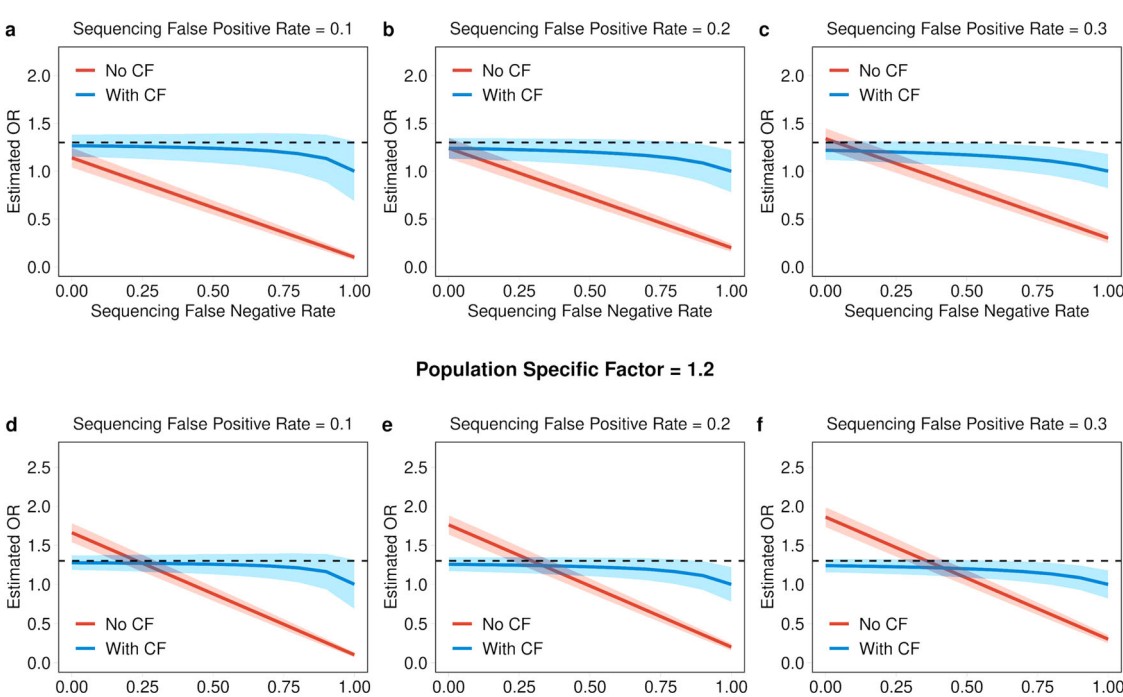

**Fig. 2 Simulated effects of combined SFN rate, SFP rate, and PSF on estimated OR.** Probability of mutation ($P$) was calculated as a function of the cumulative minor allele frequency for a single gene ($CMAF_{Gene}$: set at 0.05), the true effect size of association (OR), and the 3 major association biases (SFN, SFP, and PSF) according to Eq. (2). The simulations kept the true OR fixed at 1.3 (black dashed lines) while varying SFN rate (0–1), SFP rate (0.1, 0.2, and 0.3), and modest PSF (0.8 and 1.2). A mean estimated OR was calculated for each $P$ (corresponding to a unique combination of the 3 association biases) and plotted as a function of the combined biases (**a–f**; red lines) across 100,000 simulations. CF values were calculated as the ratio of the $P$ for each estimated OR and the original $CMAF_{Gene}$ according to Eq. (3). Each CF was then used to adjust the $CMAF_{Gene}$ to calculate an adjusted $P$ ($P^*$) according to Eq. (4). Using $P^*$, a CF-adjusted mean estimated OR was plotted as a function of the combined biases (**a–f**; blue lines) across 100,00 simulations. All shaded regions correspond to the 95% confidence interval for the mean estimated OR. OR indicates odds ratio, CF indicates correction factor, SFN indicates sequencing false negatives, SFP indicates sequencing false positives, PSF indicates population-specific factor. Equations are defined in the Methods section. Source data are provided as a Source data file.

association signals was assessed in the presence of SFN rate, SFP rate, and PSF simultaneously (Supplementary Fig. 3a, b).

**GIAB consensus sequences highlights the role of population substructure in total allele count for rare variants.** As rare variants demonstrate a higher degree of geographical specificity and are under greater selective pressure than common variants, we sought to assess the impact of population substructure on rare variant burden through use of gold-standard consensus GIAB sequences. When comparing the Ashkenazi Jewish and North-western European consensus sequences with the non-Finnish European (NFE) population in gnomAD, we observed significant deviations in iCF for rare variants ($0 < MAF \leq 0.01$) (Ashkenazi: iCF = 1.37; 95% CI, 1.22 to 1.55; $p < 0.05$ and North-western European: iCF = 1.20; 95% CI, 1.07–1.36; $p < 0.05$), where a iCF = 1 represents equal total allele counts. In contrast, deviations in iCF between these consensus sequences and gnomAD were attenuated among higher MAF bins (Fig. 3 and Supplementary Table 3). To test whether iCF deviations for rare variants would dissipate when calibrated against a more closely matched comparator population, we evaluated the iCF between the Ashkenazi Jewish consensus sequence and the gnomAD ASJ population and the East Asian consensus sequence with the gnomAD EAS population. In these scenarios, the observed iCFs closely approximated the expected iCF across all MAF threshold bins, but rare variants still deviated most from the expected iCF relative

to the remaining MAF bins (Fig. 3b, d and Supplementary Table 3).

Since the iCFs deviated mostly from the expected values among rare variants for every population comparison, we wanted to rule out the possibility that the population effects driving these deviations were restricted to non-constrained genes as this would preclude use of a single iCF to equilibrate total allele counts at the exome-wide level. To accomplish this, we allocated genes into deciles according to their missense constraint score[8] and assessed whether there was significant heterogeneity among the iCFs across all deciles. For each consensus sequence versus gnomAD comparison, we observed no significant heterogeneity after accounting for multiple hypotheses testing (pHet > 0.0125 (0.05/4) for all comparisons) (Supplementary Fig. 5).

**iCF distribution among healthy controls participants from the Myocardial Infarction Genetics exome sequencing consortium (MIGen).** A total of 8 whole-exome sequencing (WES) cohorts were selected from MIGen (Supplementary Table 1) as a discovery set for rare variant burden association ($n = 5910$ for cases and $n = 6082$ for controls). All analyses concerning the MIGen discovery cohorts were conducted using rare, pathogenic alleles (i.e., variants with a minor allele frequency <0.001 and non-synonymous single nucleotide variants (SNV) with a Mendelian clinically applicable pathogenicity score >0.025 or loss-of-function variants; see Supplementary Note 7, section D). iCFs were calculated for healthy control participants from each cohort

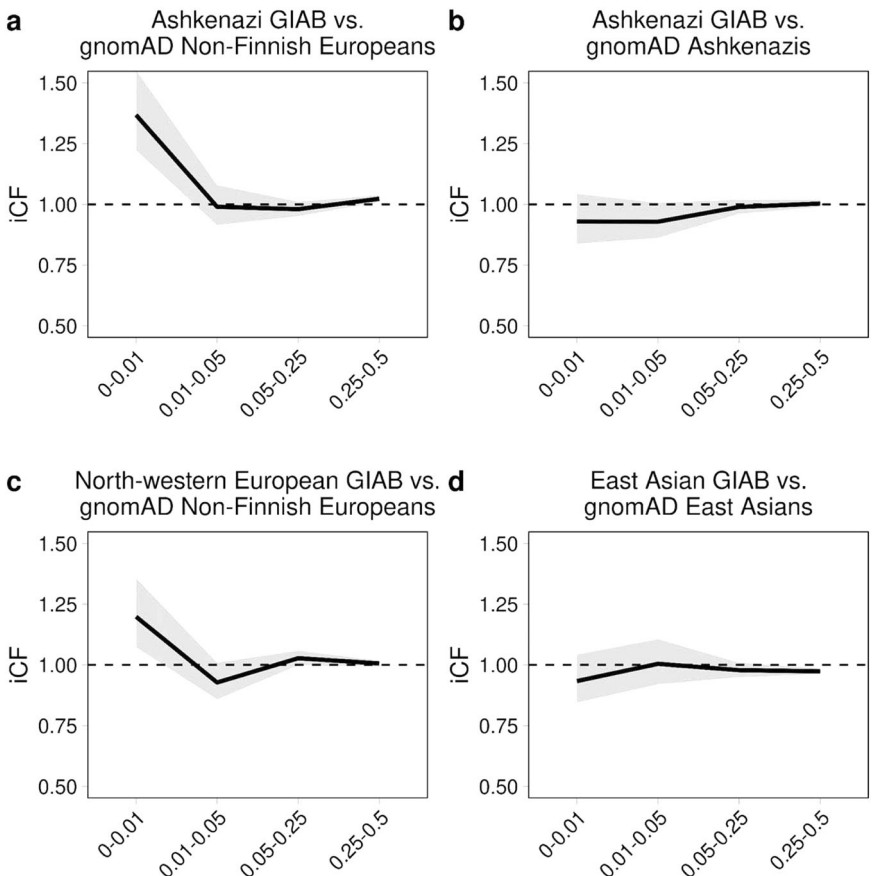

**Fig. 3 Comparison of iCF values for consensus GIAB samples stratified by allele frequency bin.** Ethnic comparisons include the Ashkenazi GIAB sample (NA24385) with gnomAD Non-Finnish Europeans and gnomAD Ashkenazis (**a**, **b**, respectively), the North-western European GIAB sample (NA12878) with gnomAD Non-Finnish Europeans (**c**), and the East Asian GIAB sample (NA24631) with gnomAD East Asians (**d**). iCF values were calculated as the exome-wide ratio of the sum of per-individual observed allele counts (OAC) to the sum of per-individual expected allele counts in gnomAD (EAC) according to Eq. (5) for all exonic alleles. Allele frequency bins were chosen to best stratify variants according to rare [0–0.01], low-frequency [0.01–0.05], common [0.5–0.25] and very common [0.25–0.5] bins. Allele frequencies were standardized to the minor allele. All shaded regions depict the 95% confidence interval of the iCF at each AF bin. Dashed lines indicate an iCF representing equal total allele counts between the GIAB sample and gnomAD (i.e., iCF = 1) across all protein-coding genes. iCF indicates individual correction factor, and GIAB indicates Genome In A Bottle, and gnomAD indicates genome aggregation database. Equations are defined in the "Methods" section. Source data are provided as a Source data file.

and their distributions were plotted to identify whether there were marked inter and intra-cohort differences in total allele count. Across the 8 MIGen cohorts, the highest and lowest median iCF values were from the Italian Atherosclerosis Thrombosis and Vascular Biology (ATVB) (median = 1.135; IQR, 0.909–1.363) and the Malmö Diet Cancer (MDC) studies (median = 0.664; IQR, 0.487-0.841), respectively (Supplementary Table 4). All pairwise between cohorts comparison of iCF among control participants were found to be significantly different, even after adjusting for sex and the first 20 principle components, which were calculated based on common SNVs (Fig. 4a). iCF distributions were also compared between cases and controls of individual MIGen cohorts, where significant differences were identified in 2 out of 8 studies. Specifically, iCF values were significantly lower among controls from the Registre Gironi del Cor (REGICOR) cohort (OR = 0.33 per 1 unit change in iCF; 95%CI, 0.12-0.92; P = 0.0352), but higher for MDC controls (OR = 2.44; 95%CI, 1.01-5.96; P = 0.0492) (Supplementary Fig. 6 and Supplementary Table 5).

**RV-EXCALIBER adequately calibrates associations using healthy controls participants from MIGen.** The efficacy of the gCF is largely determined on the degree of similarity between a

test and ranking cohort in terms of sequencing technology and population structure. In order to evaluate the adequacy of gCF adjustment in the absence of disease associations, we randomly divided healthy unrelated controls from MIGen (n = 6082) into a ranking cohort (n = 2730) in which the gCF would be calculated based on the distribution of gene-based P-values and EAC and a test cohort (n = 3352) in which the gCF would be applied (see Supplementary Information). 11,616 genes with observed and EAC of ≥1 were allocated into 50 distinct gCF bins (see Supplementary Information), corresponding to 232 genes per bin. gCF values ranged from 0.89 (bin 30) to 1.25 (bin 10) in the ranking cohort, with a significant correlation between bin index and gCF. In other words, genes that were depleted and enriched for pathogenic alleles in gnomAD trended toward higher and lower gCF values, respectively ($R^2 = 0.57$; $P = 2.1 \times 10^{-10}$) (Supplementary Fig. 7). Adjustment for gCF further attenuated deviations in observed versus EAC as compared to what is achievable using the iCF alone (Fig. 4b).

After establishing the iCF and gCF adjustments, we sought to compare the genomic inflation factors (i.e., lambda) at the median ($\lambda_{med}$) using the RV-EXCALIBER and Testing Rare variants Against Public Data (TRAPD)[15] methods in order to assess the presence of type I error. For TRAPD, inclusion of rare pathogenic nonsynonymous SNVs was based on the top 94th

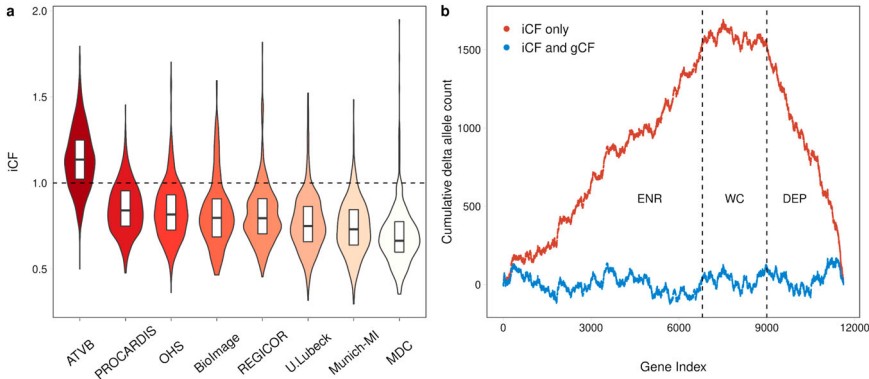

**Fig. 4 Correction factor adjustment between studies and across genes.** Distribution of iCFs for 6,082 healthy controls from 8 cohorts in MIGen were determined according to rare, pathogenic alleles across all protein-coding genes (**a**). iCF values were calculated as the exome-wide ratio of the sum of per-individual observed allele counts (OAC) to the sum of per-individual expected allele counts in gnomAD (EAC) according to Eq. (5). The violins demonstrate the spread of iCF among all 8 MIGen cohorts. The horizontal line in each boxplot indicates the median iCF values while the top and bottom lines represent the 75th and 25th percentiles of the iCF distribution, respectively. Length of boxplot represents the inter-quartile range of iCF values. Dashed line represents an iCF corresponding to equal total allele counts between a MIGen participant and gnomAD (i.e., iCF = 1). gCF values were computed as the ratio of the sum of OAC to the sum of iCF-adjusted EAC across (1) all individuals and (2) all genes that were organized into one of 50 gene bins according to Eq. (7). Gene bins were ascertained according to quintile of iCF-adjusted EAC and decile of *P*-value obtained from a rare variant association test (using burden of rare, pathogenic alleles) conducted in the ranking cohort, consisting of the remaining 2730 healthy control participants in MIGen as "cases" and gnomAD non-Finnish Europeans as controls. The iCF and gCF-adjusted EAC was thereafter calculated according to Eq. (8). The cumulative delta allele count was calculated from the cumulative sum of the per-gene difference between OAC and either the iCF-adjusted EAC (red points) or iCF and gCF-adjusted EAC (blue points) according to Eq. (9). The cumulative delta allele count demarcates genes that are systematically enriched (ENR), well-calibrated (WC), and depleted (DEP) for rare, pathogenic alleles among MIGen control participants (**b**). The height of the mountain demonstrates the degree of adjustment offered by the iCF to achieve calibration between MIGen controls and gnomAD. Implementing the gCF mitigates residual gene-level biases that cannot be accounted for by the iCF alone. iCF indicates individual correction factor, and gCF indicates gene correction factor. Equations are defined in the "Methods" section. Source data are provided as a Source data file.

percentile of quality-by-depth (QD) score according to rare synonymous SNVs in 3352 healthy control exomes in MIGen and top 90th percentile of QD scores in gnomAD, which achieved the best calibration between the 95th percentile test statistics ($\lambda_{95}$) and a uniform distribution of *P*-values ($\lambda_9 = 0.99$) (Supplementary Fig. 8). Inclusion of insertions and deletions were based on the top 96th percentile QD scores for all rare nonsynonymous SNVs in 3352 healthy control exomes in MIGen and top 90th percentile of QD scores in gnomAD ($\lambda_9 = 1.02$). After applying TRAPD and RV-EXCALIBER to 4160 protein coding genes that had observed allele counts (OAC) ≥ 1 and EAC ≥ 10 using both methods (to forego potential biases due to genes with low power to detect rare variant associations), RV-EXCALIBER achieved better calibration at the median of test statistics ($\lambda_{med} = 0.95$ and 0.41 for RV-EXCALIBER and TRAPD, respectively) (Fig. 5). In order to further benchmark RV-EXCALIBER and TRAPD, we assessed presence of residual bias by measuring the dispersion of the per-gene difference in observed and EAC (i.e., delta allele count) as a function of EAC for all genes with observed and EAC ≥ 1 in the same 3352 healthy control exomes in MIGen. Across 10,788 genes that met this count threshold for both methods, we identified an increasing deflation bias in percent delta allele count for TRAPD as gene-based EAC increased (Supplementary Fig. 9b). In contrast, RV-EXCALIBER remained unbiased, having a mean percent delta near zero across all quintiles of EAC (Supplementary Fig. 9a).

We next sought to assess the contribution of each of the three proposed adjustments to type I error control using the same 3352 healthy controls in MIGen as "cases" and gnomAD non-Finnish Europeans as controls. Genomic inflation estimates at the median were better calibrated when using RV-EXCALIBER base (which only accounts for LD between rare variants and does not incorporate iCF or iCF and gCF-adjustment to the EAC) compared to a Fisher's Exact test across 4815 protein-coding genes ($\lambda_{med} = 0.69$ and 0.57, respectively). We also demonstrated that calibration of test statistics improves in a step-wise fashion upon incorporating each additional feature of RV-EXCALIBER. Specifically, $\lambda_{med}$ estimates incrementally approach 1 using (i) RV-EXCALIBER base, (ii) RV-EXCALIBER base + iCF adjustment, and (iii) RV-CALIBER base + iCF + gCF adjustment (i.e., the fully adjusted model) (Supplementary Figs. 10 and 11).

**Rare variant association.** After adjusting gnomAD EAC with RV-EXCALIBER, we conducted an exome-wide rare variant association on protein-coding genes across 5910 discovery unrelated MI cases from the MIGen consortium, testing all 7146 genes with OAC ≥ 1 and EAC of ≥10 (again, to filter out underpowered genes). RV-EXCALIBER adequately calibrates gene-based test statistics ($\lambda_{med} = 0.92$) (Fig. 6). The low-density lipoprotein receptor (*LDLR*) was identified as the only exome-wide significant signal (OR = 2.25; 95% CI, 1.70–3.05; $P = 9.4 \times 10^{-13}$), with OAC of 160 and an adjusted EAC of 71.1. Among the 160 observed alleles, 111 were located at distinct exonic sites including 99 (89.2%) missense variants, 10 (9.0%) nonsense variants, 1 (0.9%) frameshift insertion, and 1 (0.9%) frameshift deletion (Supplementary Data 2). We also noted a suggestive signal in tyrosine protein kinase Fes/Fps (*FES*) (OR = 2.00; 95% CI, 1.20–3.42; $P = 2.5 \times 10^{-4}$) which has previously been identified as a genome-wide significant loci for CAD in multiple studies (Supplementary Data 3). In fact, a total of 35 (excluding *LDLR*) of the 7146 genes assessed in the exome-wide rare variant association were found to harbor a genome-wide significant single nucleotide polymorphism for CAD (i.e., a CAD gene set) in the CARDIoGRAMplusC4D genome-wide association study. After generating a distribution of Fisher's combined probability test statistics for 100,000 random gene sets of equivalent size under the null of no individual gene associations, we identified that the probability of observing a combined test

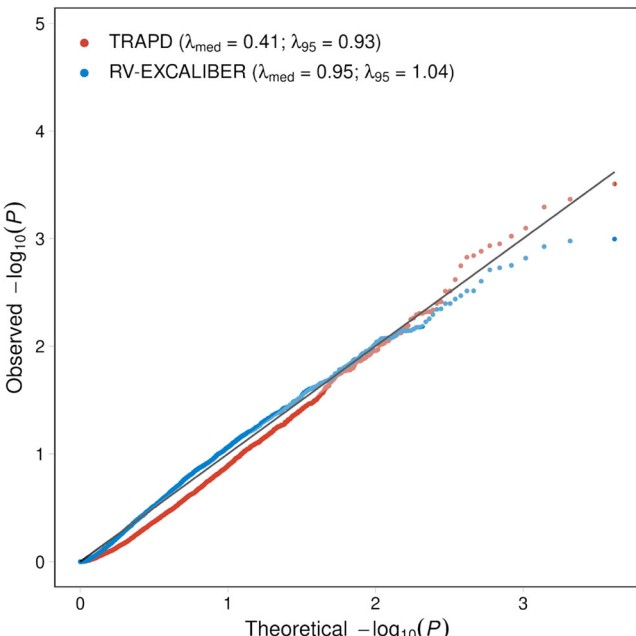

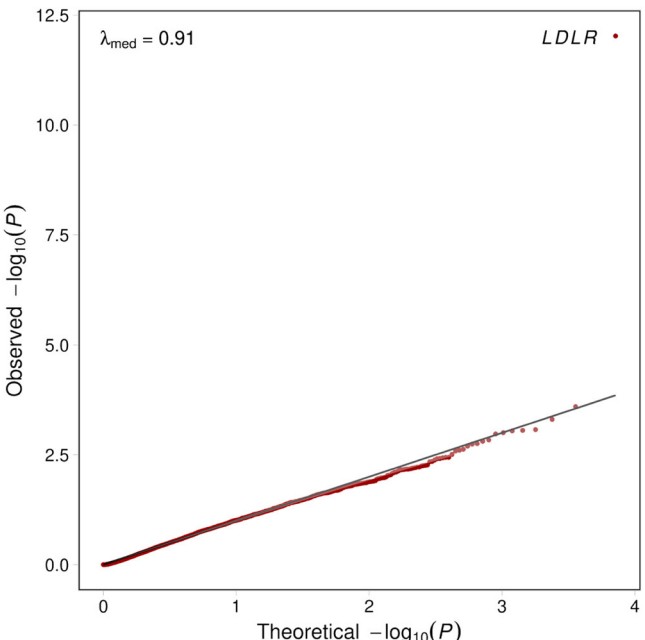

**Fig. 5 Quantile-quantile plot for gene-based test statistics generated using TRAPD and RV-EXCALIBER for 3352 healthy control participants in MIGen.** A total of 4160 genes with OAC ≥1 in MIGen and EAC ≥10 in gnomAD non-Finnish Europeans were used to conduct gene burden testing using rare, pathogenic alleles according to the TRAPD (red points) and RV-EXCALIBER (blue points) methods. To gauge calibration of test statistics between methods, 3352 healthy MIGen control participants were compared to gnomAD non-Finnish Europeans. The remaining subset of 2730 MIGen healthy controls were used as the gCF ranking cohort. The genomic inflation estimates were evaluated at the median ($\lambda_{med}$ and 95th percentile ($\lambda_{95}$) of test statistics. The solid black line indicates an expected uniform distribution of $P$-values under the null and the shaded region represents the 95% confidence interval of the expected uniform distribution of $P$-values. $\log_{10}(P)$ refers to the log base 10 of a gene-based $P$-value. Source data are provided as a Source data file.

**Fig. 6 Quantile-quantile plot for discovery gene-based test statistics generated using RV-EXCALIBER for 5910 case participants from MIGen.** A total of 7146 protein-coding genes with OAC ≥1 in MIGen and EAC ≥10 in gnomAD non-Finnish Europeans were used to conduct a gene burden testing with the RV-EXCALIBER method. Association testing was conducted using 5,910 MI cases in MIGen and gnomAD non-Finnish Europeans as controls. The genomic inflation estimate at the median ($\lambda_{med}$ was determined to be 0.91. The solid black line indicates an expected uniform distribution of $P$-values under the null and the shaded region represents the 95% confidence interval of the expected uniform distribution of $P$-values. $\log_{10}(P)$ refers to the log base 10 of a gene-based $P$-value. Source data are provided as a Source data file.

statistic greater than or equal to that of the CAD gene set was statistically significant ($P = 0.014$). We also elected to conduct a gene-based enrichment analysis using genes that were nominally (i.e., $P < 0.05$) associated with CAD in the discovery rare variant association to elucidate if any Gene Ontology (GO) biological processes were enriched. Although no biological processes reached statistical significance after multiple hypothesis correction, we observed nominal enrichment of "reactive oxygen species biosynthetic process" (GO:1903409) (fold enrichment=6.82; $P = 1.47 \times 10^{-4}$), "response to stress" (GO:0006950) (fold enrichment = 1.40; $4.74 \times 10^{-4}$), "response to oxygen-containing compound" (GO:1901700) (fold enrichment=1.58; $P = 5.39 \times 10^{-4}$), and "nitric oxide biosynthetic pathway" (GO:0006809) (fold enrichment=8.19; $P = 9.30 \times 10^{-4}$). A complete list of nominally associated biological processes can be found in Supplementary Table 6.

**A calibrated RVGRS for CAD has consistent predictive effects in European and South Asian populations.** To test whether a calibrated RVGRS could predict CAD, we calculated RVGRSs that weighted rare, pathogenic alleles using the adjusted gene-based effect size estimates that were generated using RV-EXCALIBER. Each RVGRS was calculated in each case ($n = 3843$) and control ($n = 42,007$) in unrelated UK Biobank European participants, incrementally including the top 10 to top 3,000 genes (based on $P$-value) from the MIGen association

(increments of 10 genes; 300 total scores). The RVGRS generated from the top 950 discovery genes (RVGRS950) best discriminated participants with CAD from healthy controls, with a 1.08-fold (95% CI, 1.04–1.11; $P = 2.1 \times 10^{-5}$) increased odds for developing CAD relative to healthy control subjects per 1 SD change in RVGRS (Figs. 7 and 8). As a sensitivity analysis, we removed genes known to confer risk for familial hypercholesterolemia (FH) in order to ensure that the predictive estimate of the RVGRS950 was not due to single-gene Mendelian effects. After removing *LDLR* (which was the only FH gene within RVGRS950; i.e., RVGRS950$^{LDLR-}$), a consistent predictive effect for CAD (OR = 1.07; 95% CI, 1.03–1.10; $P = 1.5 \times 10^{-4}$) was observed (Fig. 7). A major pitfall of common variants gene scores is the lack of transferability across ancestries. Although RVGRS950 did not reach significance in unrelated South Asians from the Pakistan Risk of Myocardial Infarction Study (PROMIS) ($n = 2946$ for cases and $n = 3708$ for controls), there was still a similar predictive effect observed (OR = 1.04; 95% CI, 0.99–1.08; $P = 0.096$). Nevertheless, a RVGRS generated using the top 680 genes (RVGRS680) did validate in PROMIS (OR = 1.06; 95% CI, 1.01–1.11; $P = 0.024$) with a consistent effect in the UK Biobank (OR = 1.06; 95% CI, 1.03–1.10; $P = 5.9 \times 10^{-4}$). In fact, we identified that the effect estimates generated from all RVGRS (i.e., top 10 to top 3000 genes) did not significantly differ between the UK Biobank and PROMIS according to a test for heterogeneity (Fig. 8 and Supplementary Data 4). As a negative control, we tested the same 300 RVGRS on UK Biobank participants with irritable bowel disease ($n = 3058$) and identified no significant difference in distribution of RVGRS between case and control

| Gene score | N cases / N controls | OR per SD | 95% CI | P-value |
|---|---|---|---|---|
| RVGRS950 | 3,843 / 42,007 | 1.08 | 1.04-1.11 | $2.1 \times 10^{-5}$ |
| RVGRS950$^{LDLR-}$ | 3,843 / 42,007 | 1.07 | 1.03-1.10 | $1.5 \times 10^{-4}$ |
| RVGRS950 with full adjustment | 3,843 / 42,007 | 1.07 | 1.04-1.11 | $2.9 \times 10^{-5}$ |
| RVGRS950$^{LDLR-}$ with full adjustment | 3,843 / 42,007 | 1.06 | 1.03-1.10 | $2.3 \times 10^{-4}$ |

**Fig. 7 Predictive effects of RVGRS950 and RVGRS950$^{LDLR-}$ on CAD among UK Biobank participants.** Odds ratios of RVGRS950 and RVGRS950$^{LDLR-}$ (i.e., RVGRS950 without *LDLR*) on CAD are shown for UK Biobank European participants. The same RVGRS with additional adjustment for a Framingham risk score and common variant genetic risk score (i.e., with full adjustment) are also shown. Squares represent the odds ratios that are expressed in terms of a 1 SD change in RVGRS. All odds ratios were calculated using a multivariable logistic regression model that was adjusted for age, age$^2$, sex, and the first 20 principal components of ancestry. Horizontal lines indicate the 95% confidence interval of the odds ratio. OR indicates odds ratio, SD indicates standard deviation, CI indicates confidence interval, RVGRS950 refers to rare variant genetic risk score from the top 950 discovery genes, and RVGRS950$^{LDLR-}$ refers to rare variant genetic risk score from the top 950 discovery genes, excluding *LDLR*. *LDLR* refers to low-density lipoprotein receptor. Source data are provided as a Source data file.

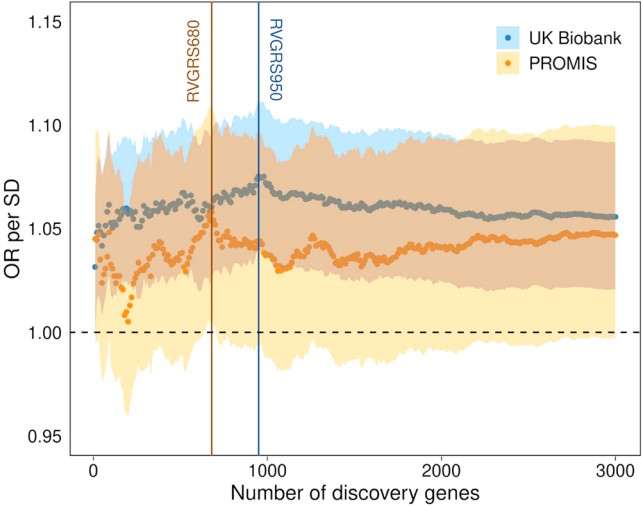

**Fig. 8 Predictive effect of RVGRS on CAD evaluated in UK Biobank Europeans and PROMIS South Asians.** Odds ratios of 300 RVGRSs generated using the top 10 to top 3000 discovery genes (in intervals of 10) on CAD are shown for Europeans in the UK Biobank (blue points) and South Asians in PROMIS (orange points). RVGRS950 and RVGRS680 correspond to the RVGRS that achieved the strongest association in the UK Biobank Europeans and PROMIS South Asians, respectively. Odds ratios are expressed in terms of a 1 SD change in RVGRS and were adjusted for age, age$^2$, and the first 20 principal components of ancestry using a multivariable logistic regression model. Each point represents the odds ratio per 1 SD change in RVGRS on CAD. Shaded regions correspond to 95% confidence interval of the odds ratio. Dashed line represents a line of no effect. OR indicates odds ratio, SD indicates standard deviation, RVGRS950 refers to rare variant genetic risk score from the top 950 discovery genes, and RVGRS680 refers to rare variant genetic risk score from the top 680 discovery genes. Source data are provided as a Source data file.

subjects. We also identified that RVGRS generated using gene-based odds ratios derived from RV-EXCALIBER resulted in higher effect estimates and improved precision for CAD association compared to RVGRS calculated using the gene-based odds ratios from TRAPD in the UK Biobank. In fact, the effect size of RV-EXCALIBER's RVGRS950, which conferred a 1.08-fold (95%

CI, 1.04–1.11; $P = 2.1 \times 10^{-5}$) increased odds of CAD per SD change, represents a >20% increase relative to the predictive power of the RVGRS with the strongest CAD association according to TRAPD-derived odds ratios, which was found to confer a 1.06-fold (95% CI, 1.03–1.10; $P = 4.4 \times 10^{-4}$) increased odds of CAD per SD change (Supplementary Data 5). Indeed, the predictive power offered by RV-EXCALIBER-derived RVGRS on simulated case status was also shown to be appreciably stronger compared to TRAPD-derived RVGRS, especially for simulated scenarios of higher disease prevalence and greater strength of genetic contribution to case status (Supplementary Fig. 12).

**RVGRS acts independently of a CVGRS and clinical risk factors.** RVGRS950 and RVGRS950$^{LDLR-}$ acted independently of both CVGRS and clinical risk factors (ascertained using a Framingham Risk Score (FRS)) while also demonstrating consistent predictive estimates (OR = 1.07; 95% CI, 1.04–1.11; $P = 2.9 \times 10^{-5}$ for RVGRS950 and OR = 1.06; 95% CI, 1.03–1.10; $P = 2.3 \times 10^{-4}$ for RVGRS950$^{LDLR-}$) (Fig. 7). Moreover, we observed a significant interaction between RVGRS950 and FRS, where individuals with higher RVGRS950 had lower clinical risk for CAD ($P_{interaction} = 0.018$). Specifically, individuals in the low, middle, and high tertiles of a FRS had odds ratios of 1.16, 1.10, 1.03 conferred through RVGRS950 for CAD, respectively. The observed interaction also remained significant when using RVGRS950$^{LDLR-}$ (Supplementary Table 7). Interestingly, the effect conferred through a CVGRS did not significantly differ across FRS tertiles, with odds ratios of 1.36, 1.26, and 1.30 for low, middle, and high tertiles of FRS, respectively ($P_{interaction} = 0.74$) (Supplementary Table 8). We next tested whether addition of RVGRS950 could improve patient classification as compared to a base model consisting of only CVGRS and FRS. Incorporation of RVGRS950 was found to significantly improve risk prediction for CAD (net reclassification index = 0.0558; $P = 9.2 \times 10^{-4}$) after adjusting models for age, age$^2$, sex and the first 20 principal components of ancestry. The net reclassification index also remained significant when using RVGRS950$^{LDLR-}$ (Supplementary Table 9). Additionally, we identified that the added variance explained by both RVGRS950 and RVGRS950$^{LDLR-}$ to be 0.1% when compared to a logistic model containing CVGRS, FRS, age, age$^2$, sex, and the first 20 principal components of ancestry alone.

**RVGRS demonstrates increased predictiveness for early-onset CAD.** Due to the substantial heritability of early-onset CAD and the enrichment of mendelian FH mutations in these subjects, we tested the predictive effect of RVGRS950$^{LDLR-}$ across 4 age categories. These included males and females with established onset of CAD at age (i) ≤40 and ≤45, (ii) ≤45 and ≤50, (iii) ≤50 and ≤55, or (iv) ≤55 and ≤60, respectively. RVGRS950$^{LDLR-}$ was found to confer the highest risk among individuals with the earliest onset of CAD, with a 1.32-fold (95% CI, 1.01–1.69; $P = 0.035$) increased odds of disease per 1 SD change in RVGRS (Fig. 9). The median RVGRS950$^{LDLR-}$ in early CAD cases corresponded to the 66th percentile of RVGRS950$^{LDLR-}$ for the control distribution, compared to only the 52nd percentile for general CAD (Supplementary Fig. 14). We also identified higher risk and a trend of increased prevalence of CAD (all cases) across increasing RVGRS950 and RVGRS950$^{LDLR-}$ percentile groupings ($P_{trend} = 1.69 \times 10^{-15}$ and $P_{trend} = 1.26 \times 10^{-14}$, respectively) (Supplementary Fig. 15a, b). Additionally, 1 in 68 (1.5%) of participants in the UK Biobank had a RVGRS950$^{LDLR-}$ score (i.e., ≥98.5th percentile), corresponding to a ≥2-fold increase in odds of early disease. The population distribution of RVGRS950 and RVGRS950$^{LDLR-}$ are provided in Table 1 for the main population strata used from UK Biobank.

| Age strata | N cases / N controls | OR per SD | 95% CI | P-value | |
|---|---|---|---|---|---|
| M ≤ 40 & F ≤ 45 | 50 / 42,007 | 1.32 | 1.01-1.70 | 0.035 | |
| M ≤ 45 & F ≤ 50 | 189 / 42,007 | 1.21 | 1.05-1.39 | $6.3 \times 10^{-3}$ | |
| M ≤ 50 & F ≤ 55 | 536 / 42,007 | 1.14 | 1.05-1.24 | $1.9 \times 10^{-3}$ | |
| M ≤ 55 & F ≤ 60 | 1,128 / 42,007 | 1.09 | 1.02-1.15 | $5.3 \times 10^{-3}$ | |

0.9 1.1 1.3 1.5 1.7
OR per SD (95% CI)

**Fig. 9 Predictive effects of RVGRS950$^{LDLR-}$ among 4 age strata in UK Biobank participants.** Odds ratios of RVGRS950$^{LDLR-}$ are shown for UK Biobank European participants across 4 age strata. Ages correspond to earliest event across all pertinent ICD-10 codes that encompassed our CAD definition. Squares represent odds ratios that are expressed in terms of a 1 SD change in RVGRS. All odds ratios were calculated using a multivariable logistic regression model that was adjusted for sex and the first 20 principal components of ancestry. Horizontal lines indicate the 95% confidence interval of the odds ratio. OR indicates odds ratio, SD indicates standard deviation, CI indicates confidence interval, M indicates male, and F indicates female, and RVGRS950$^{LDLR-}$ refers to rare variant genetic risk score from the top 950 discovery genes, excluding *LDLR*. *LDLR* refers to low-density lipoprotein receptor. Source data are provided as a Source data file.

## Discussion

We herein describe a methodological framework, which leverages summary allele frequencies in gnomAD to identify complex disease risk conferred through the modest effects of rare variant burden. Unlike existing methodologies that assess the contribution of rare variants to complex disease using external control databases[15,16], our workflow systematically calibrates rare variant burden between individual test samples and reference databases. RV-EXCALIBER uses individual and gene-level total allele counts to calibrate individual exome sequences to summary-level allele frequencies in gnomAD while also accounting for LD between rare variants. In doing so, our method addresses the primary sources of bias that confound association signals generated using a technically and phenotypically heterogeneous reference population. By instituting our corrections, we were able to leverage a very large control sample to detect modest rare variant effects which were used to generate a RVGRS for CAD.

Current methodologies that leverage gnomAD as controls have been tailored to demarcate variant and gene-based associations by calibrating functional variant inclusion and counts using benign variants[15,16]. In contrast, RV-EXCALIBER incorporates global and granular adjustments to gnomAD using functional variants to detect rare variant burden associated with complex diseases. Existing methodologies utilizing gnomAD as controls, such as TRAPD and ProxECAT, calibrate test statistics based on benign mutation rate by altering filtering criteria for functional variant inclusion[15,16], but this approach cannot account for technical, population genetic, or gene-based biases inherent to functional variants. While this procedure has been shown to be effective in this respect[15–17], informing variant inclusion through quality metrics such as QD and variant quality score recalibration does not necessarily control for the major sources of bias when using public controls such as gnomAD. Such biases include (1) differences in individual-level total allele count between case subjects and gnomAD, which reflect disparities in sequencing technology and population substructure and (2) bias among genes that demonstrate systematic enrichment or depletion relative to gnomAD. In fact, when conducting a rare variant association test, RV-EXCALIBER demonstrated better calibration of gene-based test statistics relative to TRAPD according to genomic inflation estimates calculated at the median and 95th percentile (Fig. 5). Moreover, TRAPD demonstrated a deflation bias in percent

difference between observed and EAC, especially among genes with high EAC for which bias cannot be attributed to lack of power to detect rare variants associations (Supplementary Fig. 9b). In contrast, RV-EXCALIBER had an unbiased mean percent difference across all EAC (Supplementary Fig. 9a), which further warrants the necessity to institute both individual and gene-level correction (i.e., iCF and gCF) to mitigate residual bias that cannot be accounted for when assuming that this bias is the same across all genes, which is the case for TRAPD.

We show through simulations that individual-level differences in rare variant mutation burden can be impacted by variation in population substructure (i.e., PSF) and technological biases (i.e., SFP and SFN). Detection of rare variants through exome sequencing can vary substantially based on methods of library preparation, exome enrichment strategy, read alignment[18]. To evaluate the effect of population factors, we used three consensus sequences from GIAB and compared to closely matched ethnic groups in gnomAD. For each comparison, the iCF deviated most from expectation in rare variant bins, especially among European populations (Fig. 3). This is largely because rare variants demonstrate tremendous geographical disparity as they have either arisen recently or are older variants that are functionally deleterious and so kept at low population frequency by purifying selection[11,19–21]. As such, individual rare variants tend to cluster in smaller population groups and are not geographically ubiquitous like common variants. This is especially true in Europe, as principal component analysis has revealed consistent and distinct clustering of sub-populations[13,22]. Indeed, we show that the total allele count in European consensus sequences (i.e., North-western European or Ashkenazi) deviate substantially from the Non-Finnish Europeans in gnomAD and are more closely matched across common variants (Fig. 3a and c). Surprisingly, differences in iCF between cases and controls were observed even within single studies (Supplementary Fig. 6). This can likely be attributed to batch effect sequencing, which provides further rationale to incorporate correction when comparing exomes to reference datasets, especially in case-control designs where samples are potentially sequenced in batches based on affection status. Although the iCF accounts for global deviation in sequencing and population bias, we instituted the gCF to specifically address the bias in total allele count at the gene-level, which cannot otherwise be captured on an exome-wide scale. Specifically, by tracking the cumulative exome-wide total allele count, we demonstrate that the gCF mitigates bias in genes harboring residual inflation or deflation that could not be addressed using iCF alone for global adjustment. Since genes were ranked according to both *P*-value and EAC in an independent set of healthy controls, we demarcate genes that are reproducibly "enriched", "well-calibrated", and "depleted" for rare, pathogenic alleles in relation to gnomAD non-Finnish Europeans (Fig. 4b). Overall, our results validate presence of systematic gene biases that are specific to a given cohort and effectively illustrate the extent of adjustment offered by our corrections. It is important to note that both iCF and gCF are implemented on our base method that adjusts for LD among rare variants (Supplementary Fig. 11), since LD cannot be estimated by summary-level data in gnomAD. Alternative methods do not adjust for LD when using gnomAD as controls[15,16]. A noteworthy limitation to RV-EXCALIBER is a tendency for deflation of test statistics at the significant tail (Supplementary Figs. 10 and 11), which was observed using healthy control data. Although we observed less deflation at the significant tail for cases (Fig. 5), we recommend that users thorough query top association signals to identify if any genes are strong biological candidates for the trait being assessed. Indeed, while *LDLR* demonstrated both biological and statistical significance, a gene-set enrichment analysis revealed that top discovery genes were nominally

**Table 1 Summary of the RVGRS950 and RVGRS950$^{LDLR-}$ distributions in the UK Biobank validation population.**

| | UK Biobank CAD cases | UK Biobank CAD-free controls |
|---|---|---|
| *Overall population* | | |
| n | 3843 | 42,007 |
| RVGRS950 mean (SD) | 0.039 (0.69) | −0.009 (0.67) |
| RVGRS950$^{LDLR-}$ mean (SD) | 0.033 (0.68) | −0.009 (0.67) |
| *FRS* | | |
| FRS tertile 1 | | |
| n | 519 | 14,765 |
| RVGRS950 mean (SD) | 0.086 (0.71) | −0.013 (0.67) |
| RVGRS950$^{LDLR-}$ mean (SD) | 0.075 (0.70) | −0.013 (0.67) |
| FRS tertile 2 | | |
| n | 1193 | 14,090 |
| RVGRS950 mean (SD) | 0.046 (0.70) | −0.013 (0.67) |
| RVGRS950$^{LDLR-}$ mean (SD) | 0.039 (0.70) | −0.013 (0.67) |
| FRS tertile 3 | | |
| n | 2131 | 13,152 |
| RVGRS950 mean (SD) | 0.025 (0.67) | 0.0014 (0.68) |
| RVGRS950$^{LDLR-}$ mean (SD) | 0.020 (0.66) | 0.00083 (0.67) |
| *Age strata* | | |
| Males ≤40 and Females ≤45 | | |
| n | 50 | 42,007 |
| RVGRS950$^{LDLR-}$ mean (SD) | 0.20 (0.76) | −0.0088 (0.67) |
| Males ≤45 and Females ≤50 | | |
| n | 189 | 42,007 |
| RVGRS950$^{LDLR-}$ mean (SD) | 0.13 (0.70) | −0.0088 (0.67) |
| Males ≤50 and Females ≤55 | | |
| n | 536 | 42,007 |
| RVGRS950$^{LDLR-}$ mean (SD) | 0.082 (0.70) | −0.0088 (0.67) |
| Males ≤55 and Females ≤60 | | |
| n | 1128 | 42,007 |
| RVGRS950$^{LDLR-}$ mean (SD) | 0.048 (0.69) | −0.0088 (0.67) |
| *≥2-fold risk for early CAD*[a] | | |
| n | 670 | 45,180 |
| RVGRS950$^{LDLR-}$ mean (SD) | 2.00 (0.35) | −0.035 (0.63) |

[a]Cases and controls refer to individuals with ≥2-fold risk and <2-fold risk of early CAD, respectively.

enriched for known biological processes involved in CAD onset, including response to stress, biosynthesis/response to reactive oxygen species, and nitric oxide biosynthesis (Supplementary Table 6). As such, assessing top genes individually or as a gene-set may provide biological insights, regardless of whether these genes show mild deflation at the significant tail of the distribution.

We next sought translate the gene-based associations generated by RV-EXCALIBER to construct a RVGRS. Common variant gene scoring approaches have limited trans-ancestral predictive value and clinical implications remain highest when discovery and validation sets are ethnically matched, which is largely the case only for European populations[19,20,23]. Moreover, empirical and simulation models have demonstrated that directional selection across ancestries can severely impair predictive accuracy in non-European validation populations, even after accounting for principal components and proportion of causal variants[19,20]. The large disparity in LD structure between ethnic groups

markedly impair the predictive value of CVGRS since causal variants remain largely unknown and thus tag variants are used. In contrast, rare variants are more likely to be functional and thus execute the same biological function regardless of ancestral origin. Moreover, since the RVGRS assesses the burden of rare variants at the gene-level, it does not require identical variants be present in an independent population. Instead, the RVGRS leverages the ubiquitous function of genes across ancestries to confer consistent effects, despite the geographical specificity of individual rare variants. Indeed, we show that the RVGRS demonstrates consistent predictive estimates for Europeans in UK Biobank exomes and South Asian exomes in PROMIS (Fig. 8).

In order to concretely establish the RVGRS as an independent mediator of disease, we verified that the predictive estimate elicited through RVGRS remained consistent even after adjusting for CVGRS as derived using effect estimates from the CARDIoGRAMplusC4D consortia and clinical risk factors as calculated by a FRS[22,24]. In fact, adding RVGRS to a reference model consisting of CVGRS and clinical risk factors significantly reclassified 5.6% of CAD events, which represents a marked improvement given the relatively small size of our discovery population (Supplementary Table 9). However, the RVGRS only explained 0.1% of variance in CAD affection status, which is lower compared to traditional CVGRS for CAD[25]. This reason for this large difference in explained variance are two-fold. First is the substantially larger discovery sample size used to generate weights for CVGRS, which will necessarily increase model calibration when assessing risk an independent validation population. Second is the difference in exposure frequency between the RVGRS and CVGRS, wherein the component variants used to inform the RVGRS are necessarily observed less commonly in any given population. As such, the RVGRS (as it stands) will be less powered to detect population-level variances relative to established CVGRS for CAD[25]. Despite this, we maintain that the RVGRS has potential usefulness to identify high-risk individuals, in contrast to explaining a high degree of population-level variance.

When evaluating the potential clinical efficacy of the RVGRS, we observed that it demonstrated an increase in predictive effect for individuals at low clinical risk as compared to those with a high clinical burden for developing CAD (Supplementary Table 8). We acknowledge this observation to be hypothesis-generating, wherein individuals with low clinical risk may have an enrichment of rare risk alleles of modest effect (i.e., RVGRS) compared to those with moderate or high clinical risk. Future work on gene x environment interactions that focusses on rare variants are required to evaluate the clinical utility of RVGRS, particularly among individuals with low versus high burden of environmental risk. Interestingly, no such effect gradient was observed for the CVGRS, which mitigates its ability to discern residual genetic risk for CAD among individuals that may present with limited clinical risk factors. Similar to CVGRS, we found that the risk conferred through RVGRS was strongest among very young individuals (i.e., males ≤40 and females ≤45) and that the risk markedly decreased upon inclusion of older individuals (Supplementary Fig. 14), which suggests a higher burden of rare variant associations in early-onset cases. This is consistent with earlier evidence which shows that early-onset cases have a higher genetic burden of disease due to limited exposure to traditional environmental risk factors[26–30].

It is important to note that the risk estimates for high polygenic risk derived from RVGRS and CVGRS cannot be readily compared due both to the limited complex trait heritably explained by rare variants (relative to common variants) and to the large discrepancy in discovery sample size used to compute effect estimates to weight alleles in validation samples. In our work, we included 5,910 cases to derive beta estimates while the betas used

for CVGRS in CAD/MI are obtained from the CARDIo-GRAMplusC4D meta-analysis, which has a case population that is 10.3× larger. Nevertheless, our work bridges the gap between rare and common variants under a unified polygenic framework. Despite the marked difference in case sample size, we show that a RVGRS confers high risk (i.e., ≥2-fold) for early CAD in 1.5% of the population and remains independent of Mendelian effects, clinical risk factors, and CVGRS. Notwithstanding its independent effect on CAD and ability to demarcate individual-level risk, we stress that a RVGRS should be used as an adjunct genetic variable that is used in conjunction with Mendelian variants and CVGRS to enhance the overall clinical utility of genetic factors.

We herein present a method (RV-EXCALIBER) for calibrating rare, pathogenic alleles between case exomes and gnomAD by mitigating major biases at the individual and gene-level while also correcting for putative LD between rare variants. RV-EXCALIBER demonstrates better calibration than existing methodologies that leverage gnomAD as controls and was empirically applied to generate a first RVGRS for CAD. The RVGRS conferred disease risk independent of Mendelian effects, clinical risk factors, and CVGRS while also correctly reclassifying ~6% of CAD events, which suggests that addition of RVGRS to CVGRS and Mendelian mutations may have potential clinical utility in terms of case discrimination, but further validation is necessary before establishing a full case for clinical implementation. While the RVGRS explained only 0.1% of population variance in CAD affection status after accounting for known genetic and environmental factors, it still demarcated high individual-level CAD risk (i.e. 1.5% of individuals with ≥2-fold risk). Lastly, the RVGRS demonstrates higher predicative effect among individuals with low clinical risk burden and those with early-onset CAD. These findings suggest an enriched genetic contribution to CAD in the absence of environmental risk factors and further validates the increased role of inheritance in modulating early disease risk[26,29,30]. Our results also make a strong case for large sequencing studies of common, complex traits and diseases. Additionally, leveraging publicly available data as controls in an unbiased manner may permit additional resources to be allocated to recruiting extreme disease cases, which might harbor a greater burden of rare causal variants. Lastly, we also note that RV-EXCALIBER could be extended to whole-genome sequencing, in order to better delineate the role of rare non-coding variants on disease risk.

## Methods

**Single gene simulations**. We simulated an additive rare variant association model for a single gene with differing probabilities of mutation based on 3 confounding parameters: (1) sequencing false negative (SFN) rates, (2) sequencing false positive (SFP) rates, and (3) population-specific factors (PSF). SFN is a real number between 0 and 1 that represents the proportion of true variant calls missed by sequencing. SFP is a real number ≥0 and corresponds to the ratio of called false variants to the number of (true) variants in the reference population. The PSF parameter is also a real number ≥0 and is defined as the ratio of (true) variants in a gene relative to a reference sample. PSF represents the effect of population substructure, where PSF=1 when the test and reference populations are perfectly matched. The probability of mutation $P$ for the gene is given by Eq. (1):

$$P = \text{CMAF}_{\text{Gene}}\text{PSF}(1 - \text{SFN}) + (\text{CMAF}_{\text{Gene}}\text{SFP}) \quad (1)$$

where $\text{CMAF}_{\text{Gene}}$ is the baseline cumulative minor allele frequency (CMAF) (defined as the aggregate sum of the MAF) of the simulated gene in the absence of any genetic effect, sequencing effects (i.e., SFN and SFP), or population effects (i.e., PSF). Please note use of $\text{CMAF}_{\text{Gene}}$ implicitly assumes variants are rare and LD negligible. When a genetic effect exists, Eq. (1) can be written as:

$$P = \text{CMAF}_{\text{Gene}}\text{PSF}\,\text{OR}(1 - \text{SFN}) + (\text{CMAF}_{\text{Gene}}\text{SFP}) \quad (2)$$

where $P$ represents the probability of mutation given an individual is a case and is therefore dependent upon the OR, which is the true odds ratio, or genetic effect of the association. It is important to note that the OR represents the ratio of the frequency of rare alleles in a case population to a control population using the

approximation of rare exposures:

$$\text{OR} = \frac{\text{Case}_{\text{Exposed}}/\text{Control}_{\text{Exposed}}}{\text{Case}_{\text{Nonexposed}}/\text{Control}_{\text{Nonexposed}}}$$

where

$$\frac{\text{Case}_{\text{Nonexposed}}}{\text{Control}_{\text{Nonexposed}}} \sim 1$$

for rare exposures (i.e., the CMAF attributed to the aggregate rare allele frequency within a gene). The equation for OR thus simplifies to

$$\text{OR} = \frac{\text{Case}_{\text{Exposed}}}{\text{Control}_{\text{Exposed}}}$$

which we also empirically confirmed.

We then derived a correction factor (CF) to adjust for sequencing false positives, negatives, and population substructure effects. CF is defined as following, which is a direct simplification of Eq. (2) under the assumption of no genetic effect (OR = 1):

$$\text{CF} = \text{PSF}(1 - \text{SFN}) + \text{SFP} = \frac{P}{\text{CMAF}_{\text{Gene}}} \quad (3)$$

CF can be estimated by calculating the ratio of observed variants to expected variants in the reference population (i.e., $\text{CMAF}_{\text{Gene}}$) over all protein-coding genes (or at least a large number of genes). Since CF is empirically estimated across all protein-coding genes, it is assumed that the effect of truly associated genes on genome-wide estimates is negligible. Estimated CF ($\widehat{\text{CF}}$) can then be used to calculate an adjusted $P^*$ for the gene:

$$P^* = \widehat{\text{CF}}\,\text{CMAF}_{\text{Gene}} \quad (4)$$

We then sought to determine whether use of $P^*$ (as opposed to $P = \text{CMAF}_{\text{Gene}}$) would improve association testing in the presence of confounding (i.e., SFN > 0, SFP > 0 or PSF < > 1). To test whether use of $P^*$ provides unbiased estimates of OR, we varied the true (unobserved) OR from 1 (null effect) to 2 (strong effect) at intervals of 0.1. Each OR was simulated 100,000 times in 1000 individuals and the mean (SD) estimated OR calculated. In all simulations, SPN and SPF were set to 0.3 and PSF was set at 0.8 to illustrate conditions of strong (but realistic) confounding for each source of bias. To ascertain type I and II errors we performed a second set of simulations, fixing the true (unobserved) OR to 1.3 and varying SFN and SFP from 0 to 1 and PSF from 0 to 2. Each condition was simulated 100,000 times in 1,000 participants. $P$-values were calculated using a Z-test analogous to Eq. (10), where $\bar{D}_g$ represents the difference in between the simulated observed allele counts (OAC) and either $P^*$ or $P$.

**Sequence coverage harmonization**. Differential sequence coverage driven by non-biological factors between any two cohorts can render genetic association signals prone to false positives or negatives. As such, restricting analyses to sites that are sensitive to variant detection in both gnomAD and individual-level sequences (i.e., GIAB and MIGen) will mitigate detection of spurious associations. Per variant coverage statistics for gnomAD release 2.0.1 were downloaded from Google Cloud public datasets (https://gnomad.broadinstitute.org/downloads). First, variant sites were intersected against an interval file corresponding to refGene coding regions across the genome as defined by the Reference Sequence (RefSeq) database (the interval file was retrieved from the University of Southern California (UCSC) Table Browser: https://genome.ucsc.edu/cgi-bin/hgTables). Second, the average proportion of individuals with ≥20X coverage was determined for each coding interval. Intervals in which ≥90% of individuals had ≥20X coverage were deemed as "high-coverage coding (HCC)" intervals. Lastly, the resulting HCC intervals were intersected against high-confidence regions (using the NIST high-confidence regions for GIAB samples and cohort-specific exome enrichment intervals for MIGen samples; see Supplementary Table 1). The resultant final intersected regions would therefore be characterized by HCC regions that are sensitive to detecting variants in gnomAD whilst still capturing high quality and enriched regions in GIAB and MIGen samples, respectively.

**Benchmarking correction factors for GIAB consensus sequences**. Correction factors for each individual were calculated based on the ratio of total allele counts between a GIAB comparator sample and the gnomAD dataset. Specifically, allele counts across 18,410 autosomal protein-coding genes curated in the refGene coding regions file (see Supplementary Note 7, section B) were aggregated to generate observed total allele count in the consensus GIAB sample and then compared to the aggregate expected total allele count obtained from gnomAD frequencies across the same set of genes (Eq. (5)

$$\widehat{\text{iCF}}_i = \frac{\sum_{g=1}^{M}\text{OAC}_{i,g}}{\sum_{g=1}^{M}\text{EAC}_{i,g}} \quad (5)$$

where $\widehat{\text{iCF}}_i$ represents the estimated individual correction factor for individual $i$ which is equal to the ratio between the sum of the observed allele counts (OAC) across all $M$ genes in the GIAB comparator sequence to the sum of the expected

allele counts in gnomAD (EAC) (i.e., 2CMAF) for the same $M$ genes. For simplicity, $\widehat{iCF}_i$ is hereafter referred to as iCF. Total allele counts for both gnomAD and GIAB were restricted to the intersection of variant sites between gnomAD HCC and the NIST high-confidence regions for a given GIAB sample. EAC were generated using ethnic-specific allele frequencies that best matched the ethnicity of the comparator sample in order to limit population stratification bias. For example, the EAC for comparison with the North-western European GIAB sample (NA12878) was generated using the Non-Finnish European AAFs in gnomAD (see Supplementary Table 2 for all GIAB to gnomAD ethnicity matches). A 95% confidence interval was calculated for the iCF in each allele frequency bin using cumulative Bernoulli variance where the probability of success was denoted by the gene CMAF.

**Determining iCF heterogeneity for rare variants in GIAB consensus sequences.** Since a single value of iCF is to be used to equilibrate rare mutation burden across the exome, it is imperative to identify whether iCF differs significantly across genes stratified by different levels of constraint (i.e., a gene's tolerance towards harboring deleterious mutations). To test this, constraint scores[8] for all genes were downloaded from the Exome Aggregation Consortium (ExAC) release 0.3.1 ftp repository (ftp://ftp.broadinstitute.org/pub/ExAC_release/release0.3.1/functional_gene_constraint/fordist_cleaned_exac_r03/march16_z_pli_rec_null_data.txt). Genes were stratified into deciles based on their $Z$-score for missense variant constraint. Heterogeneity of the iCF between deciles was assessed through a fixed effect meta-analysis using the iCF and standard deviation (determined from the 95% CI described in Supplementary Note 7, section C) of the iCF for each decile. $P$-value for heterogeneity was computed using rMeta R package[31].

**Calculation of iCF in MIGen exome sequences.** The iCF was calculated for each MIGen participant as per Eq. (5). Variant sites contributing to total allele count were selected from the intersection of gnomAD HCC regions and the exome enrichment intervals for each MIGen cohort (Supplementary Table 1). Upon generating an iCF for every MIGen participant (as per Eq. (5), an iCF-adjusted $P$ (i.e., $P^*$) was calculated per sample per gene according to:

$$P^*_{i,g} = iCF_i EAC_g \tag{6}$$

As for the GIAB analysis, the EAC for each gene was based on the gnomAD population that was most closely related to the ethnicity of the MIGen cohort (Supplementary Table 1). For example, the EAC for all European cohorts was generated using the NFE AAFs whereas the EAC for the South Asian cohort used the South Asian AAFs in gnomAD.

**Calculation of the gCF in MIGen exome sequences.** It was observed that some genes are consistently enriched or depleted in rare mutations in healthy controls from multiple cohorts as compared to gnomAD, pointing to gene-specific sequencing false positives/negatives or gene-specific population structure effects. Hence, a gCF was instituted to adjust for biases in gene mutation burden not captured by the iCF. To determine the gCF, genes were binned according to their association signals in an independent calibration cohort that was (1) sequenced using the same exome enrichment chemistry and technological platform as the test cohort and (2) absent of samples with phenotypes that were either identical to, or risk factors for the phenotype characterizing the test cohort (e.g., a cohort with samples positive for MI or type 2 diabetes mellitus could not be used to calibrate a test cohort of MI cases). The rationale for these criteria are to bin genes with similar biases in mutation burden (criteria 1), but not based on mutation burden due to presence of disease-conferring mutations (criteria 2). To fulfill these two criteria, gene bins were established using first, quintiles of EAC (in ascending order) and second, deciles of $P$-values (from most to least significant) within a ranking cohort consisting of 6082 disease-free controls across 8 MIGen populations (Supplementary Data 1). As $P$-values are one-sided, the ranking will order genes from the one with the greatest evidence for enrichment in rare pathogenic mutations in MIGen healthy controls versus gnomAD to the one with the greatest evidence for depletion in rare, pathogenic alleles in MIGen healthy controls versus gnomAD. Additionally, by ordering by $P$-value within quintiles of EAC we account for genes that may exhibit significant association signals by virtue of limited power to detect mutations. A gCF was then calculated according to Eq. (7):

$$\widehat{gCF}_b = \frac{\sum_{i=1}^{N}\sum_{g=1}^{M} OAC_{i,g}\mathbf{1}_b(g)}{\sum_{i=1}^{N}\sum_{g=1}^{M} EAC_{i,g}\mathbf{1}_b(g)} \tag{7}$$

where $\widehat{gCF}_b$ is the estimated gene correction factor value across $N$ individuals and $M$ genes in bin $b$ and $\mathbf{1}_b(g)$ is the indicator function with value 1 if gene $g$ is part of bin $b$ and 0 otherwise. $\widehat{gCF}_b$ is hereafter referred to as gCF for simplicity. A bin size of ~232 (50 bins across 11,616 genes with a $OAC_g$ and a $EAC_g$ of at least 1 assessed in the ranking cohort) ensures that the proportion of disease-causing genes is expected to be small in each bin. Selecting smaller bin sizes (i.e., ≤ 50) ultimately nullifies the per gene association signals, which precludes inclusion of a gCF. In other words, small bin sizes will tend to bias results towards the null of no association. In contrast, extensively large bin sizes will bin genes together that do not necessarily share a similar bias in total allele count, which will dilute the effect of

the gCF and render it ineffective. An iCF and gCF adjusted $P$ (i.e. $P^{**}$) was then calculated for each gene and individual according to:

$$P^{**}_{i,g} = gCF_{b:g\in b} iCF_i EAC_g \tag{8}$$

**Rare variant association.** To test associations of individual genes with MI, a burden test was used to evaluate the difference in aggregate counts for rare, pathogenic alleles (as defined in Supplementary Note 7, section D) between MIGen cases ($n = 5910$) and the iCF and gCF-adjusted EAC. To account for potential LD between rare variants as well as biases in mutation burden between test samples and gnomAD, we introduce a novel procedure based on individual differences between observed and expected mutation counts. Specifically, for each gene and individual, we define the delta allele count ($D$) as:

$$D_{i,g} = OAC_{i,g} - P^{**}_{i,g} \tag{9}$$

For each gene, we then test whether mean $D_{i,g}$ (i.e., $\bar{D}_g$) is equal or different than zero using a Z test since $\bar{D}_g$ is expected to be equal to zero under the null hypothesis of no association:

$$Z_g = \frac{\bar{D}_g}{\sigma(D_g)/\sqrt{n}} \tag{10}$$

This procedure has several advantages. First, it adjusts for biases in mutation burden between the test samples and gnomAD. Second, it leverages the fact LD between variants will increase variance (i.e., $\sigma^2(D_g)$ but not expected $D_g$. As $\sigma(D_g)$ is empirically derived, a priori knowledge of the LD structure is not necessary.

Genomic inflation factors calculated at the median of gene-based test statistics ($\lambda_{med}$) were generated using the estlambda function within the GenABEL R package[32].

**Evaluating enrichment in discovery genes found to be genome-wide significant loci in CARDIoGRAMplusC4D and gene-set enrichment analysis.** Genes from the discovery rare variant association in MIGen were cross referenced against loci harboring a genome-wide significant variant ($P < 5 \times 10^{-8}$) in the CARDIoGRAMplusC4D consortium[22]. A total of 36 CAD genes were identified among the 7146 discovery genes, including the low-density lipoprotein receptor ($LDLR$). Due to the disproportionate association signal for $LDLR$, a combined test statistic was generated from the remaining 35 CAD genes using Fisher's combined probability test. To test the null hypothesis that the combined test statistic for the CAD gene set was not significantly different from that of random gene sets, we generated a null distribution of combined test statistics for 100,000 randomly selected gene sets of identical size (also excluding $LDLR$) and identified the probability of observing a test statistic greater than or equivalent to that of the CAD gene set. A separate gene-set enrichment was evaluated using the GOrilla tool (http://cbl-gorilla.cs.technion.ac.il/) on all nominally associated discovery genes ($P < 0.05$) relative to all discovery genes with a OAC and EAC ≥ 1 and ≥10, respectively (7146 total). A total of 12,958 Gene Ontology (GO) biological processes were evaluated for enrichment. Fold enrichment was calculated according to the ratio of the proportion of target genes (i.e., discovery genes with $P < 0.05$) to the proportion of all genes (i.e., all discovery genes) in a given GO biological process. Enrichment $P$-values were calculated from a minimum hypergeometric test statistic[33].

**Rare variant association using TRAPD.** The distribution of gene-base test statistics generated using RV-EXCALIBER were compared to those generated using TRAPD. Specifically, healthy control subjects from the ATVB, OHS, BioImage, and Munich-MI cohorts ($n = 3352$) were used as the "case" dataset and tested against non-Finnish Europeans in gnomAD, which was used as the control population. For TRAPD, quality-by-depth (QD) values were generated in cases by computing the ratio of the variant-level quality to the sum of the depth of coverage for all non-homozygous reference genotypes across a single variant. QD values in gnomAD were extracted directly from the INFO field present in the downloaded variant call file (Supplementary Note 1). To remain consistent, variant frequency and pathogenicity filtering along with sequencing coverage harmonization was completed exactly as performed for RV-EXCALIBER (Supplementary Note 7, sections D–E and as described above). The distribution of QD values was determined for all rare synonymous SNVs in cases to obtain variant sets corresponding to the top 50th to 99th percentiles of QD scores. For gnomAD we kept only the rare synonymous SNVs corresponding to the top 70th, 75th, 80th, 85th, 90th, and 95th percentile of QD scores. Per gene $P$-values were generated for each QD percentile set from $2 \times 2$ contingency tables using a Fisher's Exact test followed by calculation of a genomic inflation factor at the 95th percentile of gene-based test statistics ($\lambda_{95}$). $\lambda_{95}$ was calculated based on the ratio of observed $-\log_{10}(P)$ to expected $-\log_{10}(P)$ at the 95th percentile of all genes, where $P$ refers to a gene-based $P$-value[15]. The percentile set with the best calibrated $\lambda_{95}$ was used to inform inclusion of nonsynonymous SNVs for burden testing. Inclusion of INDELs were informed using the same procedure, but were calibrated using rare nonsynonymous SNVs instead of synonymous SNVs.

For RV-EXCALIBER, gene-based test statistics were calculated as previously described where healthy controls from the PROCARDIS, U. Lubeck, REGICOR, and MDC cohorts ($n = 2730$) were used as the ranking set for calculation of gCF.

**Calculation of the RVGRS.** Gene-based odds ratios ($OR_g$) and corresponding regression coefficients ($\beta_g$) were generated from rare variant association analysis between the discovery MIGen samples (Supplementary Table 1) and gnomAD according to:

$$OR_g = \frac{\sum_{i=1}^{N} OAC_{i,g}}{\sum_{i=1}^{N} P_{i,g}^{**}} \quad (11)$$

and

$$\beta_g = \ln\left(OR_g\right) \quad (12)$$

A weighted rare variants genetic risk score (RVGRS) was then derived for individual $i$ by summing the product of $D_{i,g}$ and $\beta_g$ across all genes (included in the gene score):

$$RVGRS_i = \sum_{g=1}^{M} \beta_g D_{i,g} \quad (13)$$

The RV-EXCALIBER model was used to evaluate the association of $RVGRS_i$ with case status in validation cases that were independent from the derivation (discovery) sample. $RVGRS_i$ is hereafter referred to as RVGRS for simplicity. For the UK Biobank, we selected the entire population ($n = 45,850$) as the ranking cohort (due to it being an unselected prospective cohort study) in order to calculate the $D_{i,g}$ term to be weighted by the discovery effect size estimates. For PROMIS, we used the same ranking cohort as was used for the discovery association test.

Inclusion of genes from the discovery gene associations to generate the RVGRS in the validation cohorts was done in a cumulative stepwise manner. Specifically, the RVGRS were built using the top 10 to 3000 most significant genes by including 10 additional genes per step, for a total of 300 sets of RVGRS in the validation cohort. Logistic regression was performed to evaluate the association between RVGRS and CAD in the UK Biobank or MI in PROMIS while adjusting for age, $age^2$, sex, and the first 20 genetic principal components. The distribution of raw RVGRS were scaled to a mean of 0 and a SD of 1 prior to association testing. The OR for an increase of 1 SD in the score was obtained by calculating the natural exponential function of the regression coefficient. The 95% confidence intervals were calculated according to the standard error of the regression coefficient. Significance for all interaction calculations was conducted by regressing case affection status with an interaction term between the scaled RVGRS and the parameter of interest. We defined substantial risk conferred as having a RVGRS corresponding to a $\geq$ 2-fold increased odds of disease. Odds of CAD was determined based on 100 percentile groupings of RVGRS (RVGRS950 or $RVGRS950^{LDLR-}$) in the UK Biobank and was determined for the high percentile group relative to the remainder of the population (e.g., odds of CAD for individuals with $\geq$90th percentile of RVGRS relative to the remaining 10% of the population). Odds of CAD was calculated as stated above. Significance for change in CAD prevalence across increasing percentile of RVGRS was evaluated using a two-sided Cochrane-Armitage test for trend with the DescTools R package[34]. Heterogeneity between the effect conferred by RV-EXCALIBER and TRAPD-derived RVGRS on CAD in the UK Biobank was ascertained with a fixed-effect meta-analysis. The meta-analysis used the $\beta$ estimates and standard error of the $\beta$ estimate for the RVGRS (constructed using both RV-EXCALIBER and TRAPD) on CAD for a given number of discovery genes. $\beta$ estimates and the standard error of the $\beta$ estimate was ascertained from a multivariable logistic regression model that was adjusted for age, $age^2$, sex and the first 20 principal components of ancestry. $P$-values for heterogeneity were computed using the rMeta[31] R package.

**RV-EXCALIBER benchmark simulations.** Randomly sampled discovery and validation populations were obtained UK Biobank WES dataset (described in Supplementary Note 6, section A) to derive simulated gene-based effect sizes and construct a simulated RVGRS, respectively. Case status in the discovery and validation populations were ascertained according to a probability-based sampling approach (without replacement) where the probability of being a "case" ($P_i$) was informed according to Eq. (14):

$$P_i = \frac{e^{\beta_0 + \sum_{gre=1}^{M} \beta_{gre} D_{i,gre}}}{1 + e^{\beta_0 + \sum_{gre=1}^{M} \beta_{gre} D_{i,gre}}} \quad (14)$$

where $\beta_0$ represents the log-odds of the simulated case prevalence to be sampled from the UK Biobank, $\beta_{gre}$ represents a fixed regression coefficient applied to a random set of 100 "genes of real effect (gre)", and $D_{i,gre}$ is the per-individual delta allele count (defined in Eq. (9)) for each of the gre. We set $\beta_0$ to correspond to a case prevalence of either 10% or 20% and assigned $\beta_{gre}$ to correspond to an odds ratio of either 1.4, 1.6, 1.8, or 2.0. A total of 10 randomly sampled gre sets (i.e., 10 sets of 100 gre) were used to generate 10 case populations per $\beta_0$ per $\beta_{gre}$ (i.e., 80 simulated case populations). Each simulated case population was thereafter evenly split into either the discovery or validation set, where the simulated case in the discovery set underwent gene-based rare variant association testing using RV-EXCALIBER and TRAPD. The remaining participants in the UK Biobank that were not selected as

cases using probability-based sampling for a given simulation were used as the controls in the validation population. RVGRS were calculated based on the top 10 to top 1500 discovery genes (at increments of 10 g for simulated case prevalence of 10% (at increments of 10 genes from the top 10 to top 100 genes and increments 100 genes from top 100 to top 1500) or the top 10 to top 3000 discovery genes for a simulated case prevalence of 20% (at increments of 10 genes from the top 10 to top 100 genes and increments 100 genes from top 100 to top 3,000 genes) using TRAPD and RV-EXCALIBER-derived discovery gene-based effect sizes, which would weight the $OAC_{i,g}$ and $D_{i,g}$, respectively (see Eqs. (5) and (9), respectively). RVGRS were scaled to mean 0 and SD 1 and underwent univariable logistic regression to predict simulated cases status. A mean regression coefficient was thereafter obtained for a given discovery gene number across the 10 simulations performed at a given $\beta_0$ and $\beta_{gre}$. The 95% confidence intervals for the mean regression coefficient were obtained using bootstrapping. Odds ratios and their 95% confidence intervals were calculated by taking the natural exponent of the mean regression coefficient and its corresponding confidence interval.

**Calculating the FRS.** A 10-year estimation of cardiovascular disease risk was determined using a FRS[24]. The FRS incorporated age, sex, high-density lipoprotein cholesterol, total cholesterol, systolic blood pressure, antihypertensive medication, and smoking status to generate a composite score on a per-individual basis. Missing clinical data was classified as "missing completely at random" and was imputed using multivariate imputation by chained equations[35]. We opted to use recommended values of 5 imputed datasets over 50 iterations using predictive mean matching. To test the hypothesis that individuals with low burden of clinical risk factors had higher genetic risk of disease, we regressed case affection status with a multiplicative interaction term consisting of standardized RVGRS and tertile of FRS while still using age, $age^2$, sex, and the first 20 principle components of ancestry as covariates.

**Calculating a CVGRS for UK Biobank participants.** CVGRS was calculated according to the weighted sum of common risk alleles (MAF > 0.01), where the genotype of a risk allele was weighted by its corresponding $\beta$ coefficient in the CARDIoGRAMplusC4D consortium with 1000 genomes imputation[22]. LD adjustment was incorporated as published in the GraBLD study, where weights were corrected for each single nucleotide polymorphism (SNP) such that all SNPs can be included in the gene score[36]. The default value of 300 SNPs upstream and downstream from the target SNP is considered.

PLINK's --score function[37] was the main method of generating the CVGRS, which accordingly applies a linear scoring system to the inputted genotype matrix. SNPs were matched between the UK Biobank and GWAS data from the CARDIoGRAMplusC4D consortium with 1000 genomes imputation[22], after which all duplicate and triallelic SNPs were removed. All scores were corrected for threshold and corrected for LD. Initially, various $P$-value thresholds and LD windows values were tested to seek for an optimal value. A default value of 0.1 for threshold and LD window size of 300 was utilized for all initial scores. All scores are then standardized to a mean of 0 and SD of 1. After LD pruning, weights from a total of 838,897 SNPs were used to construct the CVGRS.

Interaction between standardized CVGRS and FRS tertile was conducted as described above.

**Evaluating NRI index and variance explained using RVGRS.** We evaluated the NRI index when incorporating RVGRS into a base model consisting of CVGRS and FRS. Specifically, we first generated a logistic regression model where case affection status was regressed against CVGRS, FRS, age, $age^2$, sex, and the first 20 principle components. A second model was then generated by regressing affection status against RVGRS and the aforementioned independent variables. Both models were thereafter imported into the Hmisc R package[38] to ascertain the NRI index. Variance of affection status explained by the RVGRS was evaluated using Nagelkerke's pseudo $R^2$, where the difference in $R^2$ between the full model (i.e., RVGRS and covariates) and the reduced model (i.e., covariates alone) was used as the variance attributable to the RVGRS. Model covariates included CVGRS, FRS, age, $age^2$, sex, and the first 20 principal components of ancestry. The Nagelkerke's pseudo $R^2$ metric was calculated using the rms R package[39].

**Stratifying risk conferred by the RVGRS based on age of CAD onset.** Age of onset for each ICD-10 phenotype was defined according to date of first in-patient diagnosis (UK Biobank data field 41280). Age of CAD onset was ascertained according to the first occurring event in our composite CAD definition (Supplementary Note 6, section A). Age of onset was stratified in 4 categories: (i) males $\leq$40 and females $\leq$45, (ii) males $\leq$40 and females $\leq$45, (iii) males $\leq$45 and females $\leq$50, and (iv) males $\leq$55 and females $\leq$60. Enrichment of RVGRS in cases from the 4 age strata were evaluated against the UK Biobank control population. Association with CAD was evaluated using sex and the first 20 principal components as covariates. Age and $age^2$ were not used as covariates since the age distribution between early-onset cases and controls is necessarily significant.

**Reporting summary**. Further information on research design is available in the Nature Research Reporting Summary linked to this article.

## Data availability

Individual-level genetic and phenotypic data for the 9 cohorts from MIGen were obtained under authorized access from the database of genotypes and phenotypes (dbGaP) (https://www.ncbi.nlm.nih.gov/gap/). The dbGaP accession numbers and dbGaP hyperlinks for each of the 9 MIGen cohorts are provided in Supplementary Table 1. Restrictions apply to the availability of the MIGen data since the individual-level genetic and phenotypic data are protected due to data privacy laws. The MIGen data is therefore available through controlled access for qualified researchers via an application for authorized access to the National Heart, Lung, and Blood Institute Data Access Committee through the dbGaP authorized access portal (https://dbgap.ncbi.nlm.nih.gov/aa/wga.cgi?page=login). Researchers who would like to obtain the raw data related to MIGen will be presented with a data use certification which requires that the participants will not be re-identified, data be securely stored, and no data will be shared between researchers (who are not identified as study collaborators) or uploaded onto public domains. Any queries pertaining to MIGen data access (including precise conditions of access, contact details for data access requests, and a timeframe for response to data access requests) can be addressed to the National Heart, Lung, and Blood Institute Data Access Committee (nhlbigeneticdata@nhlbi.nih.gov). Genetic data for the 3 GIAB consensus sequences were obtained from the GIAB ftp repository: NA12878; NA24631; NA24385. Data acquisition for the 3 GIAB consensus sequences is further outlined in Supplementary Note 1, section B and can also be found on the Genome In A Bottle (GIAB) Resources webpage (https://jimb.stanford.edu/giab-resources). Individual-level genetic and phenotypic data were also obtained from the UK Biobank (http://www.ukbiobank.ac.uk/), under application #15255, which is further outlined in Supplementary Note 6, section A. Access to the UK Biobank genetic and phenotypic data is also not publicly available and must be obtained via an application (https://www.ukbiobank.ac.uk/register-apply/). Summary-level allele frequency and sequencing coverage information for gnomAD variant sites was obtained from version 2.0.1 of gnomAD exomes (https://gnomad.broadinstitute.org/downloads), which is further described in Supplementary Note 1, section A. Annotated lists of gnomAD variant sites that are stratified by sequencing coverage have been deposited in our public GitHub repository[40]. Lastly, summary statistics containing the beta weights used to generate CVGRS in individuals from the UK Biobank were obtained from the 1000 genomes based CARDIoGRAMplusC4D meta-analysis (http://www.cardiogramplusc4d.org/data-downloads/). Researchers who would like to obtain the raw data related to this study will be presented with a data use certification which requires that the participants will not be re-identified, data be securely stored, and no data will be shared between researchers (who are not identified as study collaborators) or uploaded onto public domains. Source data are provided with this paper.

## Code availibility

The software package containing the code used to conduct the RV-EXCALIBER rare variant association test and its associated documentation has been deposited in a public GitHub repository[40] and can be accessed from https://github.com/GMELab/RV-EXCALIBER (https://doi.org/10.5281/zenodo.5104893). Source data are provided with this paper.

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

## Acknowledgements

This research has been conducted using the MIGen and UK Biobank resources. M.C. is supported by a Canadian Institute of Health Research award. A.L. is supported by an

Ontario Graduate Scholarship award. G.P. is a Tier 2 Canada Research Chair in Genetic and Molecular Epidemiology, and holds a CISCO Professorship in Integrated Health Biosystems. This project has also been supported by a grant from the Heart & Stroke Foundation of Canada (G-19-0026302).

## Author contributions

R.L. and G.P. had full access to all the data and take responsibility for the integrity of the data and the accuracy of the data analysis. R.L. and G.P. conceived and designed the study. R.L. and G.P drafted the study manuscript. R.L., M.C., A.O., P.S., A.L, E.C. and G.P conducted the acquisition, analysis, or interpretation of study data. R.L, M.C., A.O., P.S., A.L, E.C. and G.P performed critical revision of the manuscript for important intellectual content. R.L. performed the bioinformatic and statistical analysis. G.P. obtained study funding and conducted study supervision.

## Competing interests

M.C. has received consulting fees from Bayer. G.P. has received consulting fees from Bayer, Sanofi, Amgen, and Illumina. The remaining authors declare no competing interests.
