## [Peer Review File · Nature Communications]

Reviewers' Comments:

Reviewer #1:

Remarks to the Author:

Summary

The power to detect rare variants with modest effect size is limited by the current statistical techniques. Even with the access to large control sets there remains a risk of large underlying sources of bias in such analyses. The authors proposed a novel framework to calibrate rare variant burden (RV-EXCALIBER) which identified rare variant with modest effects and use them to generate a rare variant gene risk scores (RVGRS) for coronary artery disease (CAD) using publicly available control samples. By using this method, the authors proposed a RVGRS, which strongly associated with CAD in European and South Asian populations. This score shows increased predictiveness of a 1.5% which corresponding to a ≥ 2 -fold increase in odds of presenting with CAD, particularly early CAD in the study population. The calibrated score captures the effects of the RVs through polygenic model of inheritance and acts independently from clinical and common variant risk scores.

Comments

The page number references the number shown at the bottom right of the article's pages. The line number is referring to the number on the left side of the page.

The study appears well designed with thoughtful and careful considerations and large amount of bioinformatics process. It is noted that the manuscript is very technical.

1. Manuscript organization. I have some suggestions for organization and emphasis intended to help with conveying the main messages to readers.

Although the article is written in good depth of technical details, the main findings were scattered through the text, tables and figures. The authors should focus the main findings and put them together in abstract and conclusion, indicating novel findings or findings from novel method.

The authors should reorganize the text by following the experimental narrative line of workflow and findings. For the main text, consider transferring some of technical details into the supplemental materials.

Furthermore, the authors need to make more of an effort to discuss biological mechanisms and implications in the main text, such as the discussion.

A table for abbreviations would be helpful for the reader.

A workflow chart as a display item showing steps, methods used, inputs and outputs, will be helpful to allow readers to follow along the rationale and implementation.

Please consider overall organization of subheadings and subtitles - consider using number if this is allowed by the journal would make it easier to follow.

2. Summary display of findings. It is a little odd to put all figures in main text and all tables in supplement; it's not easy to follow. Please create a table summarizing the score distribution and percentile cutoffs for different study groups and purposes.

3. Please clarify that all cases used were unrelated.

4. Why is this not simply an observation that is skewed largely by FH cases that are driving the association? FH caused by LDLR mutations and other FH genes was known to strongly associate with CAD. As expected therefore in study results, more than a majority of "RVs" were actually pathogenic LDLR mutations. Is it also expected that none of the other FH or related genes were seen in the list? Do authors need to choose another phenotype without known monogenic causes

to test this score such as T2D, metabolic syndrome, etc.?

5. Lipid levels especially LDL-C were not included in case selection of inclusion criteria. This will exclude patients before MI but have increased exposure to high LDL-C, which is a key driver of the CAD risk especially under defected genetic background. It also could be the reason that there is no lipid related gene in RVs list other than LDLR. The same for TG level, as high triglyceride is also a potential cause for CAD. To include lipid levels as inclusion criteria might change the outcomes.

6. The idea that an RVGRS could be transferable across ancestry does not reconcile with the earlier points raised by the authors about the unique differences in rare variants across ancestries. How do the authors explain the two apparently opposing concepts raised in the article?

7. For the top 950 genes, were these genes identified from the rare variant association analysis and would they include the 35 (excluding LDLR) of the 7,146 that harbor a genome-wide significant variant for CAD? If so, how were the remainder 915 genes selected?

8. The RVGRS950 identified participants with an 8% increased odds for developing CAD relative to healthy control ($P=2.1 \times 10^{-5}$) per 1 standard deviation change in RVGRS, would the increase odds be clinically significant? Can the authors be more transparent that the overall findings may be just be biologically hypothesis generating rather than having any ultimate clinical relevance?

9. The median RVGRS950LDLR in early CAD cases corresponded to the 66th percentile of the control distribution and for the general CAD cohort the median was the 52nd percentile. Given that the median of a common variant polygenic score does not really discriminate well those at higher risk than other, the authors should consider display of the top 1, 5, and 10 percentile distribution of the RVGRS950LDLR, as has been done by Khera et al for polygenic scores of common variants.

10. For individuals with a 2-fold increased odds for CAD, what RVGRS950 percentile score do they have?

11. Please justify why the RVGRS association model was adjusted to age-squared.

Minor comments

What does CVGRS stand for? I don't think it is defined in the text.

1. Page 10, line 221 – The authors introduce 5910 discovery MI cases, but is unclear about which cohort these samples are from. Due to the many different cohorts used throughout the manuscript it is important they are clear about which are being used for what analyses.

2. Page 10, lines 231-232 – Previously, it was stated that only LDLR reached exome-wide significance, but here the authors state that there were 35 genes identified with a genome-wide significant variant. It is unclear what the difference is between these genes and the identification of LDLR. Further, the authors previously spoke of exome-wide significance and now are saying genome-wide significance, if this is indeed correct more context must be given.

3. Page 12, lines 266-269 – The odds ratios presented for RVGRS950 seemingly decrease with increased FRS. Would the opposite not have been expected? Or were these results as a result of power differences from sample sizes?

4. Page 16, lines 378-380 – It is mentioned that the CVGRS utilized the CARDIoGRAMplusC4D meta-analysis, which had a case population that was much larger than the MI cohort used in this manuscript. Is there a reason the same cohort was not used for the generation of the RVGRS so that risk estimates could have been compared?

5. Page 17, lines 391-392 – The authors state that the addition of RVGRS will have potential clinical utility, but exactly how it will be clinically relevant is not elaborated on.

Supplementary Appendix:

1. Page 2, lines 57-60 – It is unclear how the variant harmonization was done. As written, it seems as though all samples were sequenced with 5 different technologies and processed with 7 reaper mappers and 3 variant callers, therefore producing 105 datasets for each sample. Is this in fact the case? Could be rephrased for clarity.
2. Page 11, line 189 – Use of mutation, rather than variant introduces inconsistency in the writing.
3. Page 11, lines 195-199 – Does what is written here indicate that the variants were not filtered based on pathogenicity? This section could be written with more clarity.
4. Page 12, line 221 – Does MAF = 0 include variants that were not present in gnomAD, but were present in GIAB, or were those variants excluded?
5. Page 12, line 222 – “Batch effects” typically refers to non-biological sources of heterogeneity in the data, but here it seems to refer to the variation introduced due to differing ethnic group. Please clarify.
6. Page 13, lines 231-236 – How were ‘false positive’ and ‘false negative’ variants identified?
7. Page 17, line 329 – The iCF abbreviation was used prior to its definition here.

Reviewer #3:

Remarks to the Author:

Lali et al. propose a novel method for a gene-based rare-variant burden test with external shared controls. The core idea is based on a new procedure to calibrate the biases in whole-exome sequencing data. With this new burden test, they estimate the gene-based effect sizes for rare variants, construct a weighted rare-variant burden score, and demonstrate its predictive accuracy for CAD.

A gene-based burden test using external controls is not entirely new (notably, TRAPD and ProxECAT). However, these existing methods rely on the patterns of non-functional variants for calibration procedures, thus may still miss subtle biases in functional variants. The proposed method attempt to calibrate study-specific biases using the functional variants in cohort-specific control samples. This is a novel approach that can potentially produce better calibration. I think this is a great idea of interest to the community if the following concerns are addressed.

Lines 61-63 "there is currently no method to calibrate rare variant burden between test samples and public databases," lines 293-295 "methods have been developed to address the genetic contributors to Mendelian disease using external control databases," and lines 302-304 "current methodologies that leverage gnomAD as controls have been tailored to demarcate variant and gene-based associations for Mendelian disorders." I am not convinced that existing methods (e.g. TRAPD and ProxECAT) are limited to the application for Mendelian disorders only. These methods are developed as burden tests for complex traits with locus heterogeneity. ProxECAT should be cited (PMID: 30325923).

Figure 1: The simulations here seem to have considered only one bias each time while setting the other biases to 0. For example, I think Figs 1A and D simulated a range of sequencing false-negative rates under no sequencing false positive or population difference, and so on. If this is the case, this is too unrealistic. The unbiasedness or conservativeness of estimated correction factors cannot be concluded under such limited simulation conditions. Simulations need to be done in presence of all three biases (or at least two: sequencing false negatives and false positives).

Lines 142-144 (Figure 2C): NA12878 is not a "Central European." It is a CEU sample (CEPH from Utah, with Northern and Western European ancestry). In principle, it should show very close proximity to gnomAD Non-Finnish European ancestry. The extensive deviation observed in NA12878 across all frequency bins is very concerning. This sample should be very representative

of gnomAD Europeans. What is an explanation for this?

Lines 165-167 (Figure 3A): I am skeptical that "exome-wide rare variant mutation load" is a primary factor driving the observed inter-cohort differences in mean iCF. Each individual has a different mutational load, but the mean difference of mutational load is < 3% across European ancestry (PMID: 25581429, see Fig 1).

Lambda_med comparison against TRAPD (lines 206-207, 315-316 and Figure 4): It is well-known that lambda_med is not a good benchmark statistic to evaluate calibration for rare-variant burden tests because of inadequate continuity correction of Fisher's Exact Test at P=1 tail. Because of this, lambda_95 is a more adequate stat here, or to use lambda_med, underpowered genes with low numbers of rare functional variants should be excluded. It seems to me that lambda_95 is not much different between TRAPD and RV-EXCALIBER. Rather, I am concerned about the potential over-correction of RV-EXCALIBER at the significant tail. RV-EXCALIBER base is over-correcting (Fig S5). Including iCF and gCF on top of RV-EXCALIBER base helps, but there seem to be lingering over-correction issues at the significant tail, thereby potentially loss of power (Fig S6). I am particularly concerned because binning-based calibration techniques, which underlies RV-EXCALIBER, tend to err at the bin of extreme tails in general.

Benchmark comparison against TRAPD. The authors need to show a power comparison of TRAPD and RV-EXCALIBER. This could be done in simulation or real data (ideally both). In simulations, the authors should allow realistic combinations of all biases at the same time.

A potential strength of the new method may be the simplicity of application to heterogeneous cohorts. Existing methods can be applied to such a dataset, but they need to be run in each cohort separately, and then meta-analysis should be done across cohorts while accounting for overlapping shared controls.

Rare-variant risk score. It is very exciting to see the significant predictive power of rare variant burden score. This is great. How much improvement is there in predictive power, compared to other approaches, e.g. a rare-variant burden using TRAPD-derived ORs, or any other approaches?

Minor:

line 86: "due to?"

Figure legends: In a few places, equations were referred without explaining what and how things were calculated. These need to be elaborated.

line 268: FRS (Need to explain it's a Framingham risk score for the readers).

Net reclassification index: NRI is useful, but it's not very comparable across different studies. AUC and Nagelkerke's R^2 are more widely reported for PRS.

Revisions made in addition to reviewer comments:

In addition to making all changes to the manuscript main text, supplemental appendix, and figure legends in accordance with the reviewer's comments, we also made minor transcriptional changes to Supplementary Table 12-14 and Figure 7. These changes do not affect interpretation of results in any regard. These changes are provided in red below:

Supplementary Table 12: Effect estimates of RVGRS950 and RVGRS950^{LDLR-} on CAD across tertiles of FRS in the UK Biobank.

FRS tertile ^a	OR per SD (95% CI)	P-value	P _{interaction}
RVGRS950			
1	1.16 (1.06-1.27) 1.16 (1.06-1.26)	8.4x10 ⁻⁴ 7.3x10 ⁻⁴	0.015 0.018
2	1.10 (1.03-1.17) 1.10 (1.04-1.17)	2.3x10 ⁻³ 1.6x10 ⁻³	
3	1.03 (0.99-1.08) 1.04 (0.99-1.09)	0.17 0.13	
RVGRS950^{LDLR-}			
1	1.15 (1.05-1.26) 1.15 (1.05-1.25)	2.0x10 ⁻³ 2.0x10 ⁻³	0.019 0.024
2	1.09 (1.02-1.16) 1.09 (1.03-1.16)	6.2x10 ⁻³ 4.0x10 ⁻³	
3	1.03 (0.98-1.08) 1.03 (0.98-1.08)	0.29 0.22	

^a Effect estimates and P-values for each tertile were determined using a logistic regression model that used age, age², sex, and the first 20 principle components as covariates

Supplementary Table 13: Effect estimates of CVGRS on tertiles on CAD across tertile of FRS in the UK Biobank.

FRS tertile ^a	OR per SD (95% CI)	P-value	P _{interaction}
1	1.20 (1.14-1.26) 1.36 (1.24-1.49)	5.3x10 ⁻¹² 2.1x10 ⁻¹¹	0.65 0.74
2	1.14 (1.11-1.18) 1.26 (1.19-1.34)	2.2x10 ⁻¹⁴ 3.2x10 ⁻¹⁴	
3	1.17 (1.13-1.20) 1.30 (1.25-1.37)	< 2.2x10 ⁻¹⁶ < 2.2x10 ⁻¹⁶	

^a Effect estimates and P-values for each tertile were determined using a logistic regression model that used age, age², sex, and the first 20 principle components as covariates

Supplementary Table 14: Net reclassification improvement index when incorporating RVGRS950 and RVGRS950^{LDLR-} to risk models.

Reference model ^a	Test model ^a	NRI index ^b	P-value for NRI index
RVGRS950			
FRS	FRS + RVGRS	0.0542	1.7x10 ⁻³

		0.0548	1.1x10 ⁻³
CVGRS	CVGRS + RVGRS	0.0554	1.3x10 ⁻³
		0.0541	1.3x10 ⁻³
FRS + CVGRS	FRS + CVGRS + RVGRS	0.0571	9.4x10 ⁻⁴
		0.0558	9.2x10 ⁻⁴
RVGRS950^{LDLR-}			
FRS	FRS + RVGRS	0.0438	0.0011
		0.0435	9.7x10 ⁻³
CVGRS	CVGRS + RVGRS	0.0446	9.7x10 ⁻³
		0.0436	9.6x10 ⁻³
FRS + CVGRS	FRS + CVGRS + RVGRS	0.0443	0.0010
		0.0451	7.4x10 ⁻³

^a All reference and test models use age, age², sex, and the first 20 principle components as covariates

^b NRI index is comparing the test model to the reference model

Figure 7:

The P-values for “RVGRS950 with full adjustment” and “RVGRS950^{LDLR-} with full adjustment” were changed to **2.9x10⁻⁵** and **2.3x10⁻⁴**, respectively. The old P-values were 5.4x10⁻⁵ and 4.5x10⁻⁴, respectively.

We have provided the corresponding changes to our main manuscript (where applicable) below:

*“RVGRS950 and RVGRS950^{LDLR-} acted independently from both the aggregate effect of common variants and clinical risk factors, with a consistent predictive estimates (OR=1.07; 95% CI, 1.04-1.11; **P=2.9x10⁻⁵** for RVGRS950 and OR=1.06; 95% CI, 1.03-1.10; **P=2.3x10⁻⁴** for RVGRS950^{LDLR-}) (Figure 7).”* (Updated main text page 13; lines 295-298)

“Moreover, we observed a significant interaction between RVGRS950 and FRS, where individuals with higher RVGRS950 had lower clinical risk for CAD ($P_{interaction}=0.018$). Specifically, individuals in the low, middle, and high tertiles of a Framingham risk score (FRS) had odds ratios of 1.16, 1.10, 1.03 conferred through RVGRS950 for CAD, respectively. The observed interaction also remained significant when using RVGRS950^{LDLR-} (Supplementary Table 12). Interestingly, the effect conferred through a common variant genetic risk score (CVGRS) did not significantly differ across FRS tertiles, with odds ratios of 1.36, 1.26, and 1.30

for low, middle, and high tertiles of FRS, respectively ($P_{interaction}=0.74$) (Supplementary Table 13). We next tested whether addition of RVGRS950 could improve patient classification as compared to a base model consisting of only CVGRS and FRS. Incorporation of RVGRS950 was found to significantly improve risk prediction for CAD (*net reclassification index*=0.0558; $P=9.2 \times 10^{-4}$) after adjusting models for age, age², sex and the first 20 principal components of ancestry. The net reclassification index also remained significant when using RVGRS950^{LDL} (Supplementary Table 14).” (Updated main text page 13; lines 298-311)

“In fact, adding RVGRS to a reference model consisting of CVGRS and clinical risk factors significantly reclassified 5.6% of CAD events, which represents a marked improvement given the relatively small size of our discovery population.” (Updated main text page 18; lines 425-428).

Reviewer #1 (Remarks to the Author):

Main Comments:

The page number references the number shown at the bottom right of the article’s pages. The line number is referring to the number on the left side of the page. The study appears well designed with thoughtful and careful considerations and large amount of bioinformatics process. It is noted that the manuscript is very technical.

We sincerely thank the reviewer for their support of our work. We believe that the changes made in accordance with the reviewer’s comments have made our work much stronger and hope that our thorough responses to the reviewer’s comments address all concerns.

1. Manuscript organization. I have some suggestions for organization and emphasis intended to help with conveying the main messages to readers.

Although the article is written in good depth of technical details, the main findings were scattered through the text, tables and figures. The authors should focus the main findings and put them together in abstract and conclusion, indicating novel findings or findings from novel method.

We thank the reviewer for this comment and understand that there was a large amount of results presented throughout the paper. As the abstract has a word limit of 150 words (which we were already over by 6 words), we have elected to substantiate our conclusion by including a fuller description of final results, which we believe will be the best section to consolidate findings. We have provided the updated conclusion below:

“We herein present a method (RV-EXCALIBER) for calibrating rare pathogenic alleles between case exomes and gnomAD by mitigating major biases at the individual and gene-level while also correcting for putative LD between rare variants. RV-EXCALIBER demonstrates better calibration than existing methodologies that leverage gnomAD as controls and was empirically applied to generate a first RVGRS for CAD. The RVGRS conferred disease risk independent of Mendelian effects, clinical risk factors, and CVGRS while also correctly reclassifying ~6% of CAD events, *which suggests that addition of RVGRS to CVGRS and Mendelian mutations may have potential clinical utility in terms of case discrimination, but further validation is necessary*

before establishing a full case for clinical implementation While the RVGRS explained only 0.1% of population variance in CAD affection status after accounting for known genetic and environmental factors, it still demarcated high individual-level CAD risk (i.e. 1.5% of individuals with ≥ 2 -fold risk). Lastly, the RVGRS demonstrates higher predicative effect among individuals with low clinical risk burden and those with early-onset CAD. These findings suggest an enriched genetic contribution to CAD in the absence of environmental risk factors and further validate the increased role inheritance in modulating disease risk in early disease^{26,29,30}. Our results also make a strong case for large sequencing studies of common, complex traits and diseases. Additionally, leveraging publicly available data as controls in an unbiased manner may permit additional resources to be allocated to recruiting extreme disease cases, which might harbor a greater burden of rare causal variants. Lastly, we also note that RV-EXCALIBER could be extended to whole-genome sequencing, in order to better delineate the role of rare non-coding variants on disease risk. Our results also make a strong case for large sequencing studies of common, complex traits and diseases. Additionally, leveraging publicly available data as controls in an unbiased manner may permit additional resources to be allocated to recruiting extreme disease cases, which might harbor a greater burden of rare causal variants. Lastly, we also note that RV-EXCALIBER could be extended to whole-genome sequencing, in order to better delineate the role of rare non-coding variants on disease risk.” (Updated main text page 20-21 lines 466-485)

The authors should reorganize the text by following the experimental narrative line of workflow and findings. For the main text, consider transferring some of technical details into the supplemental materials. Furthermore, the authors need to make more of an effort to discuss biological mechanisms and implications in the main text, such as the discussion.

The reviewer brings forth a very valid point. As our paper is methods-based, we gave a relatively smaller focus on the biological mechanisms that underly our top discovery genes. We addressed this by conducting a gene-set enrichment analysis to identify whether there was enrichment of Gene Ontology (GO) biological processes among genes nominally associated ($P < 0.05$) with CAD according to RV-EXCALIBER. Using the GOrilla tool, we identified nominal enrichment in the following GO biological processes, which are now presented in Supplementary Table 9:

Supplementary Table 9: Gene-set enrichment for nominally associated ($P < 0.05$) discovery genes from RV-EXCALIBER.

GO term	Description	Fold enrichment	P-value
GO:0009605	response to external stimulus	1.79	8.46×10^{-6}
GO:0042221	response to chemical	1.48	5.02×10^{-5}
GO:0050896	response to stimulus	1.31	6.99×10^{-5}
GO:0010033	response to organic substance	1.52	1.13×10^{-4}
GO:0045630	positive regulation of T-helper 2 cell differentiation	20.47	1.16×10^{-4}
GO:0045628	regulation of T-helper 2 cell differentiation	20.47	1.16×10^{-4}
GO:0060059	embryonic retina morphogenesis in camera-type eye	20.47	1.16×10^{-4}

GO:1903409	reactive oxygen species biosynthetic process	6.82	1.47 x10 ⁻⁴
GO:0006950	response to stress	1.4	4.74 x10 ⁻⁴
GO:1901700	response to oxygen-containing compound	1.58	5.39 x10 ⁻⁴
GO:0007616	long-term memory	6.4	7.56 x10 ⁻⁴
GO:0009894	regulation of catabolic process	1.75	9.26 x10 ⁻⁴
GO:0006809	nitric oxide biosynthetic process	8.19	9.30 x10 ⁻⁴
GO:0042592	homeostatic process	1.52	9.47 x10 ⁻⁴

This new analysis is presented in the updated main text and updated supplementary appendix as follows:

“We also elected to conduct a gene-based enrichment analysis using genes that were nominally (i.e. $P < 0.05$) associated with CAD in the discovery rare variant association to elucidate if any Gene Ontology (GO) biological processes were enriched. Although no biological processes reached statistical significance after multiple hypothesis correction, we observed nominal enrichment of “reactive oxygen species biosynthetic process” (GO:1903409) (fold enrichment=6.82; $P=1.47 \times 10^{-4}$), “response to stress” (GO:0006950) (fold enrichment=1.40; 4.74×10^{-4}), “response to oxygen-containing compound” (GO:1901700) (fold enrichment=1.58; $P=5.39 \times 10^{-4}$), and “nitric oxide biosynthetic pathway” (GO:0006809) (fold enrichment=8.19; $P=9.30 \times 10^{-4}$). A complete list of nominally associated biological processes can be found in Supplementary Table 9.” (Updated main text page 11; lines 250-259)

“Indeed, while LDLR demonstrated both biological and statistical significance, a gene-set enrichment analysis revealed that top discovery genes were nominally enriched for known biological processes involved in CAD onset, including response to stress, biosynthesis/response to reactive oxygen species, and nitric oxide biosynthesis. As such, assessing top genes individually or as a gene-set may provide biological insights regardless of whether these genes show mild deflation at the significant tail of the distribution.” (Updated main text page 17-18; lines 402-407)

“A separate gene-set enrichment was evaluated using the GOrilla tool on the all nominally associated discovery genes ($P < 0.05$) relative to all discovery genes with an observed and expected count ≥ 1 and ≥ 10 , respectively (7,146 total). A total of 12,958 Gene Ontology (GO) biological processes were evaluated for enrichment. Fold enrichment was calculated according to the ratio of the proportion of target genes (i.e. discovery genes with $P < 0.05$) to the proportion of all genes (i.e. all discovery genes) in a given GO biological process. Enrichment P-values were calculated from a minimum hypergeometric test statistic as discussed by Eden et al. 2009⁴⁰.” (Updated supplementary appendix pages 19-20; lines 401-408)

A table for abbreviations would be helpful for the reader.

We thank the reviewer for this useful comment. A table of abbreviations has now been added to the beginning of the manuscript and has been provided below:

LIST OF ABBREVIATIONS

ASJ	Ashkenazi Jews
ATVB	Italian Atherosclerosis Thrombosis and Vascular Biology
CAD	Coronary artery disease
CF	Correction factor
CI	Confidence interval
CVGRS	Common variant genetic risk score
EAS	East Asians
ExAC	Exome Aggregation Consortium
FES	Tyrosine protein kinase Fes/Fps
FH	Familial hypercholesterolemia
FRS	Framingham risk score
gCF	gene correction factor
GIAB	Genome In A Bottle
GO	Gene Ontology
gnomAD	genome Aggregation Database
iCF	individual correction factor
LD	Linkage disequilibrium
LDLR	Low-density lipoprotein receptor
MAF	Minor allele frequency
MDC	Malmö Diet Cancer
MI	Myocardial infarction
MIGen	Myocardial Infarction Genetics
NFE	Non-Finnish Europeans
OR	Odds ratio
PSF	Population specific factors
QD	Quality-by-depth
REGICOR	Registre Gironi del Cor
RV-EXCALIBER	Rare Variant EXome CALIBration using External Repositories
RVGRS	Rare variant genetic risk score
RVGRS950	Rare variant genetic risk score using top 950 discovery genes
RVGRS950 ^{LDLR}	Rare variant genetic risk score using top 950 discovery genes (excluding LDLR)
SFN	Sequencing false negatives
SFP	Sequencing false positives
SNV	Single nucleotide variants
TRAPD	Testing RAre variants with Public Data

A workflow chart as a display item showing steps, methods used, inputs and outputs, will be helpful to allow readers to follow along the rationale and implementation.

This is an excellent suggestion made by the reviewer. We have added in a workflow diagram in the figures section (now Figure 1; shown below with caption) which outlines the main steps of RV-EXCALIBER. This workflow is meant to provide an intuitive guide to the processing steps involved in generating gene-based rare variant associations using RV-EXCALIBER.

“Figure 1: Schematic describing the RV-EXCALIBER pipeline. RV-EXCALIBER can be divided into 4 distinct stages where stage 1 and 2 are preprocessing steps while stages 3 and 4 encapsulate the three major RV-EXCALBER adjustments: the implementation of the correction factors (iCF and gCF) and z-score calculations for per-gene delta counts. MAF indicates minor allele

frequency, iCF indicates individual correction factor, gCF indicates gene correction factor, NFE indicates non-Finnish European, AFR indicates Africa, SAS indicates South Asian, EAS indicates East Asian, and AMR indicates Latino.”

Please consider overall organization of subheadings and subtitles - consider using number if this is allowed by the journal would make it easier to follow.

We thank the reviewer for the suggestion. Numbers have now been added to all headings and subheadings to facilitate manuscript readability and organization. An example is provided below from the updated main text:

“3. RESULTS”

“3.1 Single gene simulations using the correction factors (CF)”

Please note that the journal may use another means (besides numbering) to facilitate organization of headings and sub-headings.

2. Summary display of findings. It is a little odd to put all figures in main text and all tables in supplement; it's not easy to follow. Please create a table summarizing the score distribution and percentile cutoffs for different study groups and purposes.

We thank the reviewer for this suggestion. We have created a main-text table that incorporates the following data as per the reviewer’s comments:

- 1) Summary (mean and standard deviation) of the **a)** RVGRS950 and **b)** RVGRS950^{LDLR}-score distributions for the overall UK Biobank population.
- 2) Summary (mean and standard deviation) of the **a)** RVGRS950 and **b)** RVGRS950^{LDLR}-score distributions for the UK Biobank population stratified by tertile of Framingham risk score (FRS).
- 3) Summary (mean and standard deviation) of the RVGRS950^{LDLR}-score distribution for the different ‘coronary artery disease age-of-onset’ categories in the UK Biobank population
- 4) Summary (mean and standard deviation) of the RVGRS950^{LDLR}-score distribution for individuals with ≥ 2 -fold risk for early CAD.

Table 1: Summary of the RVGRS950 and RVGRS950^{LDLR}-distributions in the UK Biobank validation population.

	UK Biobank CAD cases	UK Biobank CAD-free controls
Overall population		
n	3,843	42,007
RVGRS950 mean (SD)	0.039 (0.69)	-0.009 (0.67)
RVGRS950^{LDLR}-mean (SD)	0.033 (0.68)	-0.009 (0.67)
FRS		
FRS tertile 1		
n	519	14,765
RVGRS950 mean (SD)	0.086 (0.71)	-0.013 (0.67)

RVGRS950 ^{LDLR} - mean (SD)	0.075 (0.70)	-0.013 (0.67)
FRS tertile 2		
n	1,193	14,090
RVGRS950 mean (SD)	0.046 (0.70)	-0.013 (0.67)
RVGRS950 ^{LDLR} - mean (SD)	0.039 (0.70)	-0.013 (0.67)
FRS tertile 3		
n	2,131	13,152
RVGRS950 mean (SD)	0.025 (0.67)	0.0014 (0.68)
RVGRS950 ^{LDLR} - mean (SD)	0.020 (0.66)	0.00083 (0.67)
Age strata		
Males ≤40 and Females ≤45		
n	50	42,007
RVGRS950 ^{LDLR} - mean (SD)	0.20 (0.76)	-0.0088 (0.67)
Males ≤45 and Females ≤50		
n	189	42,007
RVGRS950 ^{LDLR} - mean (SD)	0.13 (0.70)	-0.0088 (0.67)
Males ≤50 and Females ≤55		
n	536	42,007
RVGRS950 ^{LDLR} - mean (SD)	0.082 (0.70)	-0.0088 (0.67)
Males ≤55 and Females ≤60		
n	1,128	42,007
RVGRS950 ^{LDLR} - mean (SD)	0.048 (0.69)	-0.0088 (0.67)
≥2-fold risk for early CAD *		
n	670	45,180
RVGRS950 ^{LDLR} - mean (SD)	2.00 (0.35)	-0.035 (0.63)

* Cases and controls refer to individuals with ≥2-fold risk and <2-fold risk of early CAD, respectively

We also reference this table in the main text:

“The population distribution of RVGRS950 and RVGRS950^{LDLR} are provided in Table 1 for the main population strata used from UK Biobank.” (Updated main text page 14; lines 328-330)

We fully agree with the reviewer that this summary table will help consolidate all our findings pertaining to the RVGRS and will serve as a reference point for readers.

3. Please clarify that all cases used were unrelated.

We thank the reviewer for this keen observation. First, we have noted that the ranking MIGen controls were unrelated:

*“In order to evaluate the adequacy of gCF adjustment in the absence of disease associations, we randomly divided healthy **unrelated** controls from MIGen (N=6,082) into a ranking cohort (N=2,730) in which the gCF would be calculated based on the distribution of gene-based P-values and expected counts and a test cohort (N=3,352) in which the gCF would be applied (see Data Supplement).”* (Updated main text page 8; line 187)

Secondly, we have noted that all MIGen discovery cases are unrelated:

*“After adjusting gnomAD expected counts with RV-EXCALIBER, we conducted an exome-wide rare variant association on protein-coding genes across 5,910 discovery **unrelated** MI cases from the MIGen consortium, testing all 7,146 genes with an observed count ≥ 1 and an expected count of ≥ 10 (again, to filter out underpowered genes).”* (Updated main text page 10; lines 232-235)

Lastly, we have noted that all participants in the validation populations were also unrelated: UK Biobank and PROMIS:

*“Each RVGRS was calculated in each case (N=3,843) and control (N=42,007) in **unrelated** UK Biobank European participants, incrementally including the top 10 to top 3,000 genes (based on P-value) from the MIGen association (increments of 10 genes; 300 total scores).”* (Updated main text page 11; line 264-267)

*“Although RVGRS950 did not reach significance in **unrelated** South Asians from the Pakistan Risk of Myocardial Infarction Study (PROMIS) (N=2,946 for cases and N=3,708 for controls), there was still a similar predictive effect observed (OR=1.04; 95% CI, 0.99-1.08; P=0.096).”* (Updated main text page 12; line 275-278)

4. Why is this not simply an observation that is skewed largely by FH cases that are driving the association? FH caused by LDLR mutations and other FH genes was known to strongly associate with CAD. As expected therefore in study results, more than a majority of "RVs" were actually pathogenic LDLR mutations. Is it also expected that none of the other FH or related genes were seen in the list? Do authors need to choose another phenotype without known monogenic causes to test this score such as T2D, metabolic syndrome, etc.?

This is a fantastic point brought up by the reviewer as genetically identified FH cases have a substantially large risk for CAD and FH mutation status has demonstrated strong associations with CAD in previous literature, including but not limited to: 1) Khera *et al.* 2016 (DOI: 10.1016/j.jacc.2016.03.520) 2) Abul-Husn *et al.* 2016 (DOI: 10.1126/science.aaf7000) 3) Khera *et al.* 2018 (DOI: 10.1161/CIRCULATIONAHA.118.035658). This is especially true for cases of early CAD. As such, we elected to construct our RVGRS both with and without incorporation of *LDLR* (i.e. RVGRS950^{*LDLR*}), which was the only FH gene found in the RVGRS950. Indeed, our results show that RVGRSLDLR- demonstrates a similar predictive effect and strength

association as compared to RVGRS950 (OR 1.08; 95% CI 1.04-1.11; $P = 2.1 \times 10^{-5}$ for RVGRS950 and OR 1.07; 95% CI 1.03-1.10; $P = 1.5 \times 10^{-4}$ for RVGRS950^{LDLR-}). Given these consistent effects and association signals, the RVGRS950 is largely acting through the aggregate, modest effects of other CAD risk genes. In fact, the RVGRS950 correctly and significantly reclassifies 4.4% of CAD cases when incorporated into a base model consisting of other genetic and clinical CAD risk factors (see Supplementary Table 13).

The reviewer's point about not observing other FH genes among the top discovery genes can be addressed with the following two points:

- 1) None of the individual cohorts from the Myocardial Infarction Genetics (MIGen) consortium were selected on the basis of lipid traits that are risk factors for CAD (e.g. low-density lipoprotein (LDL) cholesterol, triglycerides; see inclusion criteria in Supplementary Table 1 in updated supplementary appendix). As such, we do not necessarily expect other FH genes (i.e. Apolipoprotein B (*APOB*) and Preprotein convertase subtilisin/kexin type 9 (*PCSK9*)), which confer risk via increased LDL cholesterol, to be strongly associated to CAD status.
- 2) The number of pathogenic mutations in both *APOB* and *PCSK9* that are causal for FH are not nearly as vast as those for *LDLR*. For *APOB*, pathogenic FH mutations must be localized within its LDLR-binding domain, which is the protein region that interacts with membrane-bound LDLR to facilitate receptor-mediated endocytosis of circulating LDL particles. Pathogenic mutations anywhere else in the *APOB* gene result in hypobetalipoproteinemia, which is protective for CAD. In fact, the LDLR-binding domain contributes to only 0.1% of the 4536 amino acids in *APOB*, which harbours only 5 established FH causing mutation prevalent in Europeans (p.Arg3527Gln, p.Asn3580Ser, p.Leu4384Pro, p.His3603Arg, p.Asn3590Lys). Therefore, the probability of identifying a mutation by chance in this region is low. For *PCSK9*, pathogenic FH mutations must result in a gain-of-function effect, which results in enhanced lysosomal-mediated degradation of LDLR, lead to a diminished receptor-mediated endocytosis of circulating LDL particles. Gain-of-function mutations causal for FH in *PCSK9* are rare compared to their loss-of-function counterparts due to purifying selection and there are only 3 well-defined *PCSK9* gain-of-function mutations present in the European population (p.Arg476Cys, p.Ala53Gly, p.Arg96Cys, p.Glu501=).

As such, the absence of other FH genes is due largely to a combination of 1) inclusion criteria and 2) the class/type of mutation required to necessitate an FH-causing mutation (i.e. a mutation that results in increased circulating LDL).

The reviewer also brings up a formidable point about testing a RVGRS for complex diseases that do not have strong Mendelian contributions (e.g. type 2 diabetes or metabolic syndrome). Assessing the impact of a RVGRS would be very useful for these disorders in order to ascertain 1) what proportion of individuals would be at high-risk and 2) identifying the proportion cases that can be correctly re-classified. This will be particularly interesting in discovering the missing heritability for such disorders in future work. Nevertheless, assessing the contribution of RVGRS to complex disorders with Mendelian contributors (e.g. CAD) can be evaluated by generating the RVGRS based on non-Mendelian genes (e.g. RVGRS950^{LDLR-}) as we have

shown. This approach can provide insight into the impact of the polygenic effect of rare variants that is not due to Mendelian effects.

5. Lipid levels especially LDL-C were not included in case selection of inclusion criteria. This will exclude patients before MI but have increased exposure to high LDL-C, which is a key driver of the CAD risk especially under defected genetic background. It also could be the reason that there is no lipid related gene in RVs list other than LDLR. The same for TG level, as high triglyceride is also a potential cause for CAD. To include lipid levels as inclusion criteria might change the outcomes.

The reviewer rightly states that the absence of lipid-related inclusion criteria for the MIGen cases is a reason for why other FH genes and other lipid genes (e.g. those that affect triglycerides) are not observed in the top discovery genes. This reason is further discussed in our response to the previous main comment (main comment 4) posed by the reviewer. The reviewer also correctly states that having a lipid-related inclusion criteria would change the top discovery genes associated with MI, which may correspondingly change the effect size and strength of association of the RVGRS. However, instituting inclusion criteria that are agnostic of lipid-related traits lends itself to the discovery of novel risk genes that are involved in biological pathways that are not as well understood in the context of CAD/MI. Moreover, non-lipid-related inclusion criteria allows for unbiased prevalence estimates of monogenic conditions (e.g. FH) in a population afflicted with CAD/MI. This information is particularly useful for informing appropriate genetic screening procedures in these vulnerable populations.

6. The idea that an RVGRS could be transferable across ancestry does not reconcile with the earlier points raised by the authors about the unique differences in rare variants across ancestries. How do the authors explain the two apparently opposing concepts raised in the article?

The reviewer correctly addresses two points in the main text that read as contradictory. These points are 1) that rare variants exhibit geographical specificity relative to common variants (i.e. the composition of individual rare variants tend to be specific to given ancestries compared to common variants):

“These adjustments are especially vital for rare variants as they have shown to exhibit a higher degree of geographical specificity compared to common variants...” (Updated main text page 3; lines 79-81)

and 2) that the RVGRS (which is composed of rare variants) demonstrates consistent impact on CAD across ancestries:

“In fact, we identified that the effect estimates generated from all RVGRS (i.e. top 10 to top 3000 genes) did not significantly differ between the UK Biobank and PROMIS according to a test for heterogeneity (Figure 8 and Supplementary Table 10). As a negative control, we tested the same 300 RVGRS on UK Biobank participants with irritable bowel disease (N=3,058) and identified no significant difference in distribution of RVGRS between case and control subjects.” (Updated main text page 12; lines 280-285)

“Indeed, we show that the RVGRS demonstrates consistent predictive estimates for Europeans in UK Biobank exomes and South Asian exomes in PROMIS (Figure 8).” (Updated main text page 18; lines 420-421)

These statements understandably seem conflicting without a more substantiating explanation. We have added the following description which we believe effectively reconciles these points and lends more insight to rare variant biology:

“Moreover, since the RVGRS assesses the burden of rare variants at the gene-level, it does not require identical variants be present in an independent population. Instead, the RVGRS leverages the ubiquitous function of genes across ancestries to confer consistent effects, despite the geographical specificity of individual rare variants.” (Updated main text pages 18; lines 416-420)

We apologize for the confusion and would like to thank the reviewer for bringing up this point. We believe the added description will serve to enhance the reader’s understanding of the underlying mechanism of RVGRS and why it is able to produce consistent predictive effects across diverse ancestral populations.

7. For the top 950 genes, were these genes identified from the rare variant association analysis and would they include the 35 (excluding LDLR) of the 7,146 that harbor a genome-wide significant variant for CAD? If so, how were the remainder 915 genes selected?

The RVGRS950 refers to the RVGRS generated from the top 950 most associated genes with CAD after incorporating the iCF, gCF, and accounting for linkage disequilibrium between rare variants, which are the 3 levels of correction that RV-EXCALIBER institutes. These genes were identified through gene discovery association analysis using 5,910 MIGen cases and the gnomAD non-Finnish Europeans as controls. A total of 300 RVGRS were generated using the top 10 to top 3000 (increments of 10) gene-based associations in the gene discovery analysis. After conducting our discovery gene-based association, we limited genes to those that are well-powered (i.e. with an observed count ≥ 1 and expected count ≥ 10). This filtering resulted in a total of 7,146 discovery genes, which the reviewer mentioned in the comment. The 35 genes that harbour a genome-wide significant variant for CAD were present among the total 7,146 and were not necessarily present in the top 950 genes used to generate the most predictive RVGRS. The purpose of identifying the genes that harbour a genome-wide significant variant across the total 7,146 well-powered genes was to identify whether their combined association test statistics were larger (i.e. statistically significant) than would be expected under a null-distribution of gene-sets of identical size.

We have now reworded this section to make it more compressible as where we obtained the 35 genes used to detect enrichment of CAD genes:

“In fact, a total of 35 (excluding LDLR) of the 7,146 genes assessed in the exome-wide rare variant association were found to harbor a genome-wide significant single nucleotide polymorphism for CAD (i.e. a CAD gene set) in the CARDIoGRAMplusC4D genome-wide

association study. After generating a distribution of Fisher's combined probability test statistics for 100,000 random gene sets of equivalent size under the null of no individual gene associations, we identified that the probability of observing a combined test statistic greater than or equal to that of the CAD gene set was statistically significant ($P = 0.014$)." (Updated main text pages 10-11; lines 243-250)

8. The RVGRS950 identified participants with an 8% increased odds for developing CAD relative to healthy control ($P=2.1 \times 10^{-5}$) per 1 standard deviation change in RVGRS, would the increase odds be clinically significant? Can the authors be more transparent that the overall findings may be just be biologically hypothesis generating rather than having any ultimate clinical relevance?

This is a very fair point brought forth by the reviewer. Our results provide a proof-of-concept and predictive estimates are likely to improve with increasing discovery sample sizes. We make mention of this in our main text:

"It is important to note that the risk estimates for high polygenic risk derived from RVGRS and CVGRS cannot be readily compared due to the extreme discrepancy in discovery sample size used to compute effect estimates to weight alleles in validation samples. In our work, we included 5,910 cases to derive beta estimates while the betas used for CVGRS in CAD/MI are obtained from the CARDIoGRAMplusC4D meta-analysis, which has a case population that is 10.3x larger." (Updated main text page 20; lines 454-458)

Moreover, clinical usefulness is likely to be attribute to the extreme of the RVGRS distribution, which is driven by the small fraction of individuals whose risk is predicted to be clinically significantly elevated (i.e. individuals with ≥ 2 -fold risk). Nevertheless, we fully agree with the reviewer about the importance of transparency with regard to the clinical utility of the RVGRS in its current state. As such, we have added the below statement to the main text that addresses this point, which highlights the use of RVGRS as an **adjunct** to Mendelian variants and common variant genetic risk scores:

*"Nevertheless, our work bridges the gap between rare and common variants under a unified polygenic framework. Despite the marked difference in case sample size, we show that a RVGRS confers high risk (i.e. ≥ 2 -fold) for early CAD in 1.5% of the population and remains independent of Mendelian effects, clinical risk factors, and CVGRS. **Notwithstanding its independent effect on CAD and ability to demarcate individual-level risk, we stress that a RVGRS should be used as an adjunct genetic variable that is used in conjunction with Mendelian variants and CVGRS to enhance the overall clinical utility of genetic factors.**"* (Updated main text page 20; line 459-465)

We believe this statement will make clear that the RVGRS is not meant to be used as a replacement genetic variable to be used instead of Mendelian and common variant genetic risk score when assessing in individual's risk of disease, but rather functions as supplemental genetic feature that is to be used in conjunction with traditional genetic factors.

In addition, we have also noted the hypothesis-generating nature of the findings with respect the Framingham risk score, in which an increased predictive effect was observed among individuals with low clinical risk factor burden:

“When evaluating the potential clinical efficacy of the RVGRS, we observed that it demonstrated an increase in predictive effect for individuals at low clinical risk as compared to those with a high clinical burden for developing CAD. We acknowledge this observation to be hypothesis-generating, wherein individuals with low clinical risk may have an enrichment of rare risk alleles of modest effect (i.e. RVGRS) compared to those with moderate or high clinical risk. Future work on gene x environment interactions that focusses on rare variants are required to evaluate the clinical utility of RVGRS, particularly among individuals with low versus high burden of environmental risk.” (Updated main text page 19; lines 439-446)

We again thank the reviewer for this comment as we believe that these two changes in description provide more transparency on the clinical efficacy of the RVGRS, which will be beneficial to readers, especially those with a clinical background.

9. The median RVGRS950LDLR in early CAD cases corresponded to the 66th percentile of the control distribution and for the general CAD cohort the median was the 52nd percentile. Given that the median of a common variant polygenic score does not really discriminate well those at higher risk than other, the authors should consider display of the top 1, 5, and 10 percentile distribution of the RVGRS950LDLR, as has been done by Khera et al for polygenic scores of common variants.

The reviewer makes an excellent point that identifying the odds of CAD at the high tail of the RVGRS distribution (relative to the rest of the population) may better demarcate individuals with high versus low CAD risk. We used this opportunity calculate the 1) odds of CAD and 2) prevalence of CAD among individuals within the top 50th to top 99.9th percentile (relative to the remainder of the population) of RVGRS. We identified a significant trend for increased prevalence of CAD with increasing percentile RVGRS950 ($P_{\text{trend}}=1.69 \times 10^{-15}$) and RVGRS950^{LDLR} ($P_{\text{trend}}=1.26 \times 10^{-14}$). We illustrated these findings in Supplementary Figure 10A and 10B (along with caption) shown below:

“Supplementary Figure 10: Odds and prevalence of CAD across increasing RVGRS950 and RVGRS950^{LDLR}- percentile groupings in the UK Biobank. Odds ratios of CAD were calculated across the top 100 percentile groupings (50th to 99.9th in intervals of 0.05%) and are in reference to the remaining distribution of RVGRS950 or RVGRS950^{LDLR}- (e.g. the 90th percentile refers to odds of CAD for individuals in the top 10% of RVGRS950 or RVGRS950^{LDLR}- relative to the remaining 90% of the population) (A). Shaded regions correspond to the 95% confidence interval of the odds ratio. Odds ratios were adjusted for age, age², sex, and the first 20 principal components of ancestry. CAD prevalence was calculated as the proportion of individuals with CAD across each percentile threshold of RVGRS (B). Significance for change in CAD prevalence across increasing percentile of RVGRS was evaluated using the Cochran-Armitage test for trend. OR indicates odds ratio, RVGRS950 refers to rare variant genetic risk score from the top 950 discovery genes, RVGRS950^{LDLR}- indicates rare variant genetic risk score from the top 950 discovery genes (excluding LDLR), and CAD indicates coronary artery disease.”

We also have noted this in the updated main text and updated supplementary appendix:

“The median RVGRS950^{LDLR}- in early CAD cases corresponded to the 66th percentile of RVGRS950^{LDLR}- for the control distribution, compared to only the 52nd percentile for general CAD (Figure 10). We also identified higher risk and a trend of increased prevalence of CAD (all cases) across increasing RVGRS950 and RVGRS950^{LDLR}- percentile groupings ($P_{trend}=1.69 \times 10^{-15}$ and $P_{trend}=1.26 \times 10^{-14}$, respectively) (Supplementary Figure 10A and 10B).” (Updated main text page 14; lines 322-326)

“Odds of CAD was determined based on 100 percentile groupings of RVGRS (RVGRS950 or RVGRS950^{LDLR}-) in the UK Biobank and was determined for the high percentile group relative to the remainder of the population (e.g. odds of CAD for individuals with ≥ 90 th percentile of RVGRS relative to the remaining 10% of the population) (Supplementary Figure 10A). Odds of CAD was calculated stated as above. Significance for change in CAD prevalence across increasing percentile of RVGRS was evaluated using a two-sided Cochran-Armitage test for trend (Supplementary Figure 10B) with the DescTools R package⁴².” (Updated supplementary appendix page 22; lines 458-464)

10. For individuals with a 2-fold increased odds for CAD, what RVGRS950 percentile score do they have?

We thank the reviewer for this question. We identified 1.5% of the total UK Biobank population (N=45,850) with ≥ 2 -fold increased odds of early CAD (assessed according to the effect estimate of RVGRS950^{LDLR}- on CAD among individuals with disease onset at ≤ 40 years for males and ≤ 45 years for females; updated main text page 14; lines 319-322 and Figure 9), which corresponds to the 98.5th percentile of this score. We have now noted this:

“1 in 68 (1.5%) of participants in the UK Biobank had a RVGRS950^{LDLR}- score (i.e. ≥ 98.5 th percentile), corresponding to a ≥ 2 -fold increase in odds of early disease.” (Updated main text page 14; line 326-328)

Individuals who had 2-fold odds for CAD based on RVGRS950^{LDLR} also had a RVGRS950 score in the 98.5th percentile.

11. Please justify why the RVGRS association model was adjusted to age-squared.

We adjusted our model for age-squared to account for potential non-linear effects of age on CAD, which may result in a better fit to our logistic model. Indeed, the effect of age on chronic, complex diseases such as CAD may increase in a non-linear fashion, wherein individuals >60-65 years may exhibit a risk that is not directly proportional to the risk at younger age groups. Nevertheless, we attained the same effect sizes and strength of association for RVGRS950 without incorporating age-squared as a co-variate for each RVGRS score tested (OR=1.08 per SD; 95% CI, 1.04-1.11; $P=2.2 \times 10^{-5}$).

Minor comments:

1. What does CVGRS stand for? I don't think it is defined in the text.

We thank the reviewer for bringing this to our attention. CVGRS stands for “common variant genetic risk score” and has now been appropriately defined at its first in-text occurrence:

*“Interestingly, the effect conferred through a **common variant genetic risk score (CVGRS)** did not significantly differ across FRS tertiles, with odds ratios of 1.36, 1.26, and 1.30 for low, middle, and high tertiles of FRS, respectively ($P_{interaction}=0.74$) (Supplementary Table 12).”* (Updated main text page 13; line 303)

2. Page 10, line 221 – The authors introduce 5910 discovery MI cases, but is unclear about which cohort these samples are from. Due to the many different cohorts used throughout the manuscript it is important they are clear about which are being used for what analyses.

We thank the reviewer for this comment. We have addressed this by stating that the 5,910 discovery MI cases were from the MIGen consortium:

*“After adjusting gnomAD expected counts with RV-EXCALIBER, we conducted an exome-wide rare variant association on protein-coding genes across 5,910 discovery unrelated MI cases **from the MIGen consortium**, testing all 7,146 genes with an observed count ≥ 1 and an expected count of ≥ 10 (again, to filter out underpowered genes).”* (Updated main text page 10; line 234)

3. Page 10, lines 231-232 – Previously, it was stated that only LDLR reached exome-wide significance, but here the authors state that there were 35 genes identified with a genome-wide significant variant. It is unclear what the difference is between these genes and the identification of LDLR. Further, the authors previously spoke of exome-wide significance and now are saying genome-wide significance, if this is indeed correct more context must be given.

We appreciate the reviewer bringing our attention to this point of confusion.

We agree that the wording made it difficult to appreciate the intended difference between *LDLR* achieving exome-wide significance” and an addition 35 genes with “genome-wide significance”.

Indeed, *LDLR* was the only gene to reach exome-wide significance in our rare variant discovery association analysis on protein-coding genes using MI cases from MIGen and gnomAD controls (main text page 9; lines 223-226). The other 35 genes that are referred to harboured a genome-wide significant variant in the CARDIoGRAMplusC4D consortium. We annotated genes assessed in our rare variant association analysis with corresponding p-values of SNPs located in these same genes. This served the purpose of 1) generating a CAD gene-list (based on robust association from a large CAD GWAS) and 2) using this CAD gene-list to identify whether the test statistic for their **combined** gene-based association signals from our rare variant association analysis (conducted using RV-EXCALIBER) was greater than expected by chance, according to a null distribution of test statistics from random gene sets of the same size (i.e. 35) (main text page 9; lines 237-240). This procedure is also explained in our response to reviewer comment #7.

We have now provided a more comprehensive description which we believe better differentiates genes that harbour a genome-wide significant SNP in the CARDIoGRAMplusC4D consortium from the exome-wide significant gene (i.e. *LDLR*) identified in our discovery rare variant association analysis:

“In fact, a total of 35 (excluding LDLR) of the 7,146 genes assessed in the exome-wide rare variant association were found to harbor a genome-wide significant variant for CAD (i.e. a CAD gene set) in the CARDIoGRAMplusC4D genome-wide association study.” (Updated main text pages 10-11; lines 243-246)

4. Page 12, lines 266-269 – The odds ratios presented for RVGRS950 seemingly decrease with increased FRS. Would the opposite not have been expected? Or were these results as a result of power differences from sample sizes?

The reviewer puts forth an interesting comment. Indeed, we do expect that the odds ratios for CAD will be greatest in the context of both high clinical risk (i.e. proxied by the FRS) and high genetic risk (i.e. proxied by the RVGRS). However, our objective was to instead identify whether individuals with low clinical risk harboured an increased genetic predisposition for CAD relative to those with moderate or high clinical risk burden. We hypothesized that individuals with CAD that harbour a low clinical burden for disease are enriched for rare risk alleles of moderate effect (i.e. RVGRS) that may explain disease onset. We have added this clarification to main text as follows, which acknowledges the hypothesis-generating nature of this finding:

“When evaluating the potential clinical efficacy of the RVGRS, we observed that it demonstrated an increase in predictive effect for individuals at low clinical risk as compared to those with a high clinical burden for developing CAD. We acknowledge this observation to be hypothesis-generating, wherein individuals with low clinical risk may have an enrichment of rare risk alleles of modest effect (i.e. RVGRS) relative to those with moderate or high clinical risk. Future work on gene x environment interactions that focusses on rare variants are required to evaluate the clinical utility of RVGRS, particularly among individuals with low versus high burden of environmental risk.” (Updated main text page 19; lines 439-446)

We have also provided sample size for each FRS tertile in Table 1 (provided above), which provides the distribution of the RVGRS and RVGRS950^{*LDLR*} across different population groups.

We observed that the lowest FRS tertile had the smallest number of individuals with CAD (n=519) relative to n=1,193 CAD individuals in FRS tertile 2 and n=2,131 individuals in FRS tertile 3. Nevertheless, it is expected that only the precision of the effect size (not the effect size itself) will vary by sample size, wherein any given effect estimate would demonstrate stronger statistical significance if the corresponding sample size was higher.

5. Page 16, lines 378-380 – It is mentioned that the CVGRS utilized the CARDIoGRAMplusC4D meta-analysis, which had a case population that was much larger than the MI cohort used in this manuscript. Is there a reason the same cohort was not used for the generation of the RVGRS so that risk estimates could have been compared?

The reviewer brings up an excellent point regarding why the CARDIoGRAMplusC4D consortium was not used for the discovery rare variant association analysis. Briefly, since the CARDIoGRAMplusC4D consortium was purposed to detect common genetic variants via a genotyping array, it is not well suited to detect rare coding variants in the genome. For our work, we require whole-exome sequencing data and such data has not been generated in CARDIoGRAMplusC4D to our knowledge.

6. Page 17, lines 391-392 – The authors state that the addition of RVGRS will have potential clinical utility, but exactly how it will be clinically relevant is not elaborated on.

We have now added a description for pertinent clinical utility of the RVGRS in the concluding paragraph (as referenced by the reviewer):

“The RVGRS conferred disease risk independent of Mendelian effects, clinical risk factors, and CVGRS while also correctly reclassifying ~6% of CAD events, which suggests that addition of RVGRS to CVGRS and Mendelian mutations may have potential clinical utility in terms of case discrimination, but further clinical validation is necessary before establishing a full case for clinical implementation.” (Updated main text page 20; lines 470-474)

Supplementary Appendix:

1. Page 2, lines 57-60 – It is unclear how the variant harmonization was done. As written, it seems as though all samples were sequenced with 5 different technologies and processed with 7 read mappers and 3 variant callers, therefore producing 105 datasets for each sample. Is this in fact the case? Could be rephrased for clarity.

We apologize for the confusion of this description. The combination of sequencing technologies, read mappers, and variant callers were not harmonized in a permuted fashion (i.e. $5 \times 7 \times 3 = 105$ permutations) as was stated in the reviewer’s comment. Rather, there are dedicated read mappers and variant callers that are used for a given sequencing method since they were designed to cater to the specific library preparations (e.g. PCR-based or hybridization-based) and sequencing technology (e.g. paired-end reads vs. single reads). As such, there were a total of 14 datasets for variant harmonization in the referenced paper (Zook *et al.* 2014; DOI: 10.1038/nbt.2835). However, a more recent paper (Zook *et al.* 2019; DOI: 10.1038/s41587-019-0074-6) that

provides a more relevant description of variant harmonization for the updated GIAB consensus sequence versions (i.e. 3.3 and higher, which were used in our study).

We have taken the reviewer's advice and elected to rephrase this section as follows:

“To evaluate the role of population-specific effects to exome-wide mutation load, we determined correction factors using high-confidence sequence variants corresponding to 3 reference GIAB samples of different ethnicities (using methodology developed by Zook et al. 2019)³ : 1) NA12878 (North-western European; NIST ID HG001), 2) NA24631 (East Asian; NIST ID HG005), and 3) NA24385 (Ashkenazi Jewish; NIST ID HG002). Variants called in these reference samples were harmonized across 5 sequencing technologies (Illumina, Complete Genomics, Ion Torrent, SOLiD, and 10X Genomics), which collectively use 5 read mappers and 4 variant callers to generate a “consensus” variant callset that we used as benchmark sequences to evaluate the excess or deficit in mutation load across all protein-coding genes in the gnomAD database.” (Updated supplementary appendix page 1; lines 53-62).

2. Page 11, line 189 – Use of mutation, rather than variant introduces inconsistency in the writing.

This is a keen observation made by the reviewer and we agree that use of “mutation” does introduce inconsistency in this context. We have changed our description from “*Variants classified as exonic were further classified according to 8 mutation types*” to “*Exonic variants were further classified into 8 categories*” (Updated supplementary appendix page 10; line 193).

3. Page 11, lines 195-199 – Does what is written here indicate that the variants were not filtered based on pathogenicity? This section could be written with more clarity.

We thank the reviewer for addressing this point, which could introduce confusion to the reader. We have modified the wording of the section as follows, which we believe will mitigate the ambiguity of the original wording:

*“Due to the depletion of variants in the GIAB consensus sequences after filtering with the intersection of HCC and NIST high-confidence sites, analysis was restricted to all **exonic SNV's and there was no formal pathogenicity criteria applied for variant inclusion.** These criteria were applied to variants present in the GIAB consensus sequences and gnomAD.”* (Updated supplementary appendix page 10; lines 200-203)

4. Page 12, line 221 – Does MAF = 0 include variants that were not present in gnomAD, but were present in GIAB, or were those variants excluded?

We thank the reviewer for this excellent question and believe that they meant to address line 211 and not 221 of the original supplementary appendix. We defined the MAF range for the “rare variant” bin as $0 \leq \text{MAF} \leq 0.01$, which purposefully includes variants with MAF=0 in gnomAD whilst still being present in a GIAB consensus sequence sample. Such variants are particularly important for assessing differences in exome-wide rare mutation load (which forms the basis of the individual correction factor (see equation 5 (shown below) and Figure 3) between GIAB samples (represented by the observed allele count, OAC) and gnomAD (represented by the

expected allele count, EAC) since individual rare variants are very geographically specific, even amongst globally matched ancestries (e.g. Europeans, East Asians, etc.). Therefore, not including such variants will lead to a drastic underestimation in the ratio of GIAB to gnomAD rare variant mutation load (i.e. iCF). It is very important to note that variants with MAF=0 in gnomAD were not ‘missing variants’ (i.e. a variant site that did not pass variant-level quality control procedures). Rather, such variants are those that pass all variant-level quality control metrics in gnomAD, but are not observed in a given gnomAD ancestry due to rarity.

Equation 5 (Updated supplementary appendix page 12; line 305):

$$i\widehat{CF}_i = \frac{\sum_{g=1}^M OAC_{i,g}}{\sum_{g=1}^M EAC_{i,g}}$$

5. Page 12, line 222 – “Batch effects” typically refers to non-biological sources of heterogeneity in the data, but here it seems to refer to the variation introduced due to differing ethnic group. Please clarify.

We thank the reviewer for rightfully pointing this out as the meaning of “batch effects” was ambiguous in our previous description. We have re-written the corresponding section (supplementary appendix section 8E: *Variant frequency filtering for MIGen exomes, UK Biobank 50,000 exomes, and gnomAD*) to more clearly outline frequency filtering while also elaborating on our description of “batch effects”, which does indeed refer to non-biological sequencing/technological bias as the reviewer stated. These changes are as follows:

"E. Variant frequency filtering for MIGen exomes, UK Biobank 50,000 exomes, and gnomAD All qualifying variants were annotated to their corresponding AAF's in the 5 major gnomAD ancestries: 1) Non-Finnish European, 2) African, 3) South Asian, 4) East Asian, and 5) Latin American. Prior to filtering, all AAF's annotated using gnomAD were standardized to the minor allele as stated in Supplementary Section 8C. MIGen and UK Biobank sequence variants were also annotated with cohort-specific AAF's (which were also standardized to the minor allele) in order to mitigate batch effects due to non-biological factors (e.g. technological biases intrinsic to a given sequencing technology or variant caller). All MIGen and UK Biobank sequence variants with a MAF < 0.001 in all 5 major gnomAD ancestries and in their specific cohorts were kept for downstream analysis. The same threshold was applied to variants in gnomAD, but these variants were not annotated with MIGen or UK Biobank cohort-specific frequencies."
(Updated supplementary appendix page 11; lines 225-235)

6. Page 13, lines 231-236 – How were ‘false positive’ and ‘false negative’ variants identified?

Sequencing false negatives (*SFN*) and sequencing false positives (*SFP*) are described as simulation variables in this context. As such, there is no formal “identification” of these parameters as they are not empirically determined. The *SFN* and *SFP* are simulated confounding variables that modulate the probability of detecting a mutation in a given gene (*P*) in the absence (equation 1, provided below) and presence (equation 2, provided below) of a genetic effect (i.e. OR), where $CMAF_{Gene}$ is the unconfounded probability of detecting mutation in a given gene. The purpose of introducing these parameters is to simulate the effect of bias due to differences in sequencing technology (between a local case and public control dataset) on *P*. Moreover, we

show how *SFN* and *SFP* parameters can inform the simulated correction factor (*CF*) (equation 3, provided below), which functions to mitigate this bias (described in supplementary appendix page 13; lines 265-281 and Figure 2A-C).

Equation 1 (Updated supplementary appendix page 12; line 247):

$$P = CMAF_{Gene} PSF (1 - SFN) + (CMAF_{Gene} SFP)$$

Equation 2 (Updated supplementary appendix page 12; line 252):

$$P = CMAF_{Gene} PSF OR (1 - SFN) + (CMAF_{Gene} SFP)$$

Equation 3 (Updated supplementary appendix page 12; line 267):

$$CF = PSF (1 - SFN) + SFP = \frac{P}{CMAF_{Gene}}$$

7. Page 17, line 329 – The iCF abbreviation was used prior to its definition here.

We thank the reviewer for this keen observation and have removed this second definition.

Reviewer #3 (Remarks to the Author):

Summary:

Lali et al. propose a novel method for a gene-based rare-variant burden test with external shared controls. The core idea is based on a new procedure to calibrate the biases in whole-exome sequencing data. With this new burden test, they estimate the gene-based effect sizes for rare variants, construct a weighted rare-variant burden score, and demonstrate its predictive accuracy for CAD.

A gene-based burden test using external controls is not entirely new (notably, TRAPD and ProxECAT). However, these existing methods rely on the patterns of non-functional variants for calibration procedures, thus may still miss subtle biases in functional variants. The proposed method attempt to calibrate study-specific biases using the functional variants in cohort-specific control samples. This is a novel approach that can potentially produce better calibration. I think this is a great idea of interest to the community if the following concerns are addressed.

We sincerely thank the reviewer for their interest in our work and acknowledgement of its applicability to the genetics community. We believe that the changes made in accordance with the reviewer's comments have made our work much stronger and hope that our thorough responses to the reviewer's comments address all concerns.

Main comments:

Lines 61-63 "there is currently no method to calibrate rare variant burden between test samples and public databases," lines 293-295 "methods have been developed to address the genetic contributors to Mendelian disease using external control databases," and lines 302-304 "current methodologies that leverage gnomAD as controls have been tailored to demarcate variant and gene-based associations for Mendelian disorders." I am not

convinced that existing methods (e.g. TRAPD and ProxECAT) are limited to the application for Mendelian disorders only. These methods are developed as burden tests for complex traits with locus heterogeneity. ProxECAT should be cited (PMID: 30325923).

The reviewer correctly states that the traits assessed in both the TRAPD and ProxECAT papers are complex and exhibit locus heterogeneity. We have altered the instances referenced by the reviewer where we make mistakenly mention of TRAPD/ProxECAT assessing Mendelian disease alone. We have provided the changes below:

*“While methods have been developed to address the genetic contributors to **complex** disease using external control databases...”* (Updated main text page 14; lines 334-325)

*“Current methodologies that leverage gnomAD as controls have been tailored to demarcate variant and gene-based associations **by calibrating functional variant inclusion and counts using benign variants**^{15,16}. There is currently no framework that incorporates global and granular adjustments to gnomAD **using functional variants** in order to detect rare variant burden associated with complex diseases. Existing methodologies utilizing gnomAD as controls, such as TRAPD **and ProxECAT**, calibrate test statistics based on benign mutation rate by altering filtering criteria for functional variant inclusion^{15,16}, **but this approach cannot account for technical, population genetic, or gene-based biases inherent to functional variants.**”* (Updated main text page 15; lines 343-350)

We have also appropriately referenced ProxECAT (reference 16) in the updated descriptions.

Figure 1: The simulations here seem to have considered only one bias each time while setting the other biases to 0. For example, I think Figs 1A and D simulated a range of sequencing false-negative rates under no sequencing false positive or population difference, and so on. If this is the case, this is too unrealistic. The unbiasedness or conservativeness of estimated correction factors cannot be concluded under such limited simulation conditions. Simulations need to be done in presence of all three biases (or at least two: sequencing false negatives and false positives).

Please note that the referenced Figure 1 is now Figure 2 in the updated figure legends. The reviewer correctly asserts that simulating the impact of the correction factor on effect size for a single bias at a time is unrealistic. This is especially true when comparing internal cases to external controls, where there are likely to be multiple biases that can influence the estimated effect size of gene-based associations. We have now added a figure (Supplementary Figure 1) which showcases the collective effect of each bias (Sequencing False Negatives, Sequencing False Positives, and Population-Specific Factors) on the estimated odds ratio, both with and without the correction factor. Even in the presence of all three biases, we show that the correction factor is able to robustly calibrate the estimated odds ratio to the true odds ratio. Indeed, the CF still remains conservative at high Sequencing False Positive rates (Supplementary Figure 1C and 1F) (as was observed when the single biases were assessed separately), but adequately corrects for strong Sequencing False Negative and Population Specific bias.

We have provided our figure and caption (Supplementary Figure 1) and added the corresponding results to our updated main text below:

“Simulated effects of combined SFN rate, SFP rate, and PSF on estimated OR. Probability of mutation (P) within a gene was calculated as a function of the cumulative minor allele frequency for a single gene ($CMAF_{Gene}$: set at 0.05), the true effect size of association (OR), and the 3 major association biases (SFN, SFP, and PSF) according to equation 2. The simulations kept the true OR fixed at 1.3 (black dashed lines) while varying SFN rate (0 to 1), SFP rate (0.1, 0.2, and 0.3), and PSF (0.3 and 1.7). An estimated OR was calculated for each P (corresponding to a unique combination of the 3 association biases) and plotted as a function of the combined biases (A-F; red lines). CF values were calculated as the ratio of the P for each estimated OR and the original $CMAF_{Gene}$ according to equation 3. Each CF was then used to adjust the $CMAF_{Gene}$ to calculate an adjusted P (P^*) according to equation 4. Using P^* , an adjusted estimated OR was plotted as a function of the combined biases (A-F; blue lines). OR indicates odds ratio, CF indicates correction factor, SFN indicates sequencing false negatives, SFP indicates sequencing false positives, PSF indicates population-specific factor.”

We have added this observation to our updated main text as well:

“We also observed that the CF robustly calibrates the estimated OR in the presence of all biases simultaneously (i.e. SFN, SFP, and PSF) and adheres to the same patterns as observed with single bias simulations (Supplementary Figure 1). That is, the CF adequately calibrates the estimated OR to the true OR in the presence of strong SFN and PSF bias, while providing a slightly conservative correction at high SFP rates (Supplementary Figure 1C and 1F).”
(Updated main text page 5; lines 118-122)

Lines 142-144 (Figure 2C): NA12878 is not a "Central European." It is a CEU sample (CEPH from Utah, with Northern and Western European ancestry). In principle, it should show very close proximity to gnomAD Non-Finnish European ancestry. The extensive deviation observed in NA12878 across all frequency bins is very concerning. This sample should be very representative of gnomAD Europeans. What is an explanation for this?

Please note that the referenced Figure 2C is now Figure 3C in the updated figure legends. The reviewer correctly states that the NA12878 GIAB sample is of Northern & Western European ancestry. We apologize for mistakenly classifying this sample as “Central European” and have changed this to “North-western European” in all instances throughout the updated main text, updated supplementary appendix, and updated figure legends. These were the instances:

*“When comparing the Ashkenazi Jewish and **North-western** European consensus sequences with the non-Finnish European (NFE) population in gnomAD, we observed significant deviations in iCF for rare variants ($0 < MAF \leq 0.01$) (Ashkenazi: $iCF = 1.37$; 95% CI, 1.22 to 1.55; $p < 0.05$ and **North-western** European: $iCF = 1.20$; 95% CI, 1.07 to 1.36; $p < 0.05$), where a $iCF=1$ represents equal mutational loads.”* (Updated main text page 6; lines 143-148)

*“Indeed, we show that the mutation burden in European consensus sequences (i.e. **North-western** European or Ashkenazi) deviate substantially from the Non-Finnish Europeans in gnomAD and is more closely matched across common variants (Figure 3A and C).”* (Updated main text page 16; lines 379-381)

*“To evaluate the role of population-specific effects to exome-wide mutation load, we determined correction factors using high-confidence sequence variants corresponding to 3 reference GIAB samples of different ethnicities (using methodology developed by Zook et al. 2020)³ : 1) NA12878 (**North-western** European; NIST ID HG001), 2) NA24631 (East Asian; NIST ID HG005), and 3) NA24385 (Ashkenazi Jewish; NIST ID HG002).”* (Updated supplementary appendix page 1; lines 54-58)

*“Each variant within the GIAB consensus sequence was annotated with its corresponding allele frequency in a closely related population within gnomAD. Specifically, the **North-western** European GIAB sample (NA12878) was annotated with the Non-Finnish European (NFE) population allele frequencies...”* (Updated supplementary appendix page 10; line 205-208)

*“Specifically, the EAC for comparison with the **North-western** European GIAB sample (NA12878) was generated using the Non-Finnish European AAF’s in gnomAD (see **Supplementary Table 3** for all GIAB to gnomAD ethnicity matches).”* (Updated supplementary appendix page 15; line 313-315)

These changes are also made in Supplementary Tables 3 and 4 in updated supplementary appendix:

Supplementary Table 3: Ethnicities in gnomAD used to generate the EAC for a given GIAB sample.

ID of GIAB sample	Ethnicity of GIAB sample	Ethnicity in gnomAD used to determine the EAC
NA12878	Central North-western European	Non-Finnish European
NA24631	East Asian	East Asian
NA24385	Ashkenazi Jewish	Non-Finnish European
		Ashkenazi Jewish

Supplementary Table 4: iCF values for every GIAB vs. gnomAD ethnicity across 4 allele frequency bins.

AF bin	GIAB ethnicity	gnomAD population	iCF	95% CI
0-0.01	Central European North-western European	Non-Finnish European	0.60 1.20	0.54-0.68 1.07-1.36
	East Asian	East Asian	0.93	0.84-1.04
	Ashkenazi	Ashkenazi	0.93	0.84-1.04
	Ashkenazi	Non-Finnish European	1.37	1.22-1.55
0.01-0.05	Central European North-western European	Non-Finnish European	0.59 0.93	0.55-0.65 0.86-1.01
	East Asian	East Asian	1.00	0.92-1.11
	Ashkenazi	Ashkenazi	0.93	0.86-1.01
	Ashkenazi	Non-Finnish European	0.98	0.91-1.08
0.05-0.25	Central European North-western European	Non-Finnish European	0.80 1.03	0.78-0.83 0.99-1.06
	East Asian	East Asian	0.98	0.95-1.01
	Ashkenazi	Ashkenazi	0.99	0.96-1.02
	Ashkenazi	Non-Finnish European	0.98	0.95-1.01
0.25-0.50	Central European North-western European	Non-Finnish European	0.93 1.00	0.92-0.95 0.99-1.02
	East Asian	East Asian	0.97	0.96-0.99
	Ashkenazi	Ashkenazi	1.00	0.99-1.02
	Ashkenazi	Non-Finnish European	1.02	1.01-1.04

These changes have also been made on Figure 3C and Supplementary Figure 3C in updated figure legends:

Figure 3C:

Supplementary Figure 3C:

The reviewer also brings up an excellent observation that the NA12878 GIAB sample should be well-matched with non-Finnish Europeans in gnomAD, since this gnomAD group is primarily comprised of individuals from Northern & Western Europe. Upon investigation, we realized that we were using an old version download for the NA12878 sequencing file (version 3.2.2), which seemed to have a systematic depletion of PASS variant calls. As such, the mutation load of this

sample was depleted across all allele frequency bins in the old Figure. We downloaded the more recent NA12878 GIAB sequencing file (version 3.3) and re-calculated the mutation load relative to gnomAD non-Finnish Europeans across all 4 allele frequency bins. As the reviewer expected, the mutation load for the NA12878 GIAB sample is now well-matched with non-Finnish Europeans in gnomAD for the more common allele frequency bins, but still maintains the most deviation from expectation at the rarest bin (see Figure 3C above). This observation still adheres to our expectation that rare variants demonstrate strong geographical specificity resulting in pronounced deviance in mutation loads even among ancestral groups that are seemingly well-matched.

We have noted the new iCF values for the NA12878 GIAB sample in all pertinent instances throughout the in the updated main text, updated supplementary appendix, and updated figure legends:

“When comparing the Ashkenazi Jewish and North-western European consensus sequences with the non-Finnish European (NFE) population in gnomAD, we observed significant deviations in iCF for rare variants ($0 < \text{MAF} \leq 0.01$) (Ashkenazi: $i\text{CF} = 1.37$; 95% CI, 1.22 to 1.55; $p < 0.05$ and North-western European: $i\text{CF} = 1.20$; 95% CI, 1.07 to 1.36; $p < 0.05$), where a $i\text{CF}=1$ represents equal mutational loads.” (Updated main text page 6; lines 143-148)

Supplementary Table 4 has been changed to reflect this change (see above).

These new iCF values are also reflected in Figure 3C (see above).

Lines 165-167 (Figure 3A): I am skeptical that "exome-wide rare variant mutation load" is a primary factor driving the observed inter-cohort differences in mean iCF. Each individual has a different mutational load, but the mean difference of mutational load is < 3% across European ancestry (PMID: 25581429, see Fig 1).

Please note that the referenced Figure 3A is now Figure 4A in the updated figure legends. The reviewer provides an excellent topic point with this comment. In the paper referenced (*Do et al.* 2018, DOI: [10.1038/ng.3186](https://doi.org/10.1038/ng.3186)), the authors compare the relative mutation loads for all (rare and common) nonsynonymous single nucleotide variants (SNVs) in a pairwise manner among European and African populations. Indeed, the relative mean difference in mutation load of nonsynonymous SNVs between distinct European populations (e.g. Toscani in Italia versus Spanish) is < 3%, demonstrating non-differential selection pressure. However, the variants used to inform the relative enrichment/depletion of mutation loads in our work (i.e. between cohorts in the MIGen consortium) differ from those used in the referenced paper in three respects: 1) nonsynonymous SNVs were very rare ($\text{MAF} < 0.001$), 2) rare nonsynonymous SNVs were enriched for deleterious mutations and must have either been classified as pathogenic according to the Mendelian Clinically Applicable Pathogenicity (M-CAP) score or led to loss-of-function (i.e. frameshift indels, splice site mutations, or nonsense mutations), and 3) the relative mutation loads are calculated based on variants called using heterogeneous sequencing technologies (i.e. an internal case dataset vs. an external control database). Points 1-2 relate to differences in population genetics associated with distinct variant types, whereas the 3rd point considers differences in mutation load attributable to non-biological mechanisms (i.e. technological variation).

Point 1: While the vast majority of nonsynonymous SNVs are rare (see Nelson *et al.* 2012; DOI: [10.1126/science.1217876](https://doi.org/10.1126/science.1217876)), those that are common are heavily enriched for having benign effects, since such variants are less likely to be subject to purifying selection. Therefore, these higher frequency variants are not as geographically contained as variants that are selected against (which are kept at rare frequencies). In an interim analysis conducted using the unselected UK Biobank exome sequencing cohort, we identified that 4% of all nonsynonymous SNVs were common (i.e. $MAF \geq 0.001$). As such, up to 4% of nonsynonymous SNVs that are likely to be more geographically ubiquitous were not used for our iCF calculations among MIGen controls (Figure 4A), which partially explains why we did not observe iCF values closer to 1 relative to gnomAD non-Finnish Europeans across all cohorts.

Point 2: This point is largely an extension of point 1. Whereas some rare nonsynonymous SNVs might be at low frequency due to them being de novo variants, others will be subject to purifying selection due to them having a detrimental effect on fitness (i.e. pathogenic variants). In addition to focusing on very rare variants, we limited our mutation load calculations to those exhibiting highly functional effects, characterized by nonsynonymous SNVs with an M-CAP score >0.025 and variants that substantially perturb protein function (i.e. loss of function variants). Since there is not much overlap between de novo and functional variants ($< 1\%$, see Kessler *et al.* 2020; DOI: [10.1073/pnas.1902766117](https://doi.org/10.1073/pnas.1902766117)), we expect that the vast majority of the functional variants observed in our work have been selected against. Indeed, it has been shown that the extent of rare allele sharing is highest among individual ancestries (1000 Genomes Consortium 2015; DOI: [10.1038/nature15393](https://doi.org/10.1038/nature15393)) but begins to decline rapidly as a function of geographical distance. Moreover, the decline in rare allele sharing is further amplified for rarer allele bins at any given geographical distance (Nelson *et al.* 2015; DOI: [10.1126/science.1217876](https://doi.org/10.1126/science.1217876)) and for closely related continental populations (Gravel *et al.* 2011; DOI: [10.1073/pnas.1019276108](https://doi.org/10.1073/pnas.1019276108)). Indeed, when enriching for highly functional nonsynonymous SNVs, we observed that the MAF spectrum to be significantly lower as compared to such variants without functional thresholds. As such, the extent of rare allele sharing is expected to be significantly lower in our work, which contributes to the inter-cohort differences in exome-wide mutation load shown in Figure 4A. In contrast, no functional thresholds were used to inform the relative difference in nonsynonymous SNV mutation load for Figure 1 in the referenced paper, which results in values closer to 1 due to higher MAF spectrum and greater allele sharing between continental populations. This trend can also be observed in Table 1 of the references paper, where relative mutation loads are more discrepant between populations when nonsynonymous SNVs are functional (via a “probably damaging” Polyphen 2 score in this case).

Point 3: The gnomAD exomes are an amalgamation of several cohorts which were sequenced using various sequencing chemistries and sequencing platforms. As such, the systematic biases may impact the ability to detect certain variants with high accuracy relative to the sequencing chemistry/platform used in the internal case datasets. The iCF values calculated in Figure 4A are meant to capture systematic biases due to both population genetic factors (i.e. points 1-2) and sequencing technology (point 3). Indeed, the MIGen exomes were sequenced using either the SureSelect Human all exon v2 kit (Italian Atherosclerosis Thrombosis and Vascular Biology, Ottawa Heart Study, Precocious Coronary Artery Disease Study, and Registre Gironi del Cor) or ICE capture reagent (Malmö Diet and Cancer, University of Lubeck, BioImage, and German

Heart Centre in Munich) (see Supplemental Table 1). Therefore, the differences in sequencing technology are capable of contributing to inter-cohort differences in exome-wide mutation load in this case. The relative mutation loads for nonsynonymous SNVs between European populations in the referenced paper (Table 1 and Figure 1) are not influenced by heterogeneity in sequencing chemistry/platform, since they were processed using identical sequencing workflows. Therefore, the mutation loads in the referenced paper will necessarily be closer to 1 as compared to those identified in our work.

Lambda_med comparison against TRAPD (lines 206-207, 315-316 and Figure 4): It is well-known that lambda_med is not a good benchmark statistic to evaluate calibration for rare-variant burden tests because of inadequate continuity correction of Fisher's Exact Test at P=1 tail. Because of this, lambda_95 is a more adequate stat here, or to use lambda_med, underpowered genes with low numbers of rare functional variants should be excluded. It seems to me that lambda_95 is not much different between TRAPD and RV-EXCALIBER. Rather, I am concerned about the potential over-correction of RV-EXCALIBER at the significant tail. RV-EXCALIBER base is over-correcting (Fig S5). Including iCF and gCF on top of RV-EXCALIBER base helps, but there seem to be lingering over-correction issues at the significant tail, thereby potentially loss of power (Fig S6). I am particularly concerned because binning-based calibration techniques, which underlies RV-EXCALIBER, tend to err at the bin of extreme tails in general.

Please note the following changes in the referenced figures in the updated figure legends: Figure 4 is now Figure 5, Supplementary figure 5 is now Supplementary Figure 6, Supplementary Figure 6 is now Supplementary Figure 7. We thank the reviewer for this insightful comment. It is correct that lambda_med may underestimate calibration of test statistics to the tail of the null distribution when conducting rare variant burden tests with a Fisher's Exact Test. However, as the reviewer stated, the limitation of the continuity correction can be overcome by focussing on genes that harbour sufficient counts of functional rare variants, which are well powered to detect an association signal (should one exist). We therefore implemented minimum per-gene allele counts for observed counts (informed by the additive rare pathogenic allele count in the internal case dataset) and expected counts (informed from the cumulative minor allele count of rare pathogenic count alleles in gnomAD) of ≥ 1 and ≥ 10 , respectively (see Supplementary Figure 6 caption). We impose a higher allele count threshold for gnomAD due to the high number of contributing alleles (~110,000 alleles for non-Finnish Europeans in gnomAD) that are queried to detect a given functional variant. A greater number of contributing alleles will result in more accurate allele count estimates and enable better discrimination between well-powered and underpowered genes. Moreover, our original hypothesis was that many genes that are truly associated with a given trait will confer risk through modest (or even weak) effects. Since such effects will not necessarily yield strong rare variant association signals, having a well-calibrated test throughout the spectrum of mutation load (i.e. genes with lower and higher allele counts) is a desirable property, especially when using these genes to inform gene scores on basis of their P-values.

Indeed, a similar approach was used when assessing calibration of test statistics generated using ProxECAT and ProxECAT-weighted, where the authors instituted a minor allele count cut-off of ≥ 5 per gene while using a minor allele frequency threshold of <0.001 (the same used in our

work) (see Figure 3 in Hendriks *et al.* 2018; DOI: [10.1371/journal.pgen.1007591](https://doi.org/10.1371/journal.pgen.1007591)). Using these thresholds, the authors found the lambda_med to equal 1.069 for ProxECAT-weighted, which is nearly the same percent deviation from 1 as observed in our work (lambda_med = 0.93; Supplementary Figure 7). It is important to note that Hendricks *et al.* used a likelihood ratio test, where test statistics were approximated by the chi-squared distribution, which is also subject to conservative continuity correction at the P=1 tail.

Lastly, it should be noted that the slight deflation observed in the tails of test statistic were derived from a disease-free control population, where there not expected to be strong associations at the tail. This is in contrast to our QQ-plot of MIGen cases vs. gnomAD non-Finnish Europeans (Figure 6), where the amount of deflation at the tail is markedly less upon visual inspection.

Nevertheless, we have now acknowledged that there is conservative correction of gene-based test statistics at the significant tail and noted it as a limitation to our method. We recommend that users thoroughly query top association signals to identify if any genes are strong biological candidates for the trait being assessed:

“A noteworthy limitation to RV-EXCALIBER is a tendency for deflation of test statistics at the significant tail (Supplementary Figure 6 and 7), which was observed using healthy control data. Although we observed less deflation at the significant tail for cases (Figure 6), we recommend that users thorough query top association signals to identify if any genes are strong biological candidates for the trait being assessed.” (Updated main text page 17; lines 397-402)

Benchmark comparison against TRAPD. The authors need to show a power comparison of TRAPD and RV-EXCALIBER. This could be done in simulation or real data (ideally both). In simulations, the authors should allow realistic combinations of all biases at the same time.

The reviewer presents a noteworthy comment which outlines the need to identify the degree of bias corrected for by either RV-EXCALIBER or TRAPD. Due to the drastic methodological differences between these methods, a direct “head-to-head” comparison of the two methodologies in mitigating bias is not possible under simulation conditions. This is because the method employed by TRAPD works in a manner that is agnostic of biases that may afflict individual genes. Specifically, TRAPD institutes a variant-level quality by depth cut-off (according to benign variants) to inform inclusion of pathogenic variants across the entire exome, irrespective of individual gene bias. As such, TRAPD will calibrate certain genes well, while still leading to inflation or deflation (with respect to gnomAD) in other genes. Moreover, the authors of TRAPD presented no simulations of their method (Guo *et al.* 2018; DOI: [10.1016/j.ajhg.2018.08.016](https://doi.org/10.1016/j.ajhg.2018.08.016)). Nevertheless, this presents an excellent opportunity to compare the bias accounted for by RV-EXCALBER and TRAPD using real data.

We generated dispersion plots which illustrate the delta mutation count (i.e. the per-gene difference in observed and expected count) as a function of expected count for all genes with an observed and expected count ≥ 1 according to both methods (Supplementary Figure 9A and B in the updated figure legends). We conducted this on the same 3,352 healthy controls that we used in our calibration analysis between RV-EXCALIBER and TRAPD (Figure 5). Due to the fact

that these are healthy controls, we expect that the mean delta count (i.e. mean difference between observed and expected count) per gene should equal 0 across the entire spectrum of expected counts (i.e. from genes containing very few mutations to genes containing many mutations). We show that RV-EXCALIBER does indeed result in a mean difference of 0 across all genes, while TRAPD demonstrates inflation and deflation for genes with low and high expected counts, respectively, which is indicative of residual bias. Specifically, we show that this bias is particularly pronounced among genes with high expected counts, which have the best power to detect rare pathogenic alleles associations. As such, the bias observed for these genes is not due to lack of power to detect such mutations.

We have shown this in the figure below (along with caption), which is now Supplementary Figure 9, and have added additions to the updated main text to reflect this result:

“Supplementary Figure 9: Dispersion of delta counts generated using RV-EXCALIBER or TRAPD across the spectrum of expected allele count for 3,352 healthy control participants in MIGen. Delta mutation counts were calculated according to equations 8 and 9 as the per-gene difference between observed allele count (OAC) and expected allele count (EAC) across 3,352 healthy control participants from MIGen and gnomAD, respectively. A total of 10,788 genes which had observed and expected counts ≥ 1 in both methods were included and were ascertained for rare pathogenic alleles. EAC from RV-EXCALIBER are adjusted by the both the iCF and gCF according to equation 8, while the OAC and EAC for TRAPD were obtained from percentile cut-offs of variant-level quality-by-depth scores for the same 3,352 healthy control participants from MIGen and gnomAD (see main text results). The EAC for either method was log₁₀ transformed and assigned to quintiles. A regression line representing the mean percent difference was then plotted for both RV-EXCALIBER (blue) and TRAPD (red). Horizontal dashed lines indicate zero percent difference between OAC and EAC.”

“In order to further benchmark RV-EXCALIBER and TRAPD, we assessed presence of residual bias by measuring the dispersion of the per-gene difference in observed and expected counts (i.e. delta count) as a function of expected count for all genes with observed and expected counts ≥ 1 in the same 3,352 healthy control exomes in MIGen. Across 10,788 genes that met this count threshold for both methods, we identified an increasing deflation bias in percent delta count for

TRAPD as gene-based expected counts increased. In contrast, RV-EXCALIBER remained unbiased, having a mean percent delta near zero across all quintiles of expected count (Supplementary Figure 9).” (Updated main text page 9; lines 212-219)

“Moreover, TRAPD demonstrated a deflation bias in percent difference between observed and expected counts, especially among genes with high expected counts for which bias cannot be attributed to lack of power to detect rare variants associations (Supplementary Figure 9B). However, RV-EXCALIBER had an unbiased mean percent difference across all expected counts (Supplementary Figure 9A), which further warrants the necessity to institute both individual and gene-level correction (i.e. iCF and gCF) to mitigate residual bias that cannot be accounted for when assuming that this bias is the same across all genes, which is the case for TRAPD.” (Updated main text page 16; lines 359-366)

A potential strength of the new method may be the simplicity of application to heterogeneous cohorts. Existing methods can be applied to such a dataset, but they need to be run in each cohort separately, and then meta-analysis should be done across cohorts while accounting for overlapping shared controls.

We thank the reviewer for this acknowledgment of the advantage of RV-EXCALIBER relative to other methodologies that leverage external datasets as controls. Indeed, the extensive individual and gene-level adjustments applied by RV-EXCALIBER provide means for sample pooling as the biases across heterogeneous cohorts are mitigated relative to gnomAD.

Rare-variant risk score. It is very exciting to see the significant predictive power of rare variant burden score. This is great. How much improvement is there in predictive power, compared to other approaches, e.g. a rare-variant burden using TRAPD-derived ORs, or any other approaches?

We thank the reviewer for their appreciation of the rare variant genetic risk score (RVGRS). The reviewer brings up an excellent point in regard to whether the RVGRS will have different predictive power for CAD when the discovery odds ratios (ORs) are derived from other methods, such as TRAPD. We have now included a supplementary table (Supplementary table 11) which details the OR and corresponding P-value of RVGRS calculated from both RV-EXCALIBER and TRAPD-derived ORs. We calculated 300 RVGRS scores for each method, which were constructed according to the top 10 to top 3000 discovery genes (in intervals of 10). We show that the RVGRS generated using RV-EXCALIBER ORs demonstrate both higher effect size (i.e. predictive power) and improved precision for CAD compared to RVGRS calculated from TRAPD ORs. In fact, RVGRS950 (i.e. the RVGRS with the strongest CAD association according to RV-EXCALIBER-derived ORs), which was associated with a 1.08-fold (95% CI, 1.40-1.11; $P=2.1 \times 10^{-5}$) increased odds of CAD per SD, demonstrated a >20% increase in predictive power relative to the RVGRS with the strongest CAD association calculated from TRAPD-derived ORs, which was found to confer a 1.06-fold (95% CI, 1.03-1.10; $P=4.4 \times 10^{-4}$) increased odds of CAD per SD in the UK Biobank (see Supplementary Table 11).

“We also identified that RVGRS generated using gene-based odds ratios derived from RV-EXCALIBER resulted in higher effect estimates and improved precision for CAD association compared to RVGRS calculated using the gene-based odds ratios from TRAPD in the UK

Biobank. In fact, the effect size of RV-EXCALIBER’s RVGRS950, which conferred a 1.08-fold (95% CI, 1.40-1.11; P=2.1x10⁻⁵) increased odds of CAD per SD, represents a >20% increase relative to the predictive power of the RVGRS with the strongest CAD association according to TRAPD-derived odds ratios, which was found to confer a 1.06-fold (95% CI, 1.03-1.10; P=4.4x10⁻⁴) increased odds of CAD per SD (Supplementary Table 11).” (Updated main text pages 12-13; lines 285-293)

Supplementary Table 11: Effect estimates of 300 RVGRS (calculated using gene-based odds ratios from RV-EXCALIBER or TRAPD) on CAD in the UK Biobank.

Number of discovery genes	RV-EXCALIBER RVGRS OR (95% CI)	RV-EXCALIBER RVGRS P-value	TRAPD RVGRS OR (95% CI)	TRAPD RVGRS P-value	Relative difference in OR (%) ^a
10	1.032 (0.998-1.065)	0.0592	1.024 (0.992-1.057)	0.138	33.33
20	1.048 (1.015-1.081)	0.00362	1.025 (0.992-1.059)	0.128	92.00
30	1.051 (1.018-1.085)	0.00212	1.027 (0.994-1.060)	0.11	88.89
40	1.048 (1.015-1.083)	0.00423	1.035 (1.002-1.069)	0.0352	37.14
50	1.042 (1.008-1.076)	0.0142	1.037 (1.003-1.071)	0.0293	13.51
60	1.047 (1.013-1.081)	0.00614	1.033 (0.999-1.067)	0.0525	42.42
70	1.051 (1.017-1.086)	0.00266	1.035 (1.001-1.069)	0.0414	45.71
80	1.058 (1.024-1.093)	6.87x10 ⁻⁴	1.030 (0.996-1.064)	0.077	93.33
90	1.062 (1.027-1.097)	3.36x10 ⁻⁴	1.023 (0.989-1.057)	0.179	169.57
100	1.053 (1.018-1.088)	0.00217	1.025 (0.991-1.059)	0.147	112.00
110	1.050 (1.016-1.085)	0.00346	1.031 (0.997-1.065)	0.0721	61.29
120	1.053 (1.019-1.088)	0.00215	1.035 (1.001-1.069)	0.0435	51.43
130	1.052 (1.017-1.087)	0.00274	1.035 (1.001-1.069)	0.0425	48.57
140	1.058 (1.024-1.093)	7.49x10 ⁻⁴	1.037 (1.003-1.071)	0.0317	56.76
150	1.055 (1.020-1.090)	0.00162	1.041 (1.007-1.076)	0.0159	34.15
160	1.056 (1.022-1.092)	0.00111	1.040 (1.006-1.074)	0.0204	40.00
170	1.060 (1.025-1.095)	5.93x10 ⁻⁴	1.035 (1.001-1.070)	0.0408	71.43
180	1.058 (1.024-1.094)	7.77x10 ⁻⁴	1.037 (1.003-1.072)	0.0294	56.76
190	1.060 (1.025-1.095)	5.74x10 ⁻⁴	1.036 (1.002-1.071)	0.0364	66.67
200	1.057 (1.022-1.093)	0.00103	1.033 (0.999-1.068)	0.0524	72.73
210	1.059 (1.024-1.094)	7.31x10 ⁻⁴	1.034 (1.000-1.069)	0.0465	73.53
220	1.057 (1.022-1.092)	0.00108	1.034 (1.000-1.069)	0.0461	67.65
230	1.055 (1.020-1.090)	0.00172	1.034 (1.000-1.069)	0.0502	61.76
240	1.058 (1.023-1.094)	8.52x10 ⁻⁴	1.037 (1.003-1.072)	0.0305	56.76
250	1.051 (1.016-1.086)	0.00354	1.039 (1.005-1.074)	0.0226	30.77
260	1.051 (1.016-1.086)	0.00364	1.042 (1.007-1.077)	0.0166	21.43
270	1.051 (1.016-1.086)	0.00354	1.041 (1.007-1.076)	0.0171	24.39
280	1.054 (1.019-1.089)	0.00205	1.045 (1.011-1.080)	0.0091	20.00

290	1.061 (1.026-1.097)	4.94x10 ⁻⁴	1.046 (1.012-1.082)	0.0076	32.61
300	1.058 (1.023-1.093)	0.00101	1.044 (1.009-1.079)	0.0121	31.82
310	1.059 (1.024-1.095)	7.02x10 ⁻⁴	1.047 (1.012-1.082)	0.00693	25.53
320	1.056 (1.021-1.092)	0.00134	1.047 (1.012-1.082)	0.00732	19.15
330	1.056 (1.022-1.092)	0.00125	1.049 (1.015-1.085)	0.00476	14.29
340	1.059 (1.024-1.094)	8.16x10 ⁻⁴	1.046 (1.012-1.082)	0.0078	28.26
350	1.059 (1.024-1.094)	8.12x10 ⁻⁴	1.047 (1.012-1.082)	0.00724	25.53
360	1.062 (1.027-1.098)	3.75x10 ⁻⁴	1.052 (1.017-1.087)	0.00289	19.23
370	1.058 (1.023-1.094)	8.65x10 ⁻⁴	1.049 (1.015-1.085)	0.00478	18.37
380	1.060 (1.025-1.095)	6.61x10 ⁻⁴	1.051 (1.016-1.086)	0.00359	17.65
390	1.062 (1.027-1.098)	4.29x10 ⁻⁴	1.050 (1.016-1.086)	0.00397	24.00
400	1.063 (1.028-1.099)	3.51x10 ⁻⁴	1.050 (1.015-1.085)	0.00457	26.00
410	1.060 (1.025-1.096)	6.56x10 ⁻⁴	1.049 (1.014-1.084)	0.00531	22.45
420	1.058 (1.023-1.094)	9.63x10 ⁻⁴	1.047 (1.013-1.083)	0.00664	23.40
430	1.059 (1.024-1.095)	7.04x10 ⁻⁴	1.047 (1.012-1.082)	0.00718	25.53
440	1.061 (1.026-1.097)	4.95x10 ⁻⁴	1.050 (1.015-1.086)	0.00419	22.00
450	1.062 (1.027-1.097)	4.51x10 ⁻⁴	1.050 (1.016-1.086)	0.00401	24.00
460	1.062 (1.027-1.098)	4.06x10 ⁻⁴	1.051 (1.016-1.087)	0.00357	21.57
470	1.059 (1.024-1.095)	7.32x10 ⁻⁴	1.052 (1.018-1.088)	0.0027	13.46
480	1.062 (1.027-1.098)	4.48x10 ⁻⁴	1.051 (1.017-1.087)	0.00328	21.57
490	1.063 (1.028-1.099)	3.06x10 ⁻⁴	1.051 (1.016-1.087)	0.00347	23.53
500	1.063 (1.028-1.099)	3.55x10 ⁻⁴	1.051 (1.016-1.087)	0.00364	23.53
510	1.067 (1.032-1.103)	1.33x10 ⁻⁴	1.050 (1.015-1.085)	0.00438	34.00
520	1.067 (1.032-1.103)	1.30x10 ⁻⁴	1.051 (1.016-1.087)	0.00352	31.37
530	1.066 (1.031-1.102)	1.86x10 ⁻⁴	1.053 (1.018-1.088)	0.00258	24.53
540	1.061 (1.026-1.097)	5.11x10 ⁻⁴	1.055 (1.020-1.091)	0.00173	10.91
550	1.062 (1.027-1.098)	4.12x10 ⁻⁴	1.055 (1.020-1.091)	0.00173	12.73
560	1.066 (1.031-1.102)	1.73x10 ⁻⁴	1.055 (1.020-1.090)	0.00183	20.00
570	1.063 (1.028-1.099)	3.40x10 ⁻⁴	1.053 (1.018-1.088)	0.00261	18.87
580	1.058 (1.024-1.094)	8.63x10 ⁻⁴	1.050 (1.016-1.086)	0.00402	16.00
590	1.059 (1.024-1.095)	7.70x10 ⁻⁴	1.051 (1.016-1.087)	0.00351	15.69
600	1.055 (1.021-1.091)	0.00159	1.050 (1.016-1.086)	0.00404	10.00
610	1.054 (1.020-1.090)	0.00192	1.050 (1.015-1.085)	0.00451	8.00
620	1.055 (1.020-1.091)	0.00171	1.050 (1.015-1.085)	0.0046	10.00
630	1.059 (1.024-1.095)	8.35x10 ⁻⁴	1.051 (1.017-1.087)	0.00346	15.69
640	1.062 (1.027-1.098)	4.26x10 ⁻⁴	1.051 (1.016-1.087)	0.00357	21.57
650	1.061 (1.026-1.097)	5.06x10 ⁻⁴	1.053 (1.018-1.089)	0.00258	15.09
660	1.061 (1.026-1.097)	5.22x10 ⁻⁴	1.053 (1.018-1.089)	0.00259	15.09
670	1.058 (1.023-1.094)	9.19x10 ⁻⁴	1.052 (1.017-1.088)	0.00303	11.54

680	1.060 (1.025-1.096)	5.88×10^{-4}	1.053 (1.019-1.089)	0.00237	13.21
690	1.064 (1.029-1.100)	3.02×10^{-4}	1.055 (1.020-1.091)	0.00176	16.36
700	1.065 (1.030-1.101)	2.28×10^{-4}	1.054 (1.019-1.089)	0.00227	20.37
710	1.065 (1.030-1.102)	2.03×10^{-4}	1.053 (1.018-1.088)	0.00273	22.64
720	1.065 (1.030-1.101)	2.28×10^{-4}	1.053 (1.018-1.089)	0.00257	22.64
730	1.063 (1.027-1.099)	3.81×10^{-4}	1.055 (1.021-1.091)	0.00159	14.55
740	1.064 (1.029-1.100)	2.56×10^{-4}	1.054 (1.019-1.090)	0.00223	18.52
750	1.067 (1.032-1.104)	1.30×10^{-4}	1.053 (1.018-1.089)	0.00269	26.42
760	1.066 (1.031-1.103)	1.62×10^{-4}	1.054 (1.019-1.090)	0.0022	22.22
770	1.065 (1.030-1.101)	2.15×10^{-4}	1.054 (1.019-1.090)	0.00211	20.37
780	1.064 (1.029-1.100)	3.00×10^{-4}	1.054 (1.019-1.090)	0.00202	18.52
790	1.063 (1.028-1.099)	3.72×10^{-4}	1.054 (1.019-1.090)	0.00211	16.67
800	1.066 (1.031-1.103)	1.67×10^{-4}	1.054 (1.019-1.089)	0.00233	22.22
810	1.068 (1.033-1.105)	1.04×10^{-4}	1.054 (1.019-1.090)	0.002	25.93
820	1.069 (1.033-1.105)	1.00×10^{-4}	1.056 (1.021-1.092)	0.00154	23.21
830	1.068 (1.033-1.105)	1.08×10^{-4}	1.055 (1.020-1.090)	0.0019	23.64
840	1.068 (1.033-1.104)	1.19×10^{-4}	1.056 (1.021-1.092)	0.00146	21.43
850	1.066 (1.031-1.103)	1.70×10^{-4}	1.056 (1.021-1.092)	0.00156	17.86
860	1.068 (1.032-1.104)	1.25×10^{-4}	1.056 (1.021-1.092)	0.00142	21.43
870	1.069 (1.034-1.105)	9.69×10^{-5}	1.056 (1.021-1.092)	0.00139	23.21
880	1.067 (1.032-1.103)	1.42×10^{-4}	1.056 (1.021-1.092)	0.00156	19.64
890	1.068 (1.033-1.105)	1.05×10^{-4}	1.055 (1.020-1.091)	0.00163	23.64
900	1.071 (1.035-1.107)	6.12×10^{-5}	1.055 (1.020-1.091)	0.00169	29.09
910	1.069 (1.034-1.105)	9.49×10^{-5}	1.055 (1.020-1.091)	0.00174	25.45
920	1.067 (1.032-1.104)	1.34×10^{-4}	1.055 (1.020-1.091)	0.00182	21.82
930	1.071 (1.036-1.107)	6.04×10^{-5}	1.055 (1.020-1.091)	0.00181	29.09
940	1.074 (1.039-1.110)	2.89×10^{-5}	1.054 (1.019-1.090)	0.00213	37.04
950	1.075 (1.040-1.112)	2.07×10^{-5}	1.054 (1.019-1.090)	0.00227	38.89
960	1.074 (1.039-1.111)	2.54×10^{-5}	1.054 (1.019-1.090)	0.00219	37.04
970	1.075 (1.039-1.111)	2.48×10^{-5}	1.054 (1.019-1.090)	0.00229	38.89
980	1.075 (1.040-1.112)	2.14×10^{-5}	1.054 (1.019-1.090)	0.00217	38.89
990	1.072 (1.037-1.108)	4.48×10^{-5}	1.054 (1.019-1.090)	0.00213	33.33
1000	1.071 (1.036-1.107)	5.98×10^{-5}	1.054 (1.019-1.090)	0.00232	31.48
1010	1.072 (1.036-1.108)	5.06×10^{-5}	1.053 (1.018-1.089)	0.00245	35.85
1020	1.070 (1.035-1.106)	7.65×10^{-5}	1.053 (1.018-1.089)	0.00259	32.08
1030	1.068 (1.033-1.104)	1.23×10^{-4}	1.052 (1.017-1.088)	0.00305	30.77
1040	1.067 (1.031-1.103)	1.61×10^{-4}	1.052 (1.017-1.088)	0.00301	28.85
1050	1.067 (1.032-1.103)	1.55×10^{-4}	1.052 (1.017-1.088)	0.00325	28.85
1060	1.067 (1.031-1.103)	1.59×10^{-4}	1.052 (1.017-1.088)	0.00308	28.85

1070	1.067 (1.032-1.103)	1.42x10 ⁻⁴	1.052 (1.017-1.088)	0.00311	28.85
1080	1.065 (1.030-1.101)	2.33x10 ⁻⁴	1.052 (1.017-1.088)	0.003	25.00
1090	1.065 (1.029-1.101)	2.53x10 ⁻⁴	1.053 (1.018-1.089)	0.00278	22.64
1100	1.065 (1.030-1.101)	2.37x10 ⁻⁴	1.053 (1.018-1.089)	0.00256	22.64
1110	1.065 (1.030-1.101)	2.37x10 ⁻⁴	1.053 (1.018-1.089)	0.00277	22.64
1120	1.066 (1.031-1.102)	1.95x10 ⁻⁴	1.052 (1.017-1.088)	0.00293	26.92
1130	1.067 (1.031-1.103)	1.61x10 ⁻⁴	1.052 (1.017-1.088)	0.00306	28.85
1140	1.067 (1.032-1.103)	1.44x10 ⁻⁴	1.052 (1.017-1.088)	0.00295	28.85
1150	1.067 (1.032-1.103)	1.54x10 ⁻⁴	1.052 (1.017-1.088)	0.00314	28.85
1160	1.067 (1.032-1.104)	1.43x10 ⁻⁴	1.052 (1.017-1.088)	0.00297	28.85
1170	1.066 (1.031-1.103)	1.78x10 ⁻⁴	1.053 (1.018-1.089)	0.00262	24.53
1180	1.067 (1.031-1.103)	1.68x10 ⁻⁴	1.053 (1.018-1.089)	0.00264	26.42
1190	1.067 (1.031-1.103)	1.67x10 ⁻⁴	1.053 (1.018-1.089)	0.00273	26.42
1200	1.067 (1.031-1.103)	1.62x10 ⁻⁴	1.053 (1.018-1.089)	0.00274	26.42
1210	1.065 (1.030-1.102)	2.12x10 ⁻⁴	1.053 (1.018-1.089)	0.00278	22.64
1220	1.064 (1.029-1.100)	2.82x10 ⁻⁴	1.053 (1.018-1.089)	0.00276	20.75
1230	1.065 (1.029-1.101)	2.60x10 ⁻⁴	1.052 (1.017-1.088)	0.00294	25.00
1240	1.065 (1.029-1.101)	2.52x10 ⁻⁴	1.052 (1.017-1.088)	0.00297	25.00
1250	1.066 (1.031-1.103)	1.81x10 ⁻⁴	1.052 (1.017-1.088)	0.0031	26.92
1260	1.066 (1.030-1.102)	2.06x10 ⁻⁴	1.052 (1.017-1.088)	0.00309	26.92
1270	1.065 (1.030-1.102)	2.14x10 ⁻⁴	1.052 (1.017-1.088)	0.00323	25.00
1280	1.065 (1.030-1.101)	2.31x10 ⁻⁴	1.052 (1.017-1.088)	0.00323	25.00
1290	1.065 (1.030-1.101)	2.44x10 ⁻⁴	1.052 (1.017-1.088)	0.00317	25.00
1300	1.065 (1.030-1.101)	2.43x10 ⁻⁴	1.052 (1.017-1.088)	0.00318	25.00
1310	1.065 (1.030-1.102)	2.15x10 ⁻⁴	1.052 (1.017-1.088)	0.00317	25.00
1320	1.065 (1.029-1.101)	2.53x10 ⁻⁴	1.052 (1.017-1.088)	0.00311	25.00
1330	1.062 (1.027-1.098)	4.58x10 ⁻⁴	1.052 (1.017-1.088)	0.00313	19.23
1340	1.062 (1.027-1.098)	4.46x10 ⁻⁴	1.052 (1.017-1.088)	0.00309	19.23
1350	1.063 (1.027-1.099)	3.98x10 ⁻⁴	1.052 (1.017-1.088)	0.0031	21.15
1360	1.064 (1.029-1.101)	2.72x10 ⁻⁴	1.052 (1.017-1.088)	0.00311	23.08
1370	1.064 (1.029-1.100)	3.14x10 ⁻⁴	1.052 (1.017-1.088)	0.00312	23.08
1380	1.064 (1.028-1.100)	3.17x10 ⁻⁴	1.052 (1.017-1.088)	0.00309	23.08
1390	1.063 (1.028-1.099)	3.62x10 ⁻⁴	1.052 (1.017-1.088)	0.00308	21.15
1400	1.063 (1.028-1.099)	3.67x10 ⁻⁴	1.052 (1.017-1.088)	0.00309	21.15
1410	1.063 (1.028-1.100)	3.47x10 ⁻⁴	1.052 (1.017-1.088)	0.00308	21.15
1420	1.061 (1.026-1.097)	5.48x10 ⁻⁴	1.052 (1.017-1.088)	0.00316	17.31
1430	1.060 (1.025-1.096)	6.91x10 ⁻⁴	1.052 (1.017-1.088)	0.00318	15.38
1440	1.061 (1.026-1.097)	5.97x10 ⁻⁴	1.052 (1.017-1.088)	0.00323	17.31
1450	1.060 (1.025-1.096)	6.82x10 ⁻⁴	1.051 (1.017-1.087)	0.00347	17.65

1460	1.060 (1.025-1.097)	6.34x10 ⁻⁴	1.051 (1.017-1.087)	0.00347	17.65
1470	1.062 (1.027-1.098)	4.74x10 ⁻⁴	1.051 (1.017-1.087)	0.00347	21.57
1480	1.060 (1.025-1.096)	6.46x10 ⁻⁴	1.052 (1.017-1.087)	0.00345	15.38
1490	1.060 (1.025-1.096)	6.95x10 ⁻⁴	1.052 (1.017-1.087)	0.00344	15.38
1500	1.059 (1.024-1.096)	7.74x10 ⁻⁴	1.052 (1.017-1.088)	0.00333	13.46
1510	1.061 (1.026-1.097)	5.70x10 ⁻⁴	1.052 (1.017-1.088)	0.00311	17.31
1520	1.060 (1.025-1.096)	6.57x10 ⁻⁴	1.052 (1.017-1.088)	0.00301	15.38
1530	1.060 (1.025-1.096)	7.12x10 ⁻⁴	1.052 (1.017-1.088)	0.00297	15.38
1540	1.062 (1.027-1.098)	4.33x10 ⁻⁴	1.053 (1.018-1.088)	0.00285	16.98
1550	1.061 (1.026-1.098)	5.04x10 ⁻⁴	1.053 (1.018-1.089)	0.00282	15.09
1560	1.061 (1.026-1.097)	5.98x10 ⁻⁴	1.052 (1.017-1.088)	0.00308	17.31
1570	1.060 (1.025-1.096)	7.00x10 ⁻⁴	1.052 (1.017-1.088)	0.00306	15.38
1580	1.059 (1.024-1.095)	8.39x10 ⁻⁴	1.052 (1.017-1.088)	0.00319	13.46
1590	1.059 (1.024-1.095)	8.79x10 ⁻⁴	1.052 (1.017-1.088)	0.0033	13.46
1600	1.060 (1.025-1.096)	6.99x10 ⁻⁴	1.051 (1.016-1.087)	0.00358	17.65
1610	1.060 (1.025-1.096)	6.97x10 ⁻⁴	1.052 (1.017-1.088)	0.00338	15.38
1620	1.061 (1.026-1.097)	5.95x10 ⁻⁴	1.052 (1.017-1.088)	0.00327	17.31
1630	1.061 (1.025-1.097)	6.03x10 ⁻⁴	1.052 (1.017-1.088)	0.00335	17.31
1640	1.062 (1.027-1.098)	4.78x10 ⁻⁴	1.052 (1.017-1.088)	0.00324	19.23
1650	1.062 (1.027-1.099)	4.11x10 ⁻⁴	1.052 (1.017-1.088)	0.00306	19.23
1660	1.061 (1.026-1.097)	5.49x10 ⁻⁴	1.052 (1.017-1.088)	0.00302	17.31
1670	1.061 (1.026-1.098)	5.33x10 ⁻⁴	1.053 (1.018-1.088)	0.00285	15.09
1680	1.062 (1.027-1.098)	4.64x10 ⁻⁴	1.052 (1.017-1.088)	0.00311	19.23
1690	1.062 (1.027-1.099)	4.29x10 ⁻⁴	1.052 (1.018-1.088)	0.00288	19.23
1700	1.062 (1.026-1.098)	5.00x10 ⁻⁴	1.053 (1.018-1.089)	0.00268	16.98
1710	1.061 (1.026-1.097)	5.41x10 ⁻⁴	1.053 (1.018-1.089)	0.0026	15.09
1720	1.061 (1.026-1.097)	5.85x10 ⁻⁴	1.053 (1.018-1.089)	0.00265	15.09
1730	1.062 (1.027-1.099)	4.22x10 ⁻⁴	1.053 (1.018-1.089)	0.00248	16.98
1740	1.062 (1.027-1.098)	4.56x10 ⁻⁴	1.053 (1.018-1.089)	0.00271	16.98
1750	1.062 (1.027-1.098)	4.44x10 ⁻⁴	1.053 (1.018-1.089)	0.00276	16.98
1760	1.063 (1.027-1.099)	3.99x10 ⁻⁴	1.053 (1.018-1.088)	0.00284	18.87
1770	1.062 (1.026-1.098)	5.01x10 ⁻⁴	1.053 (1.018-1.089)	0.00266	16.98
1780	1.062 (1.027-1.098)	4.87x10 ⁻⁴	1.053 (1.018-1.089)	0.00248	16.98
1790	1.062 (1.027-1.098)	4.48x10 ⁻⁴	1.053 (1.018-1.089)	0.00259	16.98
1800	1.060 (1.025-1.097)	6.33x10 ⁻⁴	1.053 (1.018-1.089)	0.00256	13.21
1810	1.061 (1.026-1.097)	5.70x10 ⁻⁴	1.053 (1.018-1.089)	0.00264	15.09
1820	1.061 (1.026-1.098)	5.18x10 ⁻⁴	1.052 (1.017-1.088)	0.00311	17.31
1830	1.061 (1.026-1.097)	5.62x10 ⁻⁴	1.051 (1.017-1.087)	0.00347	19.61
1840	1.060 (1.025-1.097)	6.48x10 ⁻⁴	1.052 (1.017-1.087)	0.00341	15.38

1850	1.060 (1.025-1.096)	7.20x10 ⁻⁴	1.052 (1.017-1.088)	0.00309	15.38
1860	1.061 (1.026-1.097)	5.80x10 ⁻⁴	1.052 (1.017-1.088)	0.00319	17.31
1870	1.061 (1.026-1.098)	5.23x10 ⁻⁴	1.052 (1.017-1.088)	0.00309	17.31
1880	1.061 (1.026-1.098)	5.10x10 ⁻⁴	1.051 (1.016-1.087)	0.00367	19.61
1890	1.061 (1.025-1.097)	6.09x10 ⁻⁴	1.051 (1.016-1.087)	0.00369	19.61
1900	1.060 (1.025-1.096)	6.89x10 ⁻⁴	1.050 (1.015-1.086)	0.00422	20.00
1910	1.060 (1.025-1.096)	6.92x10 ⁻⁴	1.051 (1.016-1.087)	0.00393	17.65
1920	1.061 (1.025-1.097)	6.16x10 ⁻⁴	1.051 (1.016-1.087)	0.00353	19.61
1930	1.061 (1.026-1.097)	5.79x10 ⁻⁴	1.052 (1.017-1.088)	0.00293	17.31
1940	1.060 (1.025-1.096)	6.75x10 ⁻⁴	1.053 (1.018-1.089)	0.00249	13.21
1950	1.058 (1.023-1.095)	9.53x10 ⁻⁴	1.053 (1.018-1.089)	0.00268	9.43
1960	1.058 (1.023-1.095)	9.61x10 ⁻⁴	1.053 (1.018-1.089)	0.00252	9.43
1970	1.059 (1.024-1.095)	8.04x10 ⁻⁴	1.053 (1.018-1.089)	0.00252	11.32
1980	1.059 (1.024-1.095)	8.48x10 ⁻⁴	1.053 (1.019-1.089)	0.00238	11.32
1990	1.060 (1.025-1.097)	6.38x10 ⁻⁴	1.053 (1.018-1.089)	0.0024	13.21
2000	1.060 (1.025-1.096)	7.20x10 ⁻⁴	1.053 (1.019-1.089)	0.00239	13.21
2010	1.059 (1.024-1.095)	8.23x10 ⁻⁴	1.053 (1.018-1.089)	0.00252	11.32
2020	1.059 (1.024-1.095)	8.39x10 ⁻⁴	1.053 (1.018-1.089)	0.00247	11.32
2030	1.059 (1.024-1.095)	8.20x10 ⁻⁴	1.052 (1.017-1.088)	0.00311	13.46
2040	1.059 (1.024-1.095)	8.46x10 ⁻⁴	1.053 (1.018-1.088)	0.00278	11.32
2050	1.059 (1.024-1.095)	8.37x10 ⁻⁴	1.054 (1.019-1.090)	0.00232	9.26
2060	1.058 (1.023-1.094)	9.97x10 ⁻⁴	1.054 (1.019-1.090)	0.00218	7.41
2070	1.058 (1.023-1.094)	0.00105	1.055 (1.020-1.091)	0.00171	5.45
2080	1.058 (1.023-1.094)	0.00111	1.057 (1.022-1.092)	0.00133	1.75
2090	1.057 (1.022-1.093)	0.00134	1.057 (1.022-1.093)	0.00124	0.00
2100	1.056 (1.021-1.092)	0.00153	1.057 (1.022-1.093)	0.00113	-1.75
2110	1.057 (1.022-1.093)	0.00136	1.057 (1.022-1.093)	0.00128	0.00
2120	1.057 (1.022-1.093)	0.00133	1.056 (1.021-1.092)	0.00146	1.79
2130	1.057 (1.022-1.093)	0.00134	1.055 (1.020-1.090)	0.00194	3.64
2140	1.057 (1.022-1.093)	0.00123	1.055 (1.020-1.091)	0.00167	3.64
2150	1.057 (1.022-1.093)	0.00125	1.056 (1.021-1.092)	0.00135	1.79
2160	1.057 (1.022-1.093)	0.00118	1.055 (1.020-1.091)	0.00165	3.64
2170	1.057 (1.022-1.093)	0.00129	1.056 (1.021-1.092)	0.00152	1.79
2180	1.057 (1.022-1.093)	0.00121	1.056 (1.021-1.092)	0.00138	1.79
2190	1.056 (1.021-1.093)	0.00139	1.056 (1.021-1.092)	0.00142	0.00
2200	1.056 (1.021-1.092)	0.0015	1.057 (1.022-1.093)	0.00128	-1.75
2210	1.056 (1.021-1.092)	0.00149	1.056 (1.021-1.092)	0.00145	0.00
2220	1.056 (1.021-1.092)	0.00145	1.056 (1.021-1.092)	0.00139	0.00
2230	1.057 (1.022-1.093)	0.00128	1.056 (1.021-1.092)	0.0014	1.79

2240	1.056 (1.021-1.092)	0.00157	1.057 (1.022-1.093)	0.00115	-1.75
2250	1.055 (1.020-1.091)	0.00185	1.056 (1.021-1.092)	0.00134	-1.79
2260	1.055 (1.020-1.091)	0.00188	1.057 (1.022-1.093)	0.00124	-3.51
2270	1.055 (1.020-1.091)	0.00199	1.058 (1.023-1.094)	9.15x10 ⁻⁴	-5.17
2280	1.056 (1.021-1.092)	0.00158	1.059 (1.024-1.095)	7.40x10 ⁻⁴	-5.08
2290	1.055 (1.020-1.091)	0.00177	1.060 (1.025-1.096)	6.32x10 ⁻⁴	-8.33
2300	1.055 (1.020-1.091)	0.00185	1.060 (1.025-1.096)	7.00x10 ⁻⁴	-8.33
2310	1.055 (1.020-1.091)	0.00175	1.059 (1.024-1.095)	7.42x10 ⁻⁴	-6.78
2320	1.055 (1.020-1.091)	0.00172	1.061 (1.026-1.097)	5.68x10 ⁻⁴	-9.84
2330	1.056 (1.021-1.092)	0.00162	1.059 (1.024-1.095)	7.47x10 ⁻⁴	-5.08
2340	1.056 (1.021-1.092)	0.00163	1.059 (1.024-1.095)	8.30x10 ⁻⁴	-5.08
2350	1.056 (1.021-1.092)	0.00158	1.060 (1.025-1.096)	6.06x10 ⁻⁴	-6.67
2360	1.055 (1.020-1.091)	0.00171	1.061 (1.026-1.097)	5.22x10 ⁻⁴	-9.84
2370	1.056 (1.021-1.092)	0.00146	1.061 (1.026-1.097)	5.52x10 ⁻⁴	-8.20
2380	1.057 (1.022-1.093)	0.00132	1.061 (1.026-1.097)	5.36x10 ⁻⁴	-6.56
2390	1.056 (1.021-1.092)	0.00143	1.060 (1.025-1.096)	6.62x10 ⁻⁴	-6.67
2400	1.057 (1.022-1.093)	0.0013	1.062 (1.027-1.098)	4.43x10 ⁻⁴	-8.06
2410	1.057 (1.022-1.093)	0.0013	1.060 (1.025-1.096)	6.02x10 ⁻⁴	-5.00
2420	1.057 (1.022-1.093)	0.00133	1.058 (1.023-1.094)	9.79x10 ⁻⁴	-1.72
2430	1.056 (1.021-1.093)	0.0014	1.057 (1.022-1.093)	0.00111	-1.75
2440	1.056 (1.021-1.092)	0.00145	1.056 (1.021-1.092)	0.00139	0.00
2450	1.056 (1.021-1.092)	0.00142	1.056 (1.021-1.092)	0.00134	0.00
2460	1.056 (1.021-1.092)	0.00163	1.056 (1.021-1.092)	0.00146	0.00
2470	1.056 (1.021-1.092)	0.00161	1.056 (1.021-1.092)	0.00152	0.00
2480	1.056 (1.021-1.092)	0.00147	1.055 (1.020-1.090)	0.00193	1.82
2490	1.056 (1.021-1.092)	0.00156	1.054 (1.020-1.090)	0.00195	3.70
2500	1.056 (1.021-1.092)	0.00145	1.053 (1.018-1.089)	0.00241	5.66
2510	1.057 (1.022-1.093)	0.00132	1.054 (1.019-1.089)	0.00229	5.56
2520	1.058 (1.022-1.094)	0.00113	1.054 (1.019-1.090)	0.00205	7.41
2530	1.058 (1.023-1.094)	0.00106	1.054 (1.019-1.090)	0.00213	7.41
2540	1.058 (1.023-1.094)	0.00105	1.056 (1.021-1.092)	0.00138	3.57
2550	1.058 (1.023-1.094)	0.00107	1.058 (1.023-1.093)	0.00108	0.00
2560	1.058 (1.023-1.094)	0.00111	1.057 (1.022-1.093)	0.0013	1.75
2570	1.057 (1.022-1.094)	0.00115	1.055 (1.020-1.091)	0.00187	3.64
2580	1.057 (1.022-1.093)	0.00128	1.055 (1.020-1.090)	0.00193	3.64
2590	1.057 (1.022-1.093)	0.00129	1.054 (1.019-1.090)	0.00205	5.56
2600	1.056 (1.021-1.092)	0.00146	1.054 (1.019-1.090)	0.00224	3.70
2610	1.056 (1.021-1.092)	0.00154	1.053 (1.018-1.089)	0.00243	5.66
2620	1.056 (1.021-1.092)	0.00154	1.053 (1.018-1.089)	0.00268	5.66

2630	1.055 (1.020-1.091)	0.00173	1.052 (1.018-1.088)	0.00291	5.77
2640	1.056 (1.021-1.092)	0.00156	1.050 (1.016-1.086)	0.00412	12.00
2650	1.057 (1.022-1.093)	0.00135	1.053 (1.018-1.089)	0.00271	7.55
2660	1.057 (1.022-1.093)	0.00128	1.054 (1.019-1.090)	0.00215	5.56
2670	1.057 (1.022-1.093)	0.00136	1.054 (1.020-1.090)	0.00197	5.56
2680	1.057 (1.022-1.093)	0.0013	1.056 (1.021-1.092)	0.00143	1.79
2690	1.056 (1.021-1.092)	0.00142	1.054 (1.019-1.090)	0.00223	3.70
2700	1.057 (1.022-1.093)	0.00135	1.054 (1.019-1.090)	0.00208	5.56
2710	1.057 (1.022-1.093)	0.00136	1.052 (1.018-1.088)	0.00282	9.62
2720	1.056 (1.021-1.093)	0.00141	1.051 (1.017-1.087)	0.00349	9.80
2730	1.057 (1.022-1.093)	0.00128	1.052 (1.017-1.088)	0.00297	9.62
2740	1.057 (1.022-1.093)	0.00134	1.052 (1.017-1.088)	0.00312	9.62
2750	1.057 (1.022-1.093)	0.00128	1.053 (1.018-1.089)	0.00261	7.55
2760	1.057 (1.022-1.093)	0.00125	1.053 (1.018-1.089)	0.00263	7.55
2770	1.057 (1.022-1.093)	0.00122	1.051 (1.017-1.087)	0.00348	11.76
2780	1.057 (1.022-1.093)	0.00125	1.049 (1.014-1.085)	0.00533	16.33
2790	1.057 (1.022-1.093)	0.00127	1.052 (1.017-1.088)	0.00297	9.62
2800	1.057 (1.022-1.093)	0.00128	1.051 (1.017-1.087)	0.00349	11.76
2810	1.057 (1.022-1.093)	0.00125	1.053 (1.018-1.089)	0.00243	7.55
2820	1.057 (1.022-1.093)	0.00135	1.055 (1.020-1.090)	0.00187	3.64
2830	1.057 (1.021-1.093)	0.00137	1.053 (1.019-1.089)	0.00239	7.55
2840	1.057 (1.022-1.093)	0.00136	1.052 (1.017-1.088)	0.00301	9.62
2850	1.057 (1.022-1.093)	0.00133	1.052 (1.017-1.088)	0.00303	9.62
2860	1.056 (1.021-1.093)	0.00139	1.051 (1.016-1.087)	0.00355	9.80
2870	1.056 (1.021-1.092)	0.00147	1.049 (1.014-1.085)	0.00511	14.29
2880	1.056 (1.021-1.092)	0.0015	1.050 (1.016-1.086)	0.00409	12.00
2890	1.056 (1.021-1.092)	0.00156	1.052 (1.017-1.088)	0.00305	7.69
2900	1.056 (1.021-1.092)	0.00152	1.051 (1.016-1.087)	0.00366	9.80
2910	1.056 (1.021-1.092)	0.00155	1.051 (1.016-1.086)	0.00398	9.80
2920	1.056 (1.021-1.092)	0.00155	1.050 (1.015-1.086)	0.00455	12.00
2930	1.056 (1.021-1.092)	0.00159	1.049 (1.014-1.084)	0.00566	14.29
2940	1.056 (1.021-1.092)	0.00166	1.049 (1.014-1.085)	0.00537	14.29
2950	1.056 (1.021-1.092)	0.00164	1.046 (1.011-1.082)	0.00884	21.74
2960	1.055 (1.020-1.091)	0.00171	1.044 (1.009-1.080)	0.012	25.00
2970	1.056 (1.021-1.092)	0.0016	1.044 (1.009-1.079)	0.0128	27.27
2980	1.056 (1.021-1.092)	0.0016	1.042 (1.007-1.077)	0.0168	33.33
2990	1.056 (1.021-1.092)	0.00162	1.040 (1.006-1.076)	0.0212	40.00
3000	1.056 (1.021-1.092)	0.00162	1.039 (1.004-1.074)	0.0274	43.59

^a Calculated as the relative percent difference in RV-EXCALIBER RVGRS OR versus TRAPD RVGRS OR

Minor comments:

line 86: "due to?"

This grammatical error has now been corrected:

“1) a correction that accounts for the increased variance *due to* the presence of rare variants in LD...” (Updated main text page 3, line 89-90)

Figure legends: In a few places, equations were referred without explaining what and how things were calculated. These need to be elaborated.

We thank the reviewer for bringing this to our attention. We have now added a thorough description for how our parameters were calculated in each figure caption in which an equation was referenced (i.e. Figure 2, Figure 3, Supplementary Figure 1, and Supplementary Figure 3). For thoroughness and consistency, we have also added descriptions for how the iCF/gCF was calculated to Figure 4, Supplementary Figure 4, and Supplementary Figure 5. We have noted the changes below:

Figure 2:

“Simulated effects of SFN rate, SFP rate, and PSF on estimated OR and association power. Probability of mutation (P) within a gene was calculated using equation 2 by simulating true OR values from 1 to 2 in 0.1 stage intervals while keeping SFN, SFP and PSF fixed. Thereafter, an estimated OR was calculated for every unique P (A-C; red lines). CF values were calculated as the ratio of the P for each estimated OR and the original cumulative minor allele frequency for a single gene ($CMAF_{Gene}$; set at 0.05) according to equation 3. Each CF was then used to adjust the $CMAF_{Gene}$ to calculate an adjusted P (P^*) according to equation 4 while keeping SFN, SFP, and PSF fixed at the same values. Using P^* , a CF-adjusted estimated OR was calculated (A-C; blue lines). Dashed lines in A-C indicate the line of expectation. CF-adjusted power curves were generated using different ranges of SFN (0 to 1), SFP (0 to 1), and PSF (0 to 2) at a fixed true OR value of 1.3 (D-F). Power for every value of SFN, SFP, and PSF was calculated as the proportion of all simulations with a p -value < 0.05 . Dashed lines in F represents a null deviation in PSF. OR indicates odds ratio, CF indicates correction factor, SFN indicates sequencing false negatives, SFP indicates sequencing false positives, PSF indicates population-specific factor.”

Figure 3:

“Comparison of iCF values for consensus GIAB samples stratified by allele frequency bin. Ethnic comparisons include the Ashkenazi GIAB sample (NA24385) with gnomAD Non-Finnish Europeans and gnomAD Ashkenazis (A and B), the North-western European GIAB sample (NA12878) with gnomAD Non-Finnish Europeans (C), and the East Asian GIAB sample (NA24631) with gnomAD East Asians (D). iCF values were calculated as the exome-wide ratio of the sum of per-individual observed allele counts (OAC) to the sum of per-individual expected allele counts (EAC) according to equation 5 for all exonic alleles. AF bins were chosen to best stratify variants according to rare [0-0.01], low-frequency [0.01-0.05], common [0.5-0.25] and very common [0.25-0.5] bins. AFs were standardized to the minor allele. All shaded regions depict the 95% confidence interval. Dashed lines indicate an iCF representing equal mutation loads between

the GIAB sample and gnomAD (i.e. $iCF = 1$) across all protein-coding genes. iCF indicates individual correction factor, and GIAB indicates Genome In A Bottle, and gnomAD indicates genome aggregation database.”

Figure 4 (we also clarified our description of how the gCF was calculated in addition to further explaining the parameters of the equation):

“Correction factor adjustment between studies and across genes. Distribution of iCF s for 6,082 healthy controls from 8 cohorts in MIGen were determined according to *rare pathogenic alleles* across all protein-coding genes (A). iCF values were calculated as the exome-wide ratio of the sum of per-individual observed allele counts (OAC) to the sum of per-individual expected allele counts (EAC) according to equation 5. The violins demonstrate the spread of iCF among all 8 MIGen cohorts. The horizontal line in each boxplot indicate the median iCF values while the top and bottom lines represent the 75th and 25th percentiles of the iCF distribution, respectively. Length of boxplot represents the inter-quartile range of iCF values. Dashed line represents an iCF corresponding to equal mutation loads between a MIGen participant and gnomAD (i.e. $iCF = 1$). gCF values were computed as the ratio of the sum of the OAC to the sum of iCF -adjusted EAC across 1) all individuals and 2) all genes that were organized into one of 50 gene bins according to equation 7. Gene bins were ascertained according to quintile of iCF -adjusted EAC and decile of P-value obtained from a rare variant association test (using burden of rare pathogenic alleles) conducted in the ranking cohort, consisting of the remaining 2,730 healthy control participants in MIGen as “cases” and gnomAD non-Finnish Europeans as controls. The iCF and gCF -adjusted EAC was thereafter calculated according to equation 8. The cumulative delta mutation count was calculated from the cumulative sum of the per-gene difference between OAC and either the iCF -adjusted EAC (red) or iCF and gCF -adjusted (blue) EAC derived across 3,352 healthy control participants in MIGen and gnomAD, respectively, according to equation 9. The cumulative delta mutation count demarcates genes that are systematically enriched (Enr), well-calibrated (WC) and depleted (Dep) for rare pathogenic alleles among MIGen control participants (B). The height of the mountain (red) demonstrates the degree of adjustment offered by the iCF in order to achieve calibration between MIGen controls and gnomAD. Implementing the gCF (blue) mitigates residual gene-level biases that cannot be accounted for by the iCF alone. iCF indicates individual correction factor, and gCF indicates gene correction factor.”

Supplementary Figure 1 (this is a new figure that was added in response the reviewer’s second main comment and contains a full explanation of the referenced equations):

“Simulated effects of combined SFN rate, SFP rate, and PSF on estimated OR. Probability of mutation (P) was calculated as a function of the cumulative minor allele frequency for a single gene ($CMAF_{Gene}$: set at 0.05), the true effect size of association (OR), and the 3 major association biases (SFN, SFP, and PSF) according to equation 2. The simulations kept the true OR fixed at 1.3 (black dashed lines) while varying SFN rate (0 to 1), SFP rate (0.1, 0.2, and 0.3), and PSF (0.3 and 1.7). An estimated OR was calculated for each P (corresponding to a unique combination of the 3 association biases) and plotted as a function of the combined biases (A-F; red lines). CF values were calculated as the ratio of the P for each estimated OR and the original $CMAF_{Gene}$ according to equation 3. Each CF was then used to adjust the $CMAF_{Gene}$ to calculate an adjusted P (P^*) according to equation 4. Using P^* , an adjusted estimated OR was plotted as a function of

the combined biases (A-F; blue lines). OR indicates odds ratio, CF indicates correction factor, SFN indicates sequencing false negatives, SFP indicates sequencing false positives, PSF indicates population-specific factor.”

Supplementary Figure 3:

“GIAB iCF values for gene groups stratified by decile of gene constraint score. Ethnic comparisons include the Ashkenazi GIAB sample (NA24385) with gnomAD non-Finnish Europeans and gnomAD Ashkenazis (A and B), the North-western European GIAB sample (NA12878) with gnomAD Non-Finnish Europeans (C), and the East Asian GIAB sample (NA24631) with gnomAD East Asians (D). All genes were grouped by decile according to their missense constraint metric (Lek et al., 2016). iCF values were calculated all genes in a given decile as ratio of the sum of per-individual observed allele counts (OAC) to the sum of per-individual expected allele counts (EAC) according to equation 5. Error bars depict 95% CI and dashed lines represent the iCF across all deciles (i.e. exome-wide iCF). P-values for heterogeneity (pHet) across deciles were calculated using a fixed effect model. Any P-value < 0.0125 (0.05/4) was considered significant. iCF indicates individual correction factor, GIAB indicates Genome In A Bottle, and gnomAD indicates genome aggregation database.”

Supplementary Figure 4:

“iCF values for cases and controls across all discovery MIGen cohorts. Distribution of iCF values are shown for case (red) and control (blue) participants across 8 MIGen cohorts. iCF values were calculated as the exome-wide ratio of the sum of per-individual observed allele counts (OAC) to the sum of per-individual expected allele counts (EAC) according to equation 5. Violins demonstrate the spread of iCF values. The horizontal line in each boxplot indicate the median iCF value while the top and bottom lines represent the 75th and 25th percentiles of the iCF distribution, respectively. Length of boxplot represents the inter-quartile range of iCF values. Cohorts marked with an asterisk were found to have a significantly different ($P < 0.05$) iCF distribution between cases and controls after adjusting for sex and the first 20 principal components of ancestry (age was not available as a phenotype in MIGen). The dashed line represents an equal mutation load between MIGen cases or controls versus gnomAD non-Finnish Europeans (i.e. $iCF=1$). iCF indicates individual correction factor.”

Supplementary Figure 5 (as done in Figure 4, we clarified our description of how the gCF was calculated in addition to further explaining the parameters of the equation):

“gCF values across 50 gene bins derived from 3,352 healthy MIGen controls. gCF values were computed as the ratio of the sum of the observed allele counts (OAC) to the sum of iCF-adjusted expected allele counts (EAC) across 1) all individuals and 2) all genes that were organized into one of 50 gene bins according to equation 7. Gene bins were ascertained according to quintile of iCF-adjusted EAC and decile of P-value obtained from a rare variant association test (using burden of rare pathogenic alleles) conducted in the ranking cohort, consisting of the remaining 2,730 healthy control participants in MIGen as “cases” and gnomAD non-Finnish Europeans as controls. R^2 and significance was evaluated through a linear regression model. gCF indicates gene correction factor.”

Supplementary Figure 9 (this is a new figure that was added in response the reviewer's sixth main comment and contains a full explanation of the parameters in the referenced equations):

“Dispersion of delta counts generated using RV-EXCALIBER or TRAPD across the spectrum of expected allele count. Percent delta mutation counts were calculated according to equation 9 as the per-gene difference between observed allele count (OAC) and expected allele count (EAC) across 3,352 healthy control participants from MIGen and gnomAD non-Finnish Europeans, respectively. EAC from RV-EXCALIBER are adjusted by the both the iCF and gCF according to equation 8, while the OAC and EAC for TRAPD were obtained from percentile cut-offs of variant-level quality-by-depth scores for the same 3,352 healthy control participants from MIGen and gnomAD, respectively (see main text results). The EAC for either method was log transformed to a normal distribution and assigned to quintiles. A regression line representing the mean percent difference between per-gene OAC and EAC was then plotted for both RV-EXCALIBER (A; blue line) and TRAPD (B; red line). Horizontal dashed lines indicate zero percent difference between OAC and EAC. EAC indicate expected allele count. EAC indicates expected allele count.”

line 268: FRS (Need to explain it's a Framingham risk score for the readers).

We thank the reviewer for the astute observation. We have now noted that FRS corresponds to a Framingham risk score:

“Moreover, we observed a significant trend of higher RVGRS950 estimates among individuals with lower clinical risk for CAD ($P_{interaction}=0.015$), where individuals in low, middle, and high tertiles of a Framingham risk score (FRS) had ORs of 1.16, 1.10, 1.03 conferred through RVGRS950 for CAD, respectively.” (Updated main text page 13; line 298-301)

Net reclassification index: NRI is useful, but it's not very comparable across different studies. AUC and Nagelkerke's R^2 are more widely reported for PRS.

We thank the reviewer for this correct assertion. Indeed, most recent studies report the % variance explained by common variant genetic risk scores (CVGRS) in terms of a pseudo R^2 metric known as Nagelkerke's R^2 . We complemented or net reclassification index by identifying the difference in Nagelkerke's R^2 between the logistic model containing the rare variant genetic risk score (RVGRS) and all covariates (i.e. CVGRS, Framingham risk score, age, age², sex, and the first 20 principal components of ancestry) and the model containing covariates alone, thus obtaining the % variance explained by the RVGRS. We identified that the RVGRS950 and RVGRS950^{LDLR} both contribute to 0.1% of variance not explained by other genetic and environmental risk factors of CAD. While this R^2 is low when compared to that of traditional CVGRS, it cannot be compared directly with it since 1) discovery sample sizes use to generate common variant PRS are far higher, thus providing more model calibration in the validation set and 2) the prevalence of the exposure variables (RVGRS vs. CVGRS) are drastically different. Moreover, it is known that AUC can be insensitive to significant and impactful variables that impact disease status discrimination (see DOI: 10.1161/CIRCULATIONAHA.106.672402). As such, we elected to keep the net reclassification index as our primary discrimination variable, but have noted the Nagelkerke's R^2 for thoroughness:

“Additionally, we identified that the variance explained by both the RVGRS950 and RVGRS950^{LDLR} to be 0.1% when comparing to a logistic model containing CVGRS, FRS, age, age², sex, and the first 20 principal components of ancestry alone.” (Updated main text page 13; lines 310-313)

“In fact, adding RVGRS to a reference model consisting of CVGRS and clinical risk factors significantly reclassified 5.7% of CAD events, which represents a marked improvement given the relatively small size of our discovery population. However, the RVGRS only explained 0.1% of variance in CAD affection status, which is lower compared to traditional CVGRS for CAD. This reason for this large difference in explained variance are two-fold. First is the substantially larger discovery sample size used to generate weights for CVGRS, which will necessarily increase model calibration when assessing risk an independent validation population. Second is the difference in exposure frequency between the RVGRS and CVGRS, wherein the component variants used to inform the RVGRS are necessarily observed less commonly in any given population. As such, the RVGRS (as it stands) will be less powered to detect population-level variances, relative to established CVGRS for CAD. Despite this, we maintain that the RVGRS demonstrates relatively high efficacy at demarcating individual-level risk compared to explaining a high degree of population-level variance. (Updated main text pages 18-19 lines 425-438)

“Variance of affection status explained by RVGRS950 and RVGRS950^{LDLR} was evaluated using Nagelkerke’s pseudo R², where the difference in R² between the full model (i.e. RVGRS and covariates) and the reduced model (i.e. covariates alone) was used as the variance attributable to the RVGRS. Model covariates included CVGRS, FRS, age, age², sex, and the first 20 principal components of ancestry. The Nagelkerke’s pseudo R² metric was calculated using the rms R package⁴⁵.” (Updated supplementary appendix page 24; lines 497-502)

Reviewers' Comments:

Reviewer #1:

Remarks to the Author:

The authors have provided an excellent and extremely comprehensive set of responses to reviewers' comments. The manuscript is substantially improved. I have no further comments.

Reviewer #3:

Remarks to the Author:

Figure 2 and Supplementary Figure 1: 1) I suggest that the main text - both the description of findings in the Results section as well as the main figure - is primarily centered around the full simulated data which account for all three bisexes, and current figure 2 should go to the supplements instead. 2) In a similar vein, the current power analysis is limited as only one factor is accounted for at each time. 3) Aren't the simulated population-specific factors of 0.3 and 1.7 too extreme in the light of Fig 4A? It looks to me that more realistic ranges are around +/- 0.2.

Mutational load: 1) I agree that iCF is heavily affected by technological artifacts. 2) If iCF is heavily influenced by sequencing technology, it shouldn't be called or interpreted as the mutational load. This is misleading. Although we sometimes use the count of rare deleterious variants in sequenced genomes as a proxy for the mutational load, it is always restricted to the highest-quality regions which are least likely to be affected by sequencing artifacts. 3) I agree that rare deleterious variants are less likely to be shared between populations. However, the issue is whether the variation in the numbers of rare deleterious variants between cohorts is driven mainly by population genetic forces underlying the mutational load, such as mutation rates or efficiency of purifying selection (namely, effective population size and growth rates, etc). Nelson et al., Gravel et al., or 1000 genomes paper does not address this issue. Kessler et al. is relevant, but I believe it's more an exception as it also states "we do not find significant differences in DNM rate between individuals of European, African, and Latino ancestry, nor across ancestrally distinct segments within admixed individuals."

Power simulation: Given the weak power of RVGRS in real data, I think that thorough and extensive benchmark simulations are necessary to present the utility of current work. 1) TRAPD was one of the first methods leveraging external controls, thus it's understandable that it lacked extensive benchmark simulations. 2) New polygenic score methods are typically demonstrated to show the improved prediction accuracy in extensive benchmark simulations AND with multiple real phenotypes. I understand applying the method to multiple real phenotypes is not simple to do, though. 3) The reported power gain over TRAPD is small and not significant in the real data. The accuracy of the proposed method should be shown to be significantly better at least in simulations. 4) Supp Fig 9 is interesting but cannot replace simulated benchmarks.

The variance explained by PRS is typically calculated on the liability scale using AUC (PMID: 20195508). But if the AUC does not improve much with RVGRS, the variance additionally explained by RVGRS would be small on the liability scale as well. NRI is interesting, but there is criticism against it, too (PMID: 26504496). It is very hard for me to see how a linear polygenic model doesn't do very well in terms of R^2 and liability but has high predictive power if we are considering a typical complex genetic architecture. What is the Nigelerkerke R^2 explained by RVGRS compared to the R^2 of the full model?

I agree that RVGRS and CVGRS need to be combined for higher accuracy, but it should be noted that the heritability explained by rare protein-altering variants might be small for MI. The predictive power of RVPRS is constrained by the heritability.

Reviewer #1 (Remarks to the Author):

The authors have provided an excellent and extremely comprehensive set of responses to reviewers' comments. The manuscript is substantially improved. I have no further comments.

We would like to extend our gratitude to the reviewer and believe that their constructive comments were essential towards strengthening our manuscript.

Reviewer #3 (Remarks to the Author):

We would like foremost like thank the reviewer for their previous comments. Addressing the points that the reviewer brought to our attention has greatly improved our manuscript's quality to this point. We have herein compressively addressed the reviewer's new comments and believe that they have further strengthened our work.

Figure 2 and Supplementary Figure 1:

1) I suggest that the main text - both the description of findings in the Results section as well as the main figure - is primarily centered around the full simulated data which account for all three biases, and current figure 2 should go to the supplements instead.

We fully agree with the reviewer that the Results section is more so focussed on our findings pertaining to the effect size simulations in the presence of all three biases simultaneously. As such, we have taken the reviewer's advice and moved "Figure 2: Simulated effects of SFN rate, SFP rate, and PSF on estimated OR and association power." to Supplementary Figure 1 and have moved "Supplementary Figure 1: Simulated effects of combined SFN rate, SFP rate, and PSF on estimated OR" to Figure 2.

We have made the appropriate changes in our references to these figures in the updated main text as follows:

"We found that incorporation of the CFs fully and significantly corrected the estimated OR to match the true OR when including either sequencing false negatives or population structure (Supplementary Figure 1A and C). Conversely, we observed that incorporation of the CF over-corrects (i.e. estimated OR < true OR) when sequencing false positives are present (Supplementary Figure 1B). We hold that the adjustment is still effective in the latter scenario given that CF-adjusted effect size of the gene-based association will be conservative. We also observed that the CF robustly calibrates the estimated OR in the presence of all biases simultaneously (i.e. SFN, SFP, and PSF) and adheres to the same patterns as observed with single bias simulations (Figure 2A-F). That is, the CF adequately calibrates the estimated OR to the true OR in the presence of strong SFN and PSF bias, while providing a slightly conservative correction at high SFP rates (Figure 2C and F)." (Updated main text page 5; lines 116-126)

2) In a similar vein, the current power analysis is limited as only one factor is accounted for at each time.

We have added a power analysis (analogous to the updated Supplementary Figure 1 D-F) accounting for all three biases (i.e. SFN, SFP, and PSF) simultaneously. We have shown this in the figure below (along with caption), which is now Supplementary Figure 3:

“Supplementary Figure 3: Power to detect gene-based association signal in the presence of SFN rate, SFP rate, and PSF. CF-adjusted power curves were generated under the simultaneous effect of SFN rate (0 to 1), SFP rate (0.1, 0.2, 0.3), and PSF (0.8 for A and 1.2 for B) at a fixed true odds ratio value of 1.3. Power for every value of SFN, SFP, and PSF was calculated as the proportion of all simulations with a p-value < 0.05, which was calculated as described in section 9A of the Data Supplement.”

We have appended a small addition to the updated main text to reflect this result:

“To estimate the power to detect a gene-based association, we similarly simulated 100,000 observed rare allele counts assuming a true OR of 1.3, which was the effect size necessary to achieve 80% power when no confounding factors exist (i.e. SFN & SFP rates of 0% and a PSF of 1) using our model (Supplementary Figure 1D-F). Under the setting where sequencing false negatives are present, we expect a decrease in mutation rate and thus, power to detect association. Incorporation of the CF mitigates the reduction in power as we observe increased power to detect association across all SFN rates in the CF-adjusted compared to the unadjusted model. When only sequencing false positives are present, we show that incorporation of the CF ameliorates the spurious increase in power observed as mutation rate rises due to increasing rates of SFPs. It is important to note that the artificial increase in power observed in the unadjusted model is accredited solely to sequencing artefacts and not true variants, and thus would translate directly into increased type I error. Lastly, we observe an increase in power as the PSF becomes greater (i.e. the variant count in a test sample is greater than a reference sample due to reasons such as population substructure). In this setting, the power increase is expected as the mutation rate is increasing due to variants that are truly present because of population genetic factors, as opposed to variant artefacts observed in the un-adjusted model including SFP. These features were also observed when the power to detect gene-based association signals was assessed in the presence

SFN rate, SFP rate, and PSF simultaneously (Supplementary Figure 3A and B).” (Updated main text page 6; lines 130-147)

3) Aren't the simulated population-specific factors of 0.3 and 1.7 too extreme in the light of Fig 4A? It looks to me that more realistic ranges are around +/- 0.2.

We agree with the reviewer that magnitude of the population specific factor biases of 0.3 and 1.7 was quite high, especially given the empirical evidence offered by the iCF distribution in Figure 4A, which does indeed show that population-specific factors of 0.8 and 1.2 (i.e., 20% difference) might be more reasonable and realistic. As such we changed the PSFs in updated Figure 2 from 0.3 and 1.7 to 0.8 and 1.2, respectively. We have appended updated Figure 2 and its associated caption below for reference:

“Figure 2: Simulated effects of combined SFN rate, SFP rate, and PSF on estimated OR. Probability of mutation (P) was calculated as a function of the cumulative minor allele frequency for a single gene ($CMAF_{Gene}$: set at 0.05), the true effect size of association (OR), and the 3 major association biases (SFN, SFP, and PSF) according to equation 2. The simulations kept the true OR fixed at 1.3 (black dashed lines) while varying SFN rate (0 to 1), SFP rate (0.1, 0.2, and 0.3), and modest PSF (0.8 and 1.2). An estimated OR was calculated for each P (corresponding to a unique combination of the 3 association biases) and plotted as a function of the combined biases (A-F; red lines). CF values were calculated as the ratio of the P for each estimated OR and the original $CMAF_{Gene}$ according to equation 3. Each CF was then used to adjust the $CMAF_{Gene}$ to calculate an adjusted P (P^*) according to equation 4. Using P^* , an adjusted estimated OR was

plotted as a function of the combined biases (A-F; blue lines). OR indicates odds ratio, CF indicates correction factor, SFN indicates sequencing false negatives, SFP indicates sequencing false positives, PSF indicates population-specific factor.”

Nevertheless, we do see utility in showcasing the consistent calibration offered by the CF even when PSF is extreme (which may or may not be the case, depending on the frequency and pathogenicity thresholds for the variants in question). Therefore, we have showcased CF calibration on estimated effect size for PSF 0.3 and 1.7 as a supplemental figure (updated Supplementary Figure 2). We have appended this figure and its associated caption below:

“Supplementary Figure 2: Simulated effects of combined SFN rate, SFP rate, and high PSF on estimated OR. Probability of mutation (P) was calculated as a function of the cumulative minor allele frequency for a single gene ($CMAF_{Gene}$: set at 0.05), the true effect size of association (OR), and the 3 major association biases (SFN, SFP, and PSF) according to equation 2. The simulations kept the true OR fixed at 1.3 (black dashed lines) while varying SFN rate (0 to 1), SFP rate (0.1, 0.2, and 0.3), and high PSF (0.3 and 1.7). An estimated OR was calculated for each P (corresponding to a unique combination of the 3 association biases) and plotted as a function of the combined biases (A-F; red lines). CF values were calculated as the ratio of the P for each estimated OR and the original $CMAF_{Gene}$ according to equation 3. Each CF was then used to adjust the $CMAF_{Gene}$ to calculate an adjusted P (P^*) according to equation 4. Using P^* , an adjusted estimated OR was plotted as a function of the combined biases (A-F; blue lines). OR indicates odds ratio, CF indicates correction factor, SFN indicates sequencing false negatives, SFP indicates sequencing false positives, PSF indicates population-specific factor.”

We have also modified our updated main text to reflect the difference in Figure 2 and Supplementary Figure 2

“We also observed that the CF robustly calibrates the estimated OR in the presence of all biases simultaneously (i.e. SFN, SFP, and PSF) and adheres to the same patterns as observed with single bias simulations (Figure 2A-F). That is, the CF adequately calibrates the estimated OR to the true OR in the presence of strong SFN and PSF bias, while providing a slightly conservative correction at high SFP rates (Figure 2C and F). While these simulations demonstrate strong calibration of effect size under realistic PSF bias for rare variants, we show that the extent of calibration remains robust when such bias is more extreme (Supplementary Figure 2A-F).” (Updated main text page 5; lines 122-129)

Mutational load:

1) I agree that iCF is heavily affected by technological artifacts. 2) If iCF is heavily influenced by sequencing technology, it shouldn't be called or interpreted as the mutational load. This is misleading. Although we sometimes use the count of rare deleterious variants in sequenced genomes as a proxy for the mutational load, it is always restricted to the highest-quality regions which are least likely to be affected by sequencing artifacts.

We agree that the term: “mutation load” is mis-leading in this context as we have taken it to be a term that represents both true and artefactual sequencing variants. As such, we have changed all instances of “*mutation load*” to “*total allele count*” in both the updated main text (10 instances) and updated supplementary appendix (9 instances). We believe that the term “*total allele count*” is very generalizable to both true sequencing variants and variants that are artefacts, which thus mitigates any misinterpretation by our paper’s readership.

3) I agree that rare deleterious variants are less likely to be shared between populations. However, the issue is whether the variation in the numbers of rare deleterious variants between cohorts is driven mainly by population genetic forces underlying the mutational load, such as mutation rates or efficiency of purifying selection (namely, effective population size and growth rates, etc). Nelson et al., Gravel et al., or 1000 genomes paper does not address this issue. Kessler et al. is relevant, but I believe it's more an exception as it also states "we do not find significant differences in DNM rate between individuals of European, African, and Latino ancestry, nor across ancestrally distinct segments within admixed individuals."

The reviewer has stated a very fair point. While the degree of deleterious rare variant sharing among even closely related populations is small, the amount to which we can attribute this to population genetic phenomena and/or selective pressure has not been assessed systematically or comprehensively. While this represents an opportunity for future works to assess, the relatively novel ability for researchers to conduct frequency-based annotation by leveraging large-scale sequencing repositories will increase ability to identify ultra-rare variants that are highly specific only to very closely related population groups. We believe that the ability to identify such variants will further substantiate the observation that very rare variants are shared only among very closely related populations.

Power simulation: Given the weak power of RVGRS in real data, I think that thorough and extensive benchmark simulations are necessary to present the utility of current work.

1) TRAPD was one of the first methods leveraging external controls, thus it's understandable that it lacked extensive benchmark simulations.

The reviewer puts forth a very fair point. A robust approach toward evaluating the performance of multiple methodological frameworks is to comparatively apply them on simulated data. We have outlined the methodology and results of our comparative benchmark simulations in our responses to the subsequent reviewer comments. Briefly, we use RV-EXCALIBER and TRAPD methods to evaluate the predictive power of a simulated RVGRS on simulated case status by leveraging the individual-level whole exome sequencing data from the UK Biobank and summary-level data from release 2.0.1 of gnomAD.

2) New polygenic score methods are typically demonstrated to show the improved prediction accuracy in extensive benchmark simulations AND with multiple real phenotypes. I understand applying the method to multiple real phenotypes is not simple to do, though.

We agree with the reviewer that benchmark simulations are an ideal approach to effectively evaluate the comparative predictive power of the RV-EXCALIBER and TRAPD methodological frameworks. We leveraged a probability-based sampling approach to simulate case status in the UK Biobank whole exome sequencing dataset, where the probability of being a case was informed by 1) a fixed regression coefficient for 100 randomly selected genes (that we refer to as the “genes of real effect” or gre for short) and 2) the corresponding delta allele counts for those gre. We fixed the (odds ratio) effect size of the gre at 1.4, 1.6, 1.8, and 2.0 to simulate the real range of effect sizes we identified in our discovery rare variant association test using MIGen cases. We then use half of the simulated “cases” to conduct a discovery gene-based rare variant association test with RV-EXCALIBER and TRAPD and subsequently apply the gene-based effect sizes to the (simulated) validation set to benchmark the power of RV-EXCALIBER-derived RVGRS against TRAPD-derived RVGRS to predict simulated case status.

By assigning case status using probability-based sampling, we are effectively and robustly simulating a novel case phenotype with every simulation we conduct. We also account for varying prevalence of a hypothetical phenotype by simulating over 2 case-prevalence parameters (10% and 20%). We thereafter elected to conduct 10 simulations for each permutation of simulated case prevalence and gre effect size, resulting in 80 total benchmark tests performed. We have provided a detailed methodology for the benchmark simulations below, which have appended in the updated supplementary appendix:

“O. RV-EXCALIBER benchmark simulations

Randomly sampled discovery and validation populations were obtained UK Biobank whole exome sequencing dataset (described in section 7A of the Data Supplement) to derive simulated gene-based effect sizes and construct a simulated RVGRS, respectively. Case status in the discovery and validation populations were ascertained according to a probability-based sampling approach

(without replacement) where the probability of being a “case” (P_i) was informed according to equation 14:

$$P_i = \frac{e^{\beta_0 + \sum_{gre=1}^M \beta_{gre} D_{i,gre}}}{1 + e^{\beta_0 + \sum_{gre=1}^M \beta_{gre} D_{i,gre}}} \quad (14)$$

where β_0 represents the log-odds of the simulated case prevalence to be sampled from the UK Biobank, β_{gre} represents a fixed regression coefficient applied to a random set of 100 “genes of real effect (gre)”, and where $D_{i,gre}$ is the per-individual delta allele count (defined in equation 9) for each of the gre. We set β_0 to correspond to a case prevalence of either 10% or 20% and assigned β_{gre} to correspond to an odds ratio of either 1.4, 1.6, 1.8, or 2.0. A total of 10 randomly sampled gre sets (i.e. 10 sets of 100 gre) were used to generate 10 case populations per β_0 per β_{gre} (i.e. 80 simulated case populations). Each simulated case population was thereafter evenly split into either the discovery or validation set, where the simulated case in the discovery set underwent gene-based rare variant association testing using RV-EXCALIBER and TRAPD. The remaining participants in the UK Biobank that were not selected as cases using probability-based sampling for a given simulation were used as the controls in the validation population. RVGRS were calculated based on the top 10 to top 1,500 (for simulated case prevalence of 10%) and top 10 to top 3,000 (for a simulated case prevalence of 20%) discovery genes. Top genes were evaluated at increments of 10 genes from the top 10 to top 100 genes and increments 100 genes from top 100 to top 1,500 or 3,000 genes) using TRAPD and RV-EXCALIBER-derived discovery gene-based effect sizes, which would weight the $D_{i,g}$ and $OAC_{i,g}$, respectively (see equations 5 and 9, respectively). RVGRS were scaled to mean 0 and SD 1 and underwent univariable logistic regression to predict simulated cases status. A mean regression coefficient was thereafter obtained for a given discovery gene number across the 10 simulations performed at a given β_0 and β_{gre} . The 95% confidence intervals for the mean regression coefficient were obtained using bootstrapping. Odds ratios and their 95% confidence intervals were calculated by taking the natural exponent of the mean regression coefficient and its corresponding confidence interval.” (Updated supplementary appendix pages 26-27; lines 517-546)

3) The reported power gain over TRAPD is small and not significant in the real data. The accuracy of the proposed method should be shown to be significantly better at least in simulations.

We have shown in our extensive benchmark simulation that RV-EXCALIBER-derived RVGRS is appreciably more predictive than TRAPD-derived RVGRS on simulated case status. Indeed, this difference shows greater significance at higher simulated case prevalence (20%) and greater contribution of genetic effect to case status (OR for gre = 1.8, 2.0). We have now noted this result in the results section our updated main text and appended this addition below:

“We also identified that RVGRS generated using gene-based odds ratios derived from RV-EXCALIBER resulted in higher effect estimates and improved precision for CAD association compared to RVGRS calculated using the gene-based odds ratios from TRAPD in the UK Biobank. In fact, the effect size of RV-EXCALIBER’s RVGRS950, which conferred a 1.08-fold (95% CI, 1.40-1.11; $P=2.1 \times 10^{-5}$) increased odds of CAD per SD, represents a >20% increase relative to the

predictive power of the RVGRS with the strongest CAD association according to TRAPD-derived odds ratios, which was found to confer a 1.06-fold (95% CI, 1.03-1.10; $P=4.4 \times 10^{-4}$) increased odds of CAD per SD (Supplementary Table 11). Indeed, the predictive power offered by RV-EXCALIBER-derived RVGRS on simulated case status was shown to be appreciably stronger compared to TRAPD-derived RVGRS, especially for simulated scenarios of higher disease prevalence and greater strength of genetic contribution to case status (Supplementary Figure 13).” (Updated main text pages 12-13; lines 294-305).

We have also appended the figure corresponding to this result (Supplementary Figure 13) and its corresponding figure caption below:

“Supplementary Figure 13: Effect of TRAPD and RV-EXCALIBER-derived RVGRS on simulated case status in the UK Biobank. Benchmark simulations were performed by conducting discovery rare variant association analysis on cases that were ascertained according to a probability-based sampling approach, where probability of being a case was ascertained according to a pre-assigned regression coefficient for 100 “genes of real effect” (gre) and the

delta allele count for the gre (equation 14 in Data Supplement) for a given individual. A total of 10 case sampling simulations were performed across 2 disease prevalence parameters: 10% (A-D); 20% (E-H) and across 4 fixed effect sizes for the gre, corresponding to odds ratios of 1.4 (A and E), 1.6 (B and F) 1.8 (C and G), and 2.0 (D and H). Odds ratios are expressed as the mean odds ratio per 1 SD change in RVGRS on simulated case status, where the mean OR was ascertained for a given number of discovery genes across 10 case simulations for each case prevalence and gre effect size parameter. Shaded regions correspond to bootstrapped 95% confidence interval of the mean odds ratio. Dashed line represents a line of no effect. OR indicates odds ratio, SD indicates standard deviation.”

4) Supp Fig 9 is interesting but cannot replace simulated benchmarks.

Please refer above to Supplementary Figure 13 which illustrates the effect conferred by a RVGRS on simulated case status in the UK Biobank. The RVGRS was informed using gene-based effect sizes ascertained from a simulated discovery population using either the RV-EXCALIBER or TRAPD method. Gene-based effects were thereafter applied to a simulated validation population to construct a simulated RVGRS. We have also appended a workflow for the benchmark simulations below (Supplementary Figure 14) along with its corresponding figure caption:

“Supplementary Figure 14: Workflow for benchmark simulations to assess the predictive power of the RV-EXCALIBER and TRAPD frameworks. For Step 2, β_0 refers to the prevalence of cases in a specific simulation, β_{gre} refers to the $\log(OR_{gre})$ outlined in Step 1, and $D_{i,gre}$ refers to the per-individual delta allele count for the *gre*. A total of 10 simulations were performed per case prevalence parameter (10% and 20%) per effect size for the *gre* (odds ratio = 1.4, 1.6, 1.8, 2.0). In this flow chart, a simulated case prevalence of ~20% (i.e. 10,000/45,850) is shown for illustration. OR indicates odds ratio, *gre* indicates “genes of real effect”, gnomAD indicates genome Aggregation Database, and RVGRS indicates rare variant genetic risk score.”

The variance explained by PRS is typically calculated on the liability scale using AUC (PMID: 20195508). But if the AUC does not improve much with RVGRS, the variance additionally explained by RVGRS would be small on the liability scale as well. NRI is interesting, but there is criticism against it, too (PMID: 26504496). It is very hard for me to see how a linear polygenic model doesn't do very well in terms of R² and liability but has high predictive power if we are considering a typical complex genetic architecture. What is the Nagelkerke R² explained by RVGRS compared to the R² of the full model?

We agree with the reviewer that the net-reclassification index does indeed have disadvantages when assessing the added utility of a predictor variable. The referenced article by Pepe *et al.*, 2015 (DOI: 10.1007/s12561-014-9118-0), shows that even predictor variables that demonstrate poor model fit have significant NRIs in an independent test set (i.e. it is not a product of overfitting). As such, a precursor of a NRI should be assessing a likelihood ratio test (LRT) between the full and reduced model (where the full model in this case is RVRGS + CVGRS + AGE + AGE² + SEX + PC1-20 and the reduced model is CVGRS + AGE + AGE² + SEX + PC1-20). We confirmed the LRT to be significant between the full model and reduced model (LR $\chi^2 = 18.72$; $P < 0.0001$), so this precludes obtaining a significant NRI from a predictor that did not benefit model fit.

We did discuss a Nagelkerke's R² in our previous rebuttal in order to assess the additional population-level variance explained when RVGRS is included in the model (see full and reduced models above). We identified that the RVGRS950 and RVGRS950^{LDLR} each individually explained an additional 0.1% of population variance in CAD affection status.

We reported this in our Results section as follows:

“Additionally, we identified that the added variance explained by both RVGRS950 and RVGRS950^{LDLR} to be 0.1% when compared to a logistic model containing CVGRS, FRS, age, age², sex, and the first 20 principal components of ancestry alone.” (Updated main text page 14; lines 323-325)

We do note that despite weak increase in explained variance, the RVGRS is best tailored toward demarcating individuals at high risk of disease (i.e. ≥ 2 -fold increase odds of CAD). Indeed, we observed that the RVGRS identifies 1.5% of the population at such risk. We have reported this in our Discussion section as follows:

“However, the RVGRS only explained 0.1% of variance in CAD affection status, which is lower compared to traditional CVGRS for CAD²⁶. This reason for this large difference in explained variance are two-fold. First is the substantially larger discovery sample size used to generate weights for CVGRS, which will necessarily increase model calibration when assessing risk an independent validation population. Second is the difference in exposure frequency between the RVGRS and CVGRS, wherein the component variants used to inform the RVGRS are necessarily observed less commonly in any given population. As such, the RVGRS (as it stands) will be less powered to detect population-level variances, relative to established CVGRS for CAD²⁶. Despite this, we maintain that the RVGRS has potential usefulness to identify high-risk individuals, in

contrast to explaining a high degree of population-level variance.” (Updated main text page 19; lines 439-448).

and

“Nevertheless, our work bridges the gap between rare and common variants under a unified polygenic framework. Despite the marked difference in case sample size, we show that a RVGRS confers high risk (i.e. ≥ 2 -fold) for early CAD in 1.5% of the population and remains independent of Mendelian effects, clinical risk factors, and CVGRS. Notwithstanding its independent effect on CAD and ability to demarcate individual-level risk, we stress that a RVGRS should be used as an adjunct genetic variable that is used in conjunction with Mendelian variants and CVGRS to enhance the overall clinical utility of genetic factors.” (Updated main text page 20; lines 470-476).

For the reviewer’s reference, the Nagelkerke’s R2 for the base/reduced model (i.e. CVGRS + AGE + AGE² + SEX + PC1-20) and full model (i.e. RVGRS + CVGRS + AGE + AGE² + SEX + PC1-20) was 11.2% and 11.3% , respectively (i.e. 0.1% added variance explained by RVGRS).

I agree that RVGRS and CVGRS need to be combined for higher accuracy, but it should be noted that the heritability explained by rare protein-altering variants might be small for MI. The predictive power of RVGRS is constrained by the heritability.

This is an excellent point brought up by the reviewer and we have now noted it in our Discussion section as follows:

*“It is important to note that the risk estimates for high polygenic risk derived from RVGRS and CVGRS cannot be readily compared due **both to the limited complex trait heritability explained by rare variants (relative to common variants)** and to the extreme discrepancy in discovery sample size used to compute effect estimates to weight alleles in validation samples.” (Updated main text page 20; lines 464-467)*

Reviewers' Comments:

Reviewer #3:

Remarks to the Author:

The authors addressed all of my concerns.